# Turbulence in Focus:
# Benchmarking Scaling Behavior of 3D Volumetric Super-Resolution with BLASTNet 2.0 Data

**Wai Tong Chung**[*,1]**, Bassem Akoush**[1]**, Pushan Sharma**[1]**, Alex Tamkin**[1]**,**
**Ki Sung Jung**[2]**, Jacqueline H. Chen**[2]**, Jack Guo**[1]**, Davy Brouzet**[1]**,**
**Mohsen Talei**[3]**, Bruno Savard**[4]**, Alexei Y. Poludnenko**[5]**, Matthias Ihme**[*,1,6]
[1]Stanford University, [2]Sandia National Laboratory, [3]University of Melbourne,
[4]Polytechnique Montréal, [5]University of Connecticut, [6]SLAC National Accelerator Laboratory

## Abstract

Analysis of compressible turbulent flows is essential for applications related to propulsion, energy generation, and the environment. Here, we present **BLASTNet 2.0**, a **2.2 TB** network-of-datasets containing **744 full-domain samples** from **34 high-fidelity direct numerical simulations**, which addresses the current limited availability of 3D high-fidelity reacting and non-reacting compressible turbulent flow simulation data. With this data, we benchmark a total of **49 variations** of five deep learning approaches for 3D super-resolution – which can be applied for improving scientific imaging, simulations, turbulence models, as well as in computer vision applications. We perform neural scaling analysis on these models to examine the performance of different machine learning (ML) approaches, including two scientific ML techniques. We demonstrate that (i) predictive performance can scale with model size and cost, (ii) architecture matters significantly, especially for smaller models, and (iii) the benefits of physics-based losses can persist with increasing model size. The outcomes of this benchmark study are anticipated to offer insights that can aid the design of 3D super-resolution models, especially for turbulence models, while this data is expected to foster ML methods for a broad range of flow physics applications. This data is publicly available with download links and browsing tools consolidated at `https://blastnet.github.io`.

## 1 Introduction

In recent years, machine learning (ML) has offered new modeling approaches for natural and engineering sciences. For example, 5 PB of ERA5 data [1] was employed towards GraphCast [2], an ML model that outperformed conventional weather modeling techniques. As such, efforts in curating large datasets for scientific ML have been growing across numerous domains including agricultural science [3], geophysics [4], and biology [5]. In many of these examples, the volume of data can significantly exceed the free limit of open-source repositories such as Kaggle (100 GB) – typically used to store and share language and image data – due to potentially high dimensions within scientific data. As such, significant resources are required for building and maintaining data storage capabilities, either through institutional collaborations [3, 5, 6] or cloud service providers [7].

In previous work [8, 9], we proposed the Bearable Large Accessible Scientific Training Network-of-datasets (BLASTNet), a cost-effective community-driven weakly centralized framework (see Section 3.1) that utilizes Kaggle for increasing access to scientific data, which provided access to 110 full-domain samples from 10 configurations (225 GB) of 3D high-fidelity flow physics direct

---

[*]Corresponding authors: {`wtchung,mihme`}`@stanford.edu`

37th Conference on Neural Information Processing Systems (NeurIPS 2023) Track on Datasets and Benchmarks.

numerical simulations (DNS). In this work, we present the updated BLASTNet 2.0 dataset, which is extended to 744 full-domain samples from 34 DNS configurations, as shown in Figure 1. This dataset aims to address limitations in data availability for compressible turbulent non-reacting and reacting flows, which is found in automotive [10, 11], propulsion [12, 13], energy [14, 15], and environmental [16, 17] applications. BLASTNet data has previously been employed for solving ML tasks related to dimensionality reduction, regime classification, and turbulence-chemistry closure modeling [9]. Beyond this, BLASTNet is potentially suited for ML problems involving predictions of physical quantities found in turbulent non-reacting and reacting flows, which can also involve inverse problems [18, 19] and physics discovery [20, 21].

In this work, we demonstrate the utility of BLASTNet data for 3D super-resolution (SR) of turbulent flows. From BLASTNet 2.0, we pre-process DNS data to form the Momentum128 3D SR dataset for benchmarking this task. To this end, we:

- Curate BLASTNet 2.0, a diverse public 3D compressible turbulent flow DNS dataset.
- Benchmark performance and cost of five 3D ML approaches [19, 22–25] for SR with this publicly accessible dataset.
- Show that SR model performance can scale with the logarithm of model size and cost.
- Demonstrate the persisting benefits of a popular physics-based gradient loss term [19] with increasing model size.

We provide an overview of related efforts in Section 2. In Section 3, we provide information on the BLASTNet 2.0 and Momentum128 3D SR datasets. Our benchmark setup is described in Section 4, with results discussed in Section 5, before the conclusions in Section 6.

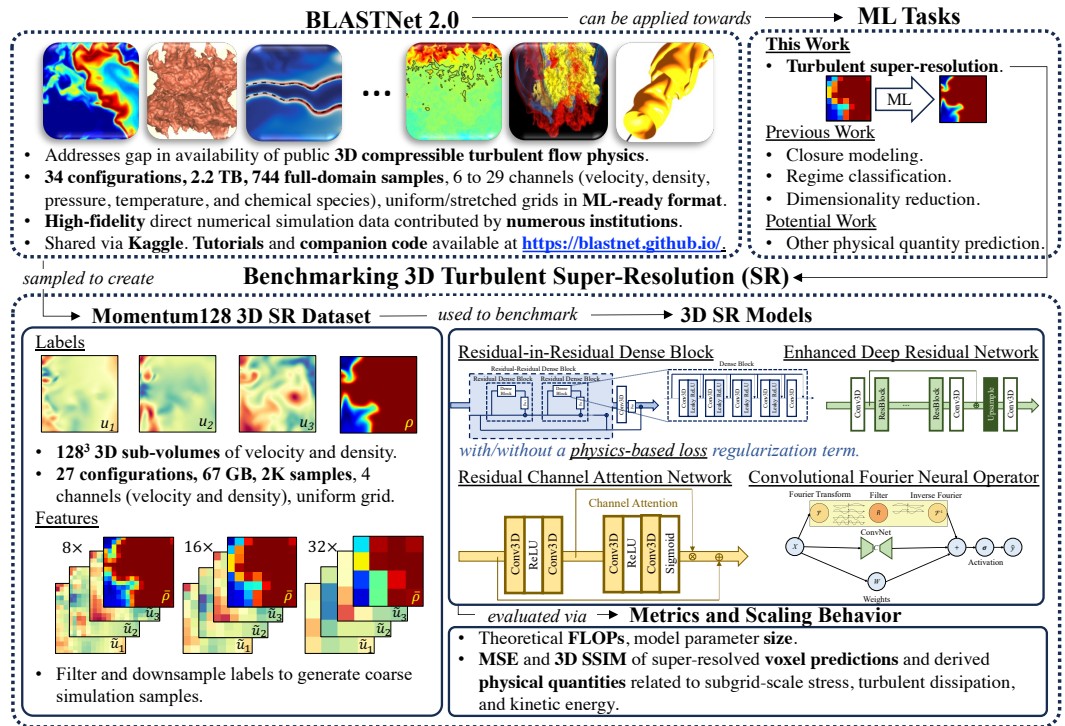

Figure 1: Summary of this study.

## 2   Related Work

**Flow Physics Simulation Datasets**   Numerical simulations accurately describe flow physics, as long as the simulation grid resolves the smallest lengthscales associated with turbulent dissipation [26]. With up to $\mathcal{O}(10^9)$ voxels, $\mathcal{O}(10^6)$ core-hours of simulation time, and $\mathcal{O}(10^4)$ cores on parallel computing facilities [27–30], high-fidelity DNS of many real-world flows cannot be performed

due to prohibitive costs. Thus, it is common to employ coarser grids with large-eddy simulations (LES) [15, 31], or by only evolving time-/ensemble-averaged quantities with Reynolds-averaged Navier-Stokes (RANS) simulations [32, 33] – both of which rely on turbulence models that can be discovered from DNS data. DNS data is also useful for applications involving scientific imaging [34], spatio-temporal modeling [35], and solving inverse problems [18]. As shown in Table 1, many existing flow simulation datasets focus on LES and RANS simulations due to data storage constraints. McConkey et al. [32] released a dataset for improving turbulence models in incompressible non-reacting RANS. AirfRANS [33] provides both 2D incompressible and compressible non-reacting RANS data, specifically on airfoil configurations. For reacting flows, Huang [31] released a 2D LES dataset for developing reduced-order models. The largest flow physics dataset, the Johns Hopkins Turbulence Database (JHTDB) [36], provides 3D DNS data from turbulent incompressible non-reacting flow simulations. Since these datasets are either 2D, incompressible, or non-reacting, they are not suitable for many applications involving aerodynamics, propulsion, or chemical processes. This is one reason why ML studies involving these applications employ self-generated inaccessible datasets [37, 38] – introducing challenges to open model evaluation. To address these gaps, we curate BLASTNet 2.0, a 3D turbulent compressible reacting and non-reacting flow dataset, which targets a balance of diversity, fidelity, and size.

Table 1: Comparison of BLASTNet 2.0 (in **bold**) with selected flow simulation datasets.

| Datasets | BLASTNet 2.0 | BLASTNet 1.0 [9] | JHTDB [36] | McConkey et al. [32] | AirfRANs [33] | Huang [31] |
|---|---|---|---|---|---|---|
| Size [TB] | **2.2** | 0.225 | 490 | 0.014 | 0.066 | 0.209 |
| Configs. | **34** | 10 | 10 | 29 | 1,000 | 1 |
| Full-domain Samples | **744** | 120 | 15,791 | 29 | 1,000 | 30,000 |
| Fidelity | **DNS** | | DNS | RANS | RANS | LES |
| Spatial Dimensions | **3D** | | 3D | 2D,3D | 2D | 2D |
| Primitive Variables[1] | $\boldsymbol{u_i, p, T, \rho, Y_k}$ | | $u_i, p, \rho, b_i$ | $\langle u_i \rangle, \langle p \rangle$ | $\langle u \rangle_i, \langle p \rangle, \langle T \rangle, \langle \rho \rangle$ | $\widetilde{u}_i, \overline{p}, \widetilde{T}, \overline{\rho}, \widetilde{Y}_k$ |
| Number of PDEs[2] | **5 to 27** | | 4 to 7 | 3 to 4 | 4 to 5 | 7 |
| Compressible Flow? | **Yes** | | No | No | Yes[3] | Yes |
| Multi-physics? | **Reactions**[4] | | MHD[5] | No | No | Reactions |
| Host | **Kaggle** | | SciServer [6] | Kaggle | Sorbonne | UMich |
| Format | **fp32 binaries** | | `.h5` | `.npy` | `.vtu,.vtp` | `ascii` |

**SR for Turbulent Flows**  Within experimental flow measurements, deep learning-based SR approaches have been employed towards improving Schlieren [40], particle-image-velocimetry [41], and tomography [42] techniques, which can often be limited by resolution restrictions. In many practical engineering analyses, coarse-grained simulations with under-resolved grid sizes are often used to bypass the extensive costs of fully-resolved DNS. A dominant source of error from this approach involves missing physics that arise from the under-resolved grid, as will be detailed in Section 4.2. While algebraic turbulence models have been traditionally used to represent the under-resolved physics [43, 44], specific algebraic models are effective only in specific flow configurations [45]. SR and related upsampling approaches have been proposed as a versatile alternative for correcting under-resolved information from coarse-grid simulations between numerical time-stepping [37, 46, 47], which would require considerations of real-time inferencing. Studies on turbulent SR have focused on demonstrating feasibility [37, 41, 46–48], mostly by modifying existing image SR models. Due to memory constraints, many of these studies focus on 2D configurations [41, 46, 48], with 3D SR investigations only demonstrated recently [37, 47]. As such, there has not yet been a detailed and reproducible benchmark study comparing SR models for 3D turbulent flows [38, 49].

---

[1]Filtered and time-/ensemble-averaged quantity $\phi$ is expressed with $\overline{\phi}$ and $\langle \phi \rangle$, respectively. In compressible LES, Favre-filtering $\widetilde{\phi}$ [39] is typically used. $T$ refers to temperature; more definitions in Equation (1).

[2]PDEs for continuity, energy, momentum, chemical species, and magnetic field $b_i$.

[3]This dataset contains a blend of compressible and incompressible flow samples.

[4]Majority are reacting flows. Some non-reacting flows are present.

[5]Only one of the ten configurations consider magneto-hydrodynamics (MHD).

**Scientific ML Approaches** Scientific ML approaches can involve developing custom architectures with implicit biases that suit specific problems and modifying loss functions with constraints related to governing equations [50]. Firstly, model architectures such as NUNet [51], MeshGraphNet [52], and Fourier Neural Operators (FNO) [53] employ graph and spectral convolution layers to ensure that flow predictions are mesh invariant. While FNOs have been successfully employed towards canonical laminar flow configurations, previous studies have demonstrated that these models are insufficiently expressive for complex configurations, involving turbulent [35] and multi-physics [25] flows. This has been attributed to regularization properties of spectral convolution layers by the FNO's original developers [25]. Recently, convolution FNO (Conv-FNO) [25] models have been proposed to ameliorate these underfitting issues by embedding convolutional blocks within FNO blocks. Secondly, modifying the loss functions can involve adding a regularization term based on the residuals of the entire governing equations [54]. A softer approach involves regularizing with individual operators within the governing equation (such as continuity, advection, diffusion, and source terms) [37]. In this work, we benchmark models based on both loss function and architecture approaches, *i.e.*, a model with a gradient-based loss function [19] (which biases ML optimization towards turbulence applications; see Appendix F.3) and a Conv-FNO model, respectively.

**SR Benchmarks** SR via deep learning has been subjected to numerous competition and benchmark studies that target various aspects of 2D image SR, including 2K-images [55], night photography [56], spectral recovery [57], and satellite images [58]. For 3D SR, benchmarks for video [59], medical resonance imaging [60], and 3D microscopy applications [61] have been performed. Another method for studying the behavior of deep learning models involves the construction of empirical scaling relationships [62]. This type of analysis is useful for studying resource requirements of large language models, and was briefly been employed for studying 2D SR with a U-Net model [63]. Given the potential real-time computing applications of turbulent SR, this analysis provides useful information on the relationship between model size, cost, and predictive performance.

## 3 Dataset

### 3.1 BLASTNet 2.0

BLASTNet 2.0 consists of turbulent compressible flow DNS data, on Cartesian spatial grids, generated by solving governing equations for mass, momentum, energy, and chemical species, respectively:

$$\partial_t \rho + \nabla \cdot (\rho \mathbf{u}) = 0 \,, \tag{1a}$$

$$\partial_t (\rho \mathbf{u}) + \nabla \cdot (\rho \mathbf{u} \otimes \mathbf{u}) = -\nabla p + \nabla \cdot \boldsymbol{\tau} \,, \tag{1b}$$

$$\partial_t (\rho e^t) + \nabla \cdot [\mathbf{u}(\rho e^t + p)] = -\nabla \cdot \mathbf{q} + \nabla \cdot [(\boldsymbol{\tau}) \cdot \mathbf{u}] \,, \tag{1c}$$

$$\partial_t (\rho Y_k) + \nabla \cdot (\rho \mathbf{u} Y_k) = -\nabla \cdot \mathbf{j}_k + \dot{\omega}_k \,, \tag{1d}$$

with density $\rho$, velocity vector $\mathbf{u}$, pressure $p$, specific total energy $e^t$, stress tensor $\boldsymbol{\tau}$, and heat flux $\mathbf{q}$. $Y_k$, $\mathbf{j}_k$, and $\dot{\omega}_k$ are the mass fraction, diffusion flux, and source term for chemical species $k = [1, N_s - 1]$, where $N_s$ is the number of species. Molecular fluxes are typically modeled using the mixture-averaged diffusion model.

The BLASTNet 2.0 dataset is developed with these properties in mind:

**Fidelity** All DNS data is collected from well-established numerical solvers [29, 64–67] with spatial discretization schemes ranging from 2nd- to 8th-order accuracy, while time-advancement accuracy range from 2nd- to 4th-order. Low-order schemes require finer discretizations compared to high-order schemes, to achieve similar accuracy and numerical stability [68]. However, all simulations are spatially resolved to the order of the Kolmogorov lengthscale, ranging from 3.9 to 41 $\mu$m depending on the configuration, with a corresponding temporal discretization that ensures numerical stability.

**Size and Diversity** BLASTNet 2.0 contains a total of 744 full-domain samples (2.2 TB) from a diverse collection of 34 simulation configurations: non-reacting decaying homogeneous isotropic turbulence (HIT) [20], reacting forced HIT [67], two parametric variations of reacting jet flows [29], six configurations of non-reacting transcritical channel flows [69], a reacting channel flow [28], a partially-premixed slot burner configuration [27], and 22 parametric variations (with different turbulent and chemical timescales) of a freely-propagating flame configuration [70].

**Community-involvement**   BLASTNet 2.0 consists of data contributions from six different institutions. As mentioned in Appendix C, our long-term vision and maintenance plan for this dataset involves seeking additional contributions from members of the broader flow community.

**Cost-effective Storage, Distribution, and Browsing**   To circumvent Kaggle storage constraints, we partition the data into a network of $< 100$ GB subsets, with each subset containing a separate simulation configuration. This partitioned data can then be uploaded as separate datasets on Kaggle. To consolidate access to this data, all Kaggle download links are presented in `https://blastnet.github.io`, with the inclusion of a `bash` script for downloading all data through the Kaggle API. In addition, Kaggle notebooks are attached to each subset to enable convenient data browsing on Kaggle's cloud computing platform. This approach enables cost-effective distribution of scientific data that adheres to FAIR principles [71], as further detailed in Appendix C.

**Consistent Format**   Data, generated from different numerical solvers, initially exists in a range of formats (`.vtk`, `.vtu`, `.tec`, and `.dat`) that are not readily formatted for training ML models. Thus, all flowfield data are processed into a consistent format – little-endian single-precision binaries that can be read with `np.fromfile`/`np.memmap`. The choice of this data format enables high I/O speed in loading arrays. We provide `.json` files that store additional information on configurations, chemical mechanisms and transport properties. See Appendix D for more details.

**Licensing and Ethics**   All data is generated by the present authors and licensed via CC BY-NC-SA 4.0. Other than the contributors' names and institutions, no personal-identifiable information is published in this data. No offensive content is published with this flow physics dataset. Further discussion on negative impact is provided in Section 6.

### 3.2   Momentum128 3D SR Dataset

BLASTNet 2.0 is further processed for training due to constraints in (i) memory and (ii) grid properties. Currently, the single largest sample (92 GB) in BLASTNet 2.0 contains 1.3B voxels and 15 channels, which cannot fit into typical GPU memory. In addition, the spatial grid is stretched depending on the resolution requirements of the flow domain. As shown in Figure 1, we circumvent these two issues by sampling $128^3$ sub-volumes of density $\rho$ and velocity $\mathbf{u}$ from the uniform-grid regions from all BLASTNet data. This results in 12,750 sub-volume samples (427 GB). We choose this sub-volume size to enable $32\times$ SR (the resulting feature sub-volume is $4^3$ which is larger than a kernel size of 3), while maintaining a low memory footprint. In order to develop a compressible turbulence benchmark dataset that can be easily downloaded, we select 2,000 sub-volumes to form a 67 GB dataset that can fit into a single Kaggle repository. To ensure that these 2,000 samples are representative of the different flows encountered in each configuration, we:

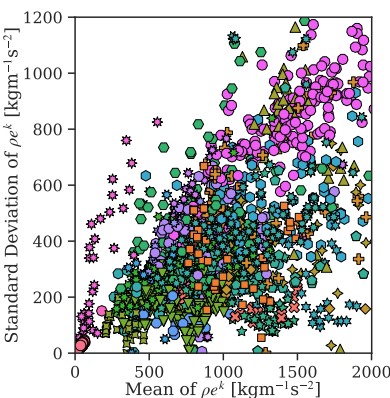

Figure 2: Statistics of the specific kinetic energy $\rho e^k$ of each $128^3$ sub-volume in the Momentum128 3D SR dataset. Marker type represents DNS configuration. See detailed legend in Appendix E.1.5

1. Extract mean, variance, skewness, and kurtosis (statistical moments for characterizing turbulence [26]) from the three velocity components of 12,750 sub-volumes.

2. Apply k-means clustering with the elbow method (using the statistical moments as features) to partition the sub-volumes in 18 clusters.

3. Select 2,000 samples while ensuring that the proportion of clusters are well-balanced.

The resulting sub-volumes form the labels of BLASTNet Momentum128 3D SR dataset. Figure 2 demonstrate the mean and standard deviation of the specific kinetic energy:

$$\rho e^k = \rho(u_1{}^2 + u_2{}^2 + u_3{}^2)/2\,, \tag{2}$$

which we use to characterize all channel variables. Each distinct marker represents a different simulation configuration. Since flows from the same configuration possess similar statistics, the different configurations from BLASTNet 2.0 can result in a dataset with a variety of flow conditions.

Due to stochastic and chaotic nature of turbulence [72], it is not possible to obtain matching pairs of coarse (also known as implicit LES [45]) and fine DNS data. Thus, we employ a canonical method [37, 45, 73] for obtaining implicit LES surrogates, known as finite-volume optimal LES [74] (see Appendix E.1.6 on their validity). Specifically, we Favre-filter [39] and downsample the labels by 8, 16, and $32\times$ to obtain a representative range of coarse resolution samples (LES is typically an order of magnitude coarser than DNS [26, 75]) to generate inputs for turbulent SR:

$$\widetilde{\phi} = \frac{1}{\overline{\rho}V_f} \int_{V_f} \rho\phi \, dV_f \tag{3}$$

where $\overline{\phi}$ denotes a uniform-filtered quantity, $\widetilde{\phi}$ is a Favre-filtered quantity, and $V_f$ is a subvolume with the size of the filter width. In our SR dataset, the channels of each label correspond to $\phi = \{\rho, u_1, u_2, u_3\}$, while the feature channels consist of $\phi_f = \{\overline{\rho}, \widetilde{u}_1, \widetilde{u}_2, \widetilde{u}_3\}$. For the purpose of the present benchmark study, we further split the 2,000 sub-volumes as follows:

**Train, Validation, and Baseline Test Sets**  80:10:10 split via random selection with a uniform distribution. The training set contains 1,382 samples, and both validation and baseline test sets contain 173 samples each.

**Parametric Variation Set**  A 144-sample subset for model evaluation from an unseen parametric variation configuration with approximately $15\%$ higher mean velocities and velocity fluctuations than the train, validation, and baseline test sets.

**Forced HIT Set**  A 128-sample subset for model evaluation from an unseen flow type (forced HIT) with 30-fold higher pressure and 34-fold lower velocity fluctuations.

## 4  Benchmark Configuration

### 4.1  Models and Methods

As shown in Figure 1, three well-studied 2D ResNet-based [76] SR models are modified from their original repositories for 3D SR: (i) Residual-in-Residual Dense Block (RRDB) [22], (ii) Enhanced Deep Residual Super-resolution (EDSR) [23], and (iii) Residual Channel Attention Networks (RCAN) [24]. Convolution networks possess inductive biases that are suitable for problems involving spatial grids such as in flow physics [50, 77]. We choose to study these models due to their differences in architecture paradigms. Specifically, RRDB employs residual layers within residual layers; EDSR features an expanded network width; RCAN utilizes long skip connections and channel attention mechanisms. In addition, we consider two additional scientific ML approaches: (i) a Conv-FNO model [25], modified for SR (see details in Appendix F.2), and (ii) an RRDB model regularized with the weighted MSE of the gradient of channel variables $\lambda\Delta^2 \sum_{k=1}^{3} \mathrm{MSE}[(\nabla\hat{\phi})_k, (\nabla\phi)_k]$ [19] to the loss $(1 - \lambda)\mathrm{MSE}(\hat{\phi}, \phi)$, where $\Delta$ is the distance between each voxel. Details on the gradient-based loss and its weighting factor $\lambda$ are provided in Appendix F.3. We compare model predictions with a baseline approach, *i.e.*, tricubic interpolation. To investigate the scaling behavior of the model architectures, we vary the number of parameters by changing the network depth and width.

Similar to other turbulent SR studies [37, 46], all models are trained with mean-squared-error (MSE) loss, unless otherwise stated. For evaluation, we select models with the best MSE after training for 1,500 epochs with a batch size of 64 across 16 Nvidia V100 GPUs. Learning rate is initialized at `1e-4`, and halved every 300 epochs. Both the number of training iterations and learning scheduling are chosen to match other SR studies [22–24] and are found to be sufficient for the SR predictions, as will be shown in Section 5. All other hyperparameters are maintained from their original studies, with He initialization [78] used on all initial model weights. Data augmentation is performed via variants of random rotation and flip – modified to ensure that augmented data remains consistent with continuity (Equation (1a)). Training is performed with automatic mixed-precision from `Lightning 1.6.5` [79]. Prior to training, data is normalized with means and standard deviations of density and velocity extracted from the train set. During evaluation, this normalization resulted in poor accuracy for the Forced HIT set, due to the significantly different magnitudes of density and velocity. However, Section 5 will show that good performance can be achieved when normalization is performed with the mean and standard deviation of each distinct evaluation set. Thus, all evaluation sets are normalized

with their own mean and standard deviation, prior to testing. All 40 model variations are trained with three different seeds, resulting in a total computational cost of approximately 15,000 GPU-hours on the Lassen Supercomputer [80]. Further information on model hyperparameters, data augmentation, training, normalization, as well as links to code and model weights are found in Appendix A, F, and G.

## 4.2 Metrics

We compare the performance of each model by examining local and global quantities of each sample. For the local quantities, we employ Metrics $= \{\text{SSIM}, \text{NRMSE}\}$, where SSIM is the 3D extension [81] of the structural similarity image measure [82] and NRMSE is the normalized root-mean-squared error (see Appendix F.1). For quantities with multiple channels:

$$\text{Metric}_{\rho,\mathbf{u}} \equiv \frac{1}{4}\left[\text{Metric}(\hat{\rho},\rho) + \sum_{i=1}^{3}\text{Metric}(\hat{u}_i, u_i)\right], \tag{4a}$$

$$\text{Metric}_{sgs} \equiv \frac{1}{3}\sum_{k=1}^{3}\text{Metric}[(\nabla\cdot\hat{\boldsymbol{\tau}}^{\text{sgs}})_k, (\nabla\cdot\boldsymbol{\tau}^{\text{sgs}})_k]. \tag{4b}$$

with $\hat{\phi}$ denoting an arbitrary predicted quantity. SSIM is a common image metric, but has also become a popular ML metric for evaluating flow simulations due to its employment of mean, variance, and covariance quantities – suited for evaluating the statistical nature of turbulence [26, 83, 84]. In addition, this metric is intuitive for both readers familiar and unfamiliar with turbulent flows – SSIM of 0 denotes dissimilar fields while SSIM of 1 denotes highly similar fields. Metric$_{\rho,\mathbf{u}}$ evaluates each channel of the predictions via macro-averaging. To measure the suitability of SR for turbulence modeling in coarse-grid simulations, we measure Metric$_{sgs}$, which evaluates the predicted divergence of the subgrid-scale (SGS) stress $\nabla\cdot\boldsymbol{\tau}^{\text{sgs}}$. $\nabla\cdot\boldsymbol{\tau}^{\text{sgs}}$ represents physics information lost during coarse-graining, and originates from the Favre-filtered/LES momentum equation (Equation (1b)):

$$\partial_t(\overline{\rho}\widetilde{\mathbf{u}}) + \nabla\cdot(\overline{\rho}\widetilde{\mathbf{u}}\otimes\widetilde{\mathbf{u}}) = -\nabla\overline{p} + \nabla\cdot(\overline{\boldsymbol{\tau}} + \boldsymbol{\tau}^{\text{sgs}}), \quad \text{where}$$

$$\tau^{\text{sgs}} = \overline{\rho}(\widetilde{\mathbf{u}\otimes\mathbf{u}} - \widetilde{\mathbf{u}}\otimes\widetilde{\mathbf{u}}). \tag{5}$$

We evaluate global physical properties of ML predictions by considering the NRMSE$_{\{E^k,\varepsilon\}}$ of turbulent dissipation rate $\varepsilon$ (rate of conversion of turbulent kinetic energy to heat) and volume-averaged kinetic energy $E^k$ (momentum component in energy conservation of a fixed control volume):

$$E^k = \frac{1}{V_s}\int_V \rho e^k \, dV, \tag{6a}$$

$$\varepsilon = \frac{1}{V}\int_V \frac{\boldsymbol{\tau}}{\rho}:\nabla\mathbf{u} \, dV, \tag{6b}$$

with sample volume $V$ and velocity fluctuation $\mathbf{u}'$.

## 5 Experiment Results

We summarize SSIMs of RRDB, EDSR, RCAN, and Conv-FNO in Table 2, along with model parameters $N_p$ and inferencing cost (in FLOPs for a batch size of 1; see Appendix F.6 for details). The 8× SR models shown here possess the best SSIMs across different sizes for a given model approach, as shown in Appendix G.1. Models with the same network depth and width, are then initialized and trained for 16 and 32× SR. For 8 and 16× SR, RRDB (with gradient loss) performs the best across most of the metrics and evaluation sets, with RCAN demonstrating the highest SSIM$_{\rho,\mathbf{u}}$ at 8× SR. At 32× SR, all shown models exhibit lower SSIM$_{\rho,\mathbf{u}}$ than tricubic interpolation in the baseline test set, indicating that SR is difficult to learn at high ratios. However, all models exhibit higher SSIM$_{sgs}$ than tricubic interpolation for all SR ratios. This indicates that SR models may still be useful for turbulence modeling at high SR ratios. Figure 3, demonstrates that model predictions of specific kinetic energy $\rho e^k$ (Equation (2)) (a physical quantity that combines predictions of all four channels) from all models presented in Table 2 increasingly lose fine turbulent structures as SR ratio increase. Nevertheless, when compared to tricubic interpolation, the SR models can still recover the

magnitudes of the energy at these SR ratios. Further examination of the NRMSE metrics from the $8\times$ SR models (see Appendix G.1 for $16, 32\times$ SR), in Table 3, also demonstrates that all ML models significantly outperform tricubic interpolation on baseline test and forced HIT sets. Here, gradient loss RRDB performs best in most of the metrics. However, EDSR outperforms with $\text{NRMSE}_{E^k}$, as the gradient loss only offers minor improvements to $E^k$.

Table 2: Comparison of SSIM of five models at three SR ratios, with tricubic interpolation. Mean and standard deviation from three seeds are reported here. **Bold** term represents best mean.

| Models | Baseline Test Set | | Param. Variation Set | | Forced HIT Set | | Size | Cost |
|---|---|---|---|---|---|---|---|---|
| | $\uparrow\text{SSIM}_{\rho,\mathbf{u}}$ | $\uparrow\text{SSIM}_{sgs}$ | $\uparrow\text{SSIM}_{\rho,\mathbf{u}}$ | $\uparrow\text{SSIM}_{sgs}$ | $\uparrow\text{SSIM}_{\rho,\mathbf{u}}$ | $\uparrow\text{SSIM}_{sgs}$ | $N_p$ | $\downarrow$GFLOPs |
| Tricubic 8$\times$ | 0.820 | 0.431 | 0.800 | 0.418 | 0.951 | 0.711 | – | **23** |
| RRDB 8$\times$ | 0.907±0.003 | 0.715±0.004 | 0.898±0.003 | 0.755±0.002 | 0.997±0.000 | 0.891±0.003 | 50.2M | 1430 |
| (+ Grad. Loss) | **0.936±0.003** | **0.802±0.003** | **0.929±0.001** | **0.825±0.001** | 0.998±0.000 | **0.944±0.005** | | |
| EDSR 8$\times$ | 0.928±0.004 | 0.748±0.012 | 0.916±0.005 | 0.775±0.010 | 0.999±0.000 | 0.937±0.005 | 34.6M | 2122 |
| RCAN 8$\times$ | 0.928±0.000 | 0.753±0.002 | 0.916±0.001 | 0.778±0.001 | **0.999±0.000** | 0.941±0.003 | 16.4M | 671 |
| Conv-FNO 8$\times$ | 0.846±0.016 | 0.566±0.019 | 0.845±0.011 | 0.614±0.015 | 0.993±0.001 | 0.845±0.008 | 33.0M | 1276 |
| Tricubic 16$\times$ | 0.652 | 0.175 | 0.620 | 0.173 | 0.876 | 0.432 | – | **23** |
| RRDB 16$\times$ | 0.724±0.001 | 0.506±0.004 | 0.700±0.001 | 0.512±0.002 | 0.971±0.000 | 0.805±0.003 | 50.3M | 1074 |
| (+ Grad. Loss) | **0.739±0.008** | **0.554±0.001** | **0.719±0.004** | **0.556±0.002** | **0.973±0.000** | **0.816±0.001** | | |
| EDSR 16$\times$ | 0.716±0.005 | 0.477±0.018 | 0.693±0.005 | 0.481±0.019 | 0.969±0.001 | 0.783±0.008 | 37.8M | 1944 |
| RCAN 16$\times$ | 0.672±0.039 | 0.408±0.066 | 0.665±0.024 | 0.415±0.058 | 0.961±0.009 | 0.737±0.050 | 17.3M | 573 |
| Conv-FNO 16$\times$ | 0.629±0.020 | 0.343±0.027 | 0.640±0.013 | 0.355±0.022 | 0.951±0.006 | 0.690±0.022 | 34.6M | 1068 |
| Tricubic 32$\times$ | **0.508** | 0.060 | 0.476 | 0.087 | 0.758 | 0.156 | – | **23** |
| RRDB 32$\times$ | 0.503±0.001 | **0.194±0.005** | 0.482±0.000 | 0.186±0.006 | 0.845±0.001 | 0.494±0.011 | 50.4M | 1030 |
| (+ Grad. Loss) | 0.505±0.001 | 0.184±0.009 | **0.483±0.001** | **0.188±0.002** | **0.850±0.000** | **0.516±0.012** | | |
| EDSR 32$\times$ | 0.502±0.004 | 0.173±0.006 | 0.481±0.002 | 0.187±0.004 | 0.845±0.001 | 0.463±0.005 | 40.9M | 1921 |
| RCAN 32$\times$ | 0.473±0.006 | 0.168±0.007 | 0.469±0.002 | 0.185±0.005 | 0.837±0.003 | 0.448±0.012 | 18.2M | 561 |
| Conv-FNO 32$\times$ | 0.476±0.004 | 0.155±0.012 | 0.470±0.001 | 0.178±0.003 | 0.842±0.002 | 0.435±0.013 | 36.2M | 1023 |

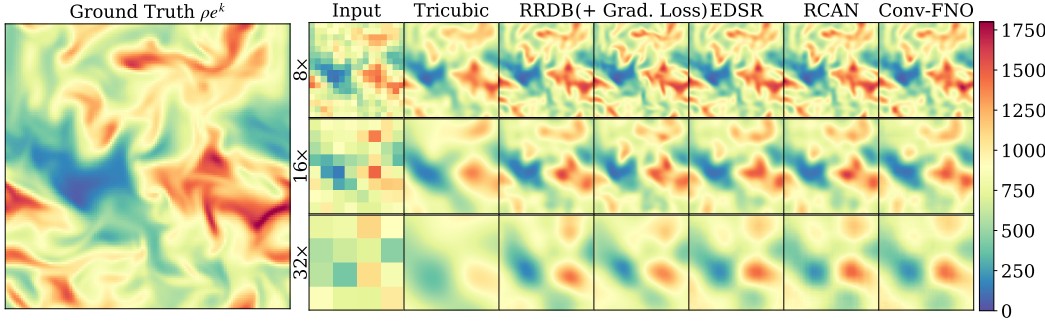

Figure 3: Specific kinetic energy $\rho e^k$ prediction of one sample from the parametric variation set with models from Table 2. Prediction errors are shown in Appendix G.1

Table 3: Comparison of NRMSE for five models at $8\times$ SR ratio, with tricubic interpolation. Mean and standard deviation from three seeds are reported here. **Bold** term represents best mean.

| Models | Baseline Test Set | | | | Forced HIT Set | | | |
|---|---|---|---|---|---|---|---|---|
| | $\downarrow\text{NRMSE}_{\rho,\mathbf{u}}$ $(\times10^{-2})$ | $\downarrow\text{NRMSE}_{sgs}$ $(\times10^{-1})$ | $\downarrow\text{NRMSE}_{E^k}$ $(\times10^{-4})$ | $\downarrow\text{NRMSE}_{\epsilon}$ $(\times10^{-1})$ | $\downarrow\text{NRMSE}_{\rho,\mathbf{u}}$ $(\times10^{-3})$ | $\downarrow\text{NRMSE}_{sgs}$ $(\times10^{-2})$ | $\downarrow\text{NRMSE}_{E^k}$ $(\times10^{-6})$ | $\downarrow\text{NRMSE}_{\epsilon}$ $(\times10^{-4})$ |
| Tricubic 8$\times$ | 5.09 | 7.51 | 8.89 | 4.33 | 8.82 | 31.12 | 734.55 | 451.68 |
| RRDB 8$\times$ | 0.92±0.01 | 2.46±0.04 | 0.39±0.10 | 1.16±0.00 | 0.19±0.01 | 2.15±0.08 | 39.83±32.51 | 0.74±0.29 |
| (+ Grad. Loss) | **0.60±0.00** | **1.41±0.01** | 0.41±0.17 | **0.54±0.01** | 0.13±0.01 | **1.23±0.17** | 33.77±21.44 | 0.55±0.17 |
| EDSR 8$\times$ | 0.86±0.04 | 2.30±0.15 | **0.29±0.06** | 1.10±0.06 | 0.10±0.01 | 1.67±0.25 | **0.60±0.24** | **0.21±0.03** |
| RCAN 8$\times$ | 0.86±0.00 | 2.31±0.01 | 0.32±0.01 | 1.14±0.00 | **0.09±0.00** | 1.39±0.11 | 0.62±0.05 | 0.23±0.02 |
| ConvFNO 8$\times$ | 1.46±0.07 | 4.42±0.23 | 0.74±0.19 | 1.64±0.05 | 0.56±0.11 | 6.94±0.75 | 163.50±191.46 | 3.66±2.22 |

Scaling behavior of RRDB is shown in Figure 4, which compares ground truth and input values of $\rho e^k$ (shown in the first column) with $8\times$ SR predictions from tricubic interpolation and variations of RRDB models. For the model predictions, the first row visualizes the specific kinetic energy $\hat{\rho}\hat{e}^k$, while the second row shows the error $|\epsilon_{\rho e^k}| = |\hat{\rho}\hat{e}^k - \rho e^k|$ normalized by $\rho e^k_{max}$. Our discussion

is focused on the predictions in the cyan box. At $N_p = 0.6$M, RRDB is unable to reconstruct $\rho e^k$ accurately. RRDB's prediction is more accurate than tricubic interpolation at $N_p = 4.9$M, but spurious structures that originate from the coarse grid can be seen. For $N_p = 50.2$M, the model is sufficiently expressive for eliminating the spurious structures from the flow. The addition of the gradient loss term is shown to reduce prediction errors from RRDB 50.2M. This trend in improvement is also visible in the bottom row, which shows the mean divergence of SGS stresses (Equation (5)).

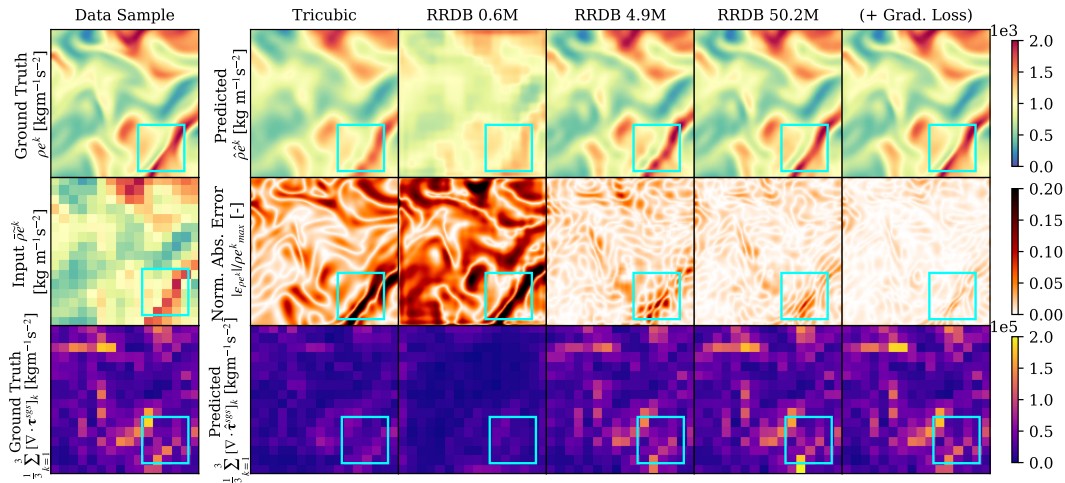

Figure 4: Predictions from various RRDB models, showing gradual improvement in the cyan box.

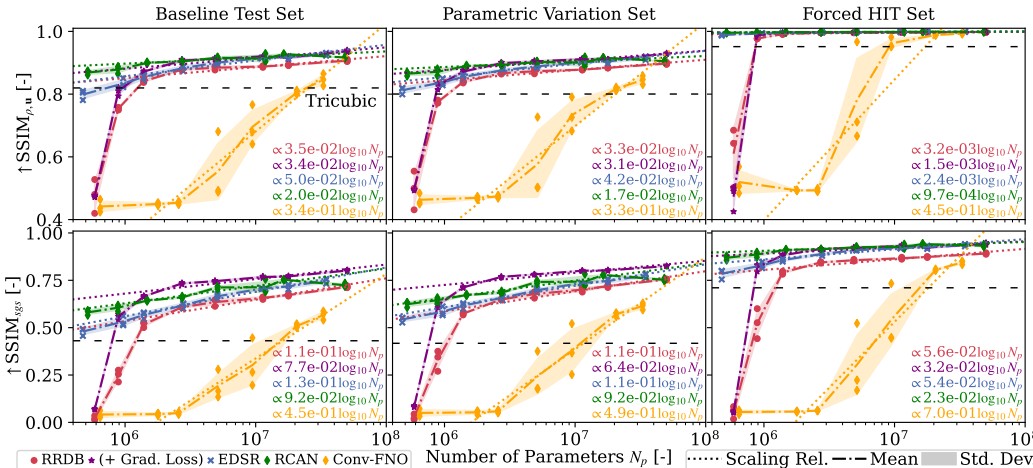

Figure 5: Scaling behavior of RRDB (with and without gradient-based loss), EDSR, RCAN and Conv-FNO. RRDB, EDSR and Conv-FNO models continue to scale at large model sizes.

Scaling behavior of RRDB (with and without gradient loss), EDSR, RCAN, and Conv-FNO models are examined in Figure 5. $\text{SSIM}_{sgs}$ scales differently compared to $\text{SSIM}_{\rho,\mathbf{u}}$, indicating the importance of evaluating derived physical quantities from model predictions in flow physics applications. For both SSIMs across all evaluation sets, RCAN models demonstrate better performance than EDSR and vanilla RRDB models for $N_p < 17$M, but performance deteriorates after this model size. The gradient loss term improves RRDB predictions for all model parameters explored, resulting in $\text{SSIM}_{sgs}$ that exceeds RCAN after $N_p = 1.4$M for the baseline test and Parametric Variation sets. Thus, this loss term is shown to benefit moderately sized models ($N_p = 50.2$M) and data (67 GB), which is in contrast to the notion that physics-based losses are mostly helpful for small models and datasets [50]. Conv-FNO is seen to outperform the baseline tricubic prediction after approximately 20M parameters. FNO layers are memory-intensive due to high number of dimensions found in the spectral convolution weights (six in total: one for batches, two for channels, and three for Fourier modes). This memory-intensive nature has been acknowledged by FNO's original developers, with attempts to address this remaining an active research pursuit [85].

For all models, both SSIMs are found to scale with $\log_{10} N_p$. All ResNet-based models share similar slopes in the scaling relationship between $\text{SSIM}_{sgs}$ and $\log_{10} N_p$ in the test and Parametric Variation set. However, these slopes can differ when evaluated on another flow configuration. This is seen with the idealized flows in the Forced HIT set, where higher SSIMs from all predictions and baseline are observed.

Figure 6 shows the relationship between $\text{SSIM}_{sgs}$ and inference cost (in FLOPs) for the five model approaches. $\text{SSIM}_{sgs}$ for EDSR, RCAN, and RRDB (with gradient loss) models scales with cost in a similar fashion, after approximately 100 GFLOPs. A steeper scaling relationship is observed for both Conv-FNO and vanilla RRDB. Vanilla RRDB models also do not demonstrate a strong linear relationship with $\log_{10}$ GFLOPs when tested on the Forced HIT set.

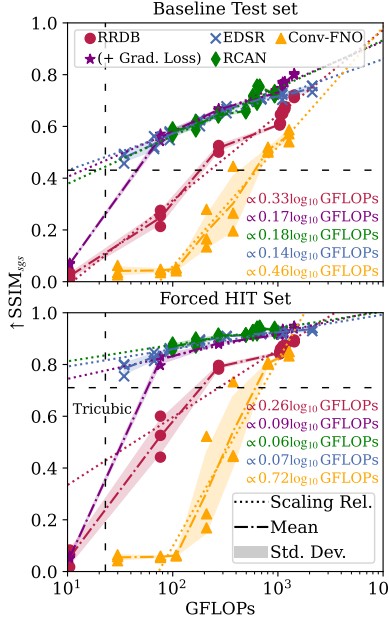

Figure 6: Scaling behavior with cost.

## 6    Conclusion

In this work, we released BLASTNet 2.0, a public 3D compressible turbulent reacting and non-reacting flow dataset. From this data, we extracted the Momentum128 3D SR dataset, which we employed for benchmarking 3D SR models at 8, 16, and $32\times$ SR. SR models are shown to score well in SSIM-based metrics and capture fine turbulent structures at $8\times$ SR. For the higher SR ratios, these fine structures cannot be captured, but the SR models can still recover the magnitude of large flow structures. Through our scaling analysis, we demonstrate that benefits from a gradient-based physics-based loss persist with model scale – providing empirical evidence that disagrees with the postulated notion that physics-based methods are useful mostly in small model scenarios [50]. However, we recognize that this observation is applicable only to one type of physics-based ML technique, and is not necessarily extendable to other physics-based ML approaches. We observe that model performance scales with the logarithm of model parameters, and that the scaling relationship between $\text{SSIM}_{sgs}$ and inference cost are similar for RRDB (with gradient loss), EDSR, and RCAN. We also demonstrate that the choice of model architecture can matter significantly, especially when developing small models for real-time scientific computing applications, and that physics-based losses can improve some metrics of poorly performing architectures. With this work, we demonstrate that BLASTNet 2.0 can provide a rich resource for evaluating models for scientific and engineering turbulent flows.

**Limitations and Negative Impact**   The simple geometry, skeletal finite-rate mechanisms and mixture-averaged transport used in these DNS provide high-fidelity information of fundamental processes, but are not fully representative of real-world systems. However, rectifying this would require complex geometry, detailed mechanisms (introducing an order of magnitude more PDEs) and multi-component transport that can result in intractable calculations [86]. Another limitation is that the DNS data originate from proprietary-licensed numerical solvers (see Appendix D.2), resulting in data that cannot be thoroughly inspected through open-source means. However, expertise, peer review from published research, and solver reputation ensure that these DNS data meet community-accepted standards. In this work, we are limited to using Favre-filtered (Equation (3)) DNS to generate low-resolution inputs in the Momentum128 3D SR dataset, as it is not feasible to obtain pairs of coarse simulation sample that matches a corresponding ground truth DNS sample, as the chaotic nature of turbulence will result in uncorrelated pairs [72]. However, Favre-filtered DNS is a canonical surrogate [74] for coarse simulations with strong theoretical foundations [73, 45]. Further discussion on the validity of employing Favre-filtered DNS is provided in Appendix E.1.6. Data generation incurs up to $\mathcal{O}(10^6)$ CPU-hours per case, while this study used 15,000 GPU-hours – resulting in significant carbon emissions. However, we attempted to ameliorate this by curating previously unreleased already-generated DNS from existing publications, and employing mixed-precision training for this study. In addition, this work can improve fundamental knowledge on carbon-free combustion, which can reduce society-wide reliance on hydrocarbons.

## Acknowledgments and Disclosure of Funding

The authors acknowledge financial support and computing resources from the U.S. Department of Energy, National Nuclear Security Administration under award No. DE-NA0003968. We also thank the NASA Early Stage Innovation Program with award No. 80NSSC22K0257, the Department of Energy, Office of Science under award No. DE-SC0022222, and Office of Energy Efficiency Renewable Energy (EERE) with award No. DE-EE0008875 for funding this work. Wai Tong Chung is grateful for partial financial support from the Stanford Institute for Human-centered Artificial Intelligence Graduate Fellowship. This research used resources of the National Energy Research Scientific Computing Center (NERSC), a U.S. Department of Energy Office of Science User Facility located at Lawrence Berkeley National Laboratory. We are also grateful to members of the Kaggle staff for fruitful discussions that have contributed to this work.

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

# A  URL and Links

**BLASTNet 2.0 Dataset**   The landing page for the Bearable Large Accessible Scientific Training Network-of-Datasets (BLASTNet) is hosted at `https://blastnet.github.io`. Specifically, links for browsing and downloading the dataset is consolidated at `https://blastnet.github.io/datasets`.

**Momentum128 3D SR Dataset**  This 3D super-resolution dataset for turbulent flows is available at `https://www.kaggle.com/datasets/waitongchung/blastnet-momentum-3d-sr-dataset`.

**Code**   Code for training and evaluating models in this study are available in `https://github.com/blastnet/blastnet2_sr_benchmark`, with instructions provided in the `README` of the repository.

**Model Weights**   All weights of models in this study are also available at `https://www.kaggle.com/datasets/waitongchung/blastnet-momentum-3d-sr-dataset`.

**DOI**   All BLASTNet datasets [87] share a common DOI at Zenodo: `https://doi.org/10.5281/zenodo.7242864`.

**Tutorials and Browsing Tools**   Kaggle notebooks for browsing BLASTNet simulation data are attached to each of the Kaggle URLs of specific direct numerical simulation (DNS) data, as listed in Appendix D.2. These notebooks are also consolidated at `https://github.com/blastnet/kaggle_tutorials`. In addition, a table that summarizes the datasets is also provided in `https://blastnet.github.io/datasets`. Additional tutorials on reading the data, contributing to BLASTNet, and using the Kaggle command-line API are also provided in `https://blastnet.github.io/tutorial`.

# B  Licensing and Author Statement of Responsibility

All data is generated by the present authors and licensed via CC BY-NC-SA 4.0. The present authors bear responsibility in case of violation of rights.

# C  Maintenance Plan and Long Term Preservation

The contributors to BLASTNet 2.0 are committed to maintaining and preserving this dataset. Maintenance of this dataset will largely involve tracking and fixing issues that might be discovered after release. To facilitate this, we host an issues webpage (`https://github.com/blastnet/blastnet.github.io/issues`) for user feedback. All data is shared via Kaggle, ensuring that the data will be preserved and available in the long-term. In addition, our maintenance plan involves adhering to the FAIR principles [71] for scientific data management, with the specific details as follows:

**Findable**   All data are indexed and can be easily searched via both Kaggle and BLASTNet platforms. To ensure that the data is findable, a `http://schema.org` structured metadata is employed, as detailed in Appendices D and E. All BLASTNet datasets share a global and persistent DOI at Zenodo: `https://doi.org/10.5281/zenodo.7242864`.

**Accessible**   Both data and descriptive metadata are retrievable via the Kaggle command-line API. This protocol is free and available at `https://github.com/Kaggle/kaggle-api`, with authentication and authorization provided through a Kaggle account. We provide a `bash` script for users to download all data (shared in multiple repositories) at once with this API. Users can also download the data directly from Kaggle repositories.

**Interoperable**   The data and descriptive metadata use accessible formats that can be read by standard `python numpy` and `json` packages. BLASTNet's `http://schema.org` structured metadata also references the structured metadata of each separate BLASTNet repository (providing information on specific contributors and Kaggle URLs). We have attempted to use accessible language when generating these metadata.

**Reusable**   The descriptive metadata contains information on the flow configuration (initial conditions, chemistry, numerics, source publication, *etc.*). In addition, all Kaggle repositories employ a CC BY-SA NC 4.0 license. The structured `http://schema.org` metadata provides rich information that passes the rich results test (`https://search.google.com/test/rich-results`). All data and descriptive metadata are presented in consistent little-endian single-precision binaries and `.json` files, guaranteeing acceptable standards for fast I/O, sufficient floating-point precision, and broad accessibility via widely-used `python` packages.

## D   Additional BLASTNet 2.0 Details

BLASTNet 2.0 contains pre-processed DNS data shared via a network of Kaggle repositories, with links consolidated at the landing page `https://blastnet.github.io`.

### D.1   Data Format and Directory Structure

Data, generated from different numerical solvers (as detailed in Appendix D.2), initially exists in a range of formats (`.vtk`, `.vtu`, `.tec`, and `.dat`) that are not readily formatted for training ML models. Thus, we pre-process all generated data into a consistent and convenient format consisting of physical and chemical data (Appendix D.1.1), descriptive metadata (Appendix D.1.2), and web metadata (Appendix D.1.3), along with instructions for reading the data in Appendix D.1.4. This information is summarized in Figure 7.

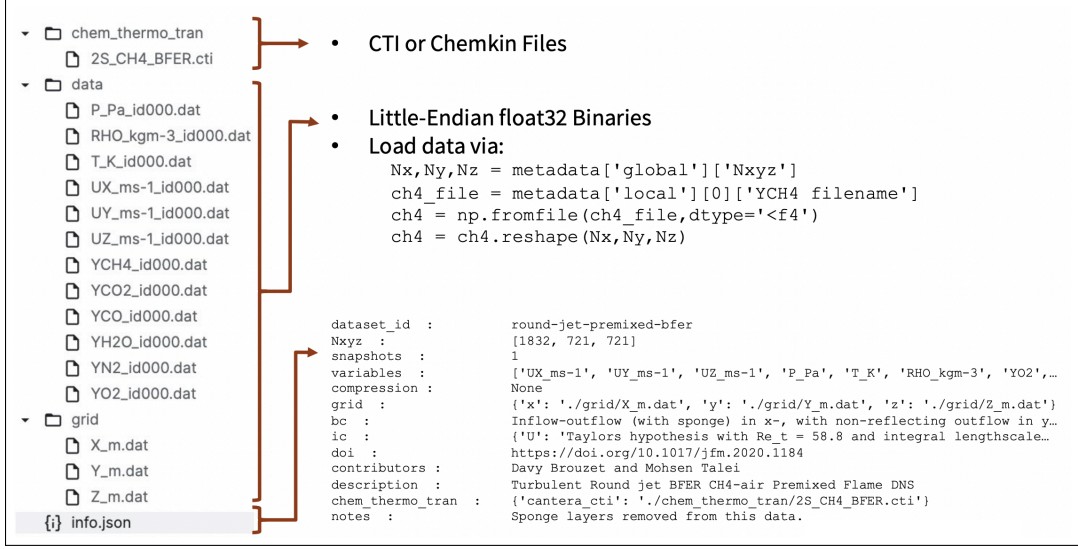

Figure 7: Directory structure and reading instructions for an instance of a BLASTNet configuration.

### D.1.1   Files on Flow Physics and Chemistry

All flowfield data are processed into a consistent format – little-endian single-precision binaries that can be read with `np.fromfile`/`np.memmap`, as shown in Figure 7. The choice of this data format enables high I/O speed in loading arrays. As also shown in Figure 7, we also provide `.json` files (see Appendix D.1.2) that store additional information on configurations, contributors, solvers, and corresponding source publications. Chemical mechanisms and transport properties are shared through Cantera [88] `.cti`/`.xml`/`.yaml` or Chemkin [89] `fortran` files. Thus BLASTNet data contains

all information needed to reconstruct any derived auxiliary quantities (such as vorticity, viscosity, turbulence closure terms, along with heat and chemical transport coefficients) from the conservation equations.

### D.1.2 Descriptive Metadata

The binary data described in Appendix D.1.1 contains information on physical and chemical data, without much context. Details involving global information such as configuration, boundary/initial conditions, solvers, related publications, and spatial grid information, as well as local temporal information (if any) are provided through an `info.json` file in each Kaggle repository. Listings 1 and 2 present the `python` code used to generate global and local information in one example of `info.json`.

```
metadata['global'] = {
        "dataset_id": "waitongchung/inert-ch4o2-hit-dns",
             "Nxyz": [129,129,129],
        "snapshots": 98,
        "variables": ["UX_ms-1","UY_ms-1","UZ_ms-1",
                      "P_Pa","T_K","RHO_kgm-3",
                      "YO2","YCH4"],
      "compression": "None",
             "grid": {"x": "./grid/X_m.dat",
                      "y": "./grid/Y_m.dat",
                      "z": "./grid/Z_m.dat"},
         "numerics": {"spatial": "4th order central-differencing
                                  with 2nd order ENO",
                      "temporal": "3rd-order SSP-RK3 (non-stiff)
                                   and semi-implicit ROWPLUS (stiff)",
                      "solver": "CharlesX"},
               "bc": "Periodic in x-, y-, and z-directions.",
               "ic": {"U": "HIT Von Karman Pao with Re_t = 80 and
                           integral lengthscale of 62.5E-6m",
                 "T [K]": 300,
                 "P [Pa]": 101325,
                 "Mixture": "CH4-O2 inert branch from 1D
                             cantera counterflow calculations."},
              "doi": "https://doi.org/10.1016/j.combustflame.2021.111758",
     "contributors": "Wai Tong Chung and Matthias Ihme",
      "description": "Compressible Inert CH4-O2 Homogeneous
                      Isotropic Turbulence DNS",
  "chem_thermo_tran": {"description": "FRC and Mixture-Averaged Transport
                                       with constant lewis number",
                       "cantera_xml": "./chem_thermo_tran/bfer.xml"}
}
```

Listing 1: Python command for generating global metadata for a BLASTNet Kaggle repository.

```
metadata['local'] = [
                {"id": 0,
            "time [s]": 6.88389e-06,
     "UX_ms-1 filename": "./data/UX_ms-1_id000.dat",
     "UY_ms-1 filename": "./data/UY_ms-1_id000.dat",
     "UZ_ms-1 filename": "./data/UZ_ms-1_id000.dat",
     "  P_Pa filename": "./data/P_Pa_id000.dat",
        "T_K filename": "./data/T_K_id000.dat",
   "RHO_kgm-3 filename": "./data/RHO_kgm-3_id000.dat",
        "YO2 filename": "./data/YO2_id000.dat",
       "YCH4 filename": "./data/YCH4_id000.dat"},
                {"id": 1, ...},
                    ...,
                {"id": 97, ...}
]
```

Listing 2: Python command for generating local metadata for a BLASTNet Kaggle repository.

### D.1.3 Structured Web Metadata

A `http://schema.org` metadata has been added to `https://blastnet.github.io/datasets`, and tested with `https://search.google.com/test/rich-results`.

### D.1.4 Reading Data

As shown in Figure 7, BLASTNet data can be read by (i) loading the descriptive metadata with the `json` package on `python`, and (i) using `np.fromfile`/`np.memmap` to load and reshape the data. Links to tutorials to perform this are also shared in Appendix A.

### D.2 DNS Configurations

BLASTNet 2.0 contains data from 34 different DNS configurations. The Kaggle link of all configurations are provided in Table 4. The details of each DNS configuration are provided in this section as well.

Table 4: The Kaggle link of all 34 DNS configurations. $Ka_u$, $U_{in}$ and $S_L$ denote the Karlovitz number, inlet bulk velocity and laminar burning velocity, respectively.

| Index | Sec. | Short Description | Kaggle URL |
|---|---|---|---|
| 1 | D.2.1 | Inert HIT [20] | www.kaggle.com/datasets/waitongchung/inert-ch4o2-hit-dns |
| 2 | D.2.2 | Reacting forced HIT [67] | www.kaggle.com/datasets/waitongchung/forced-hit-ch4-air-ffcm |
| 3 | D.2.3 | Reacting jet flow [29] | www.kaggle.com/datasets/waitongchung/round-jet-premixed-bfer |
| 4 | | Reacting jet flow [29] | www.kaggle.com/datasets/waitongchung/round-jet-premixed-coffee |
| 5-10 | D.2.4 | Inert transcrit. chan. flow [69] | www.kaggle.com/datasets/jguo96/transcritical-n2-channel-dns |
| 11 | D.2.5 | Reacting channel flow [28] | www.kaggle.com/datasets/waitongchung/premixed-flame-wall-ch4-air-dns-gri |
| 12 | D.2.6 | Slot burner [27] | www.kaggle.com/datasets/waitongchung/full-lifted-flame-dns-li |
| 13 | | Freely-propagating flame [70] ($Ka_u = 2.4, U_{in}/S_L = 2.45$) | www.kaggle.com/datasets/waitongchung/free-propagating-h2-vit-air-li-case-2 |
| 14 | | ($Ka_u = 6.8, U_{in}/S_L = 2.45$) | www.kaggle.com/datasets/waitongchung/free-propagating-h2-vit-air-li-case-3 |
| 15 | | ($Ka_u = 13, U_{in}/S_L = 2.45$) | www.kaggle.com/datasets/waitongchung/free-propagating-h2-vit-air-li-case-4 |
| 16 | | ($Ka_u = 2.4, U_{in}/S_L = 3.67$) | www.kaggle.com/datasets/waitongchung/free-propagating-h2-vit-air-li-case-5 |
| 17 | | ($Ka_u = 6.8, U_{in}/S_L = 3.67$) | www.kaggle.com/datasets/waitongchung/free-propagating-h2-vit-air-li-case-6 |
| 18 | | ($Ka_u = 13, U_{in}/S_L = 3.67$) | www.kaggle.com/datasets/waitongchung/free-propagating-h2-vit-air-li-case-7 |
| 19 | | ($Ka_u = 19, U_{in}/S_L = 3.67$) | www.kaggle.com/datasets/waitongchung/free-propagating-h2-vit-air-li-case-8 |
| 20 | | ($Ka_u = 36, U_{in}/S_L = 3.67$) | www.kaggle.com/datasets/waitongchung/free-propagating-h2-vit-air-li-case-9 |
| 21 | | ($Ka_u = 2.4, U_{in}/S_L = 4.63$) | www.kaggle.com/datasets/waitongchung/free-propagating-h2-vit-air-li-case-11 |
| 22 | D.2.7 | ($Ka_u = 6.8, U_{in}/S_L = 4.63$) | www.kaggle.com/datasets/waitongchung/free-propagating-h2-vit-air-li-case-12 |
| 23 | | ($Ka_u = 13, U_{in}/S_L = 4.63$) | www.kaggle.com/datasets/waitongchung/free-propagating-h2-vit-air-li-case-13 |
| 24 | | ($Ka_u = 2.4, U_{in}/S_L = 5.51$) | www.kaggle.com/datasets/waitongchung/free-propagating-h2-vit-air-li-case-17 |
| 25 | | ($Ka_u = 6.8, U_{in}/S_L = 5.51$) | www.kaggle.com/datasets/waitongchung/free-propagating-h2-vit-air-li-case-18 |
| 26 | | ($Ka_u = 19, U_{in}/S_L = 5.51$) | www.kaggle.com/datasets/waitongchung/free-propagating-h2-vit-air-li-case-19 |
| 27 | | ($Ka_u = 1.7, U_{in}/S_L = 3.67$) | www.kaggle.com/datasets/waitongchung/free-propagating-h2-vit-air-li-case-22 |
| 28 | | ($Ka_u = 4.8, U_{in}/S_L = 3.67$) | www.kaggle.com/datasets/waitongchung/free-propagating-h2-vit-air-li-case-23 |
| 29 | | ($Ka_u = 8.9, U_{in}/S_L = 3.67$) | www.kaggle.com/datasets/waitongchung/free-propagating-h2-vit-air-li-case-24 |
| 30 | | ($Ka_u = 1.7, U_{in}/S_L = 4.63$) | www.kaggle.com/datasets/waitongchung/free-propagating-h2-vit-air-li-case-26 |
| 31 | | ($Ka_u = 4.8, U_{in}/S_L = 4.63$) | www.kaggle.com/datasets/waitongchung/free-propagating-h2-vit-air-li-case-27 |
| 32 | | ($Ka_u = 8.9, U_{in}/S_L = 4.63$) | www.kaggle.com/datasets/waitongchung/free-propagating-h2-vit-air-li-case-28 |
| 33 | | ($Ka_u = 1.7, U_{in}/S_L = 5.51$) | www.kaggle.com/datasets/waitongchung/free-propagating-h2-vit-air-li-case-30 |
| 34 | | ($Ka_u = 8.9, U_{in}/S_L = 5.51$) | www.kaggle.com/datasets/waitongchung/free-propagating-h2-vit-air-li-case-32 |

### D.2.1 Non-reacting homogeneous isotropic turbulence (HIT)

The HIT DNS simulation [20] is performed on a 3D cubic domain of length $L$, where a spherical gaseous-oxygen core of radius $r = 0.25L$ at 300 K is initialized in gaseous methane environment of 300 K at 1 atm pressure, providing an idealized representation of an inert gaseous fuel-air mixture in a rocket engine. The simulation setup is shown in Figure 8. Periodic boundary conditions are used at all boundaries. A synthetic turbulence generator by Saad et al. [90] based on von Kármán-Pao energy spectrum with zero mean velocity is used to generate the initial velocity profile. Ideal gas law is used as the equation of state (EoS) to relate pressure, temperature and density.

The simulation is performed in an unstructured compressible finite-volume solver [64]. The solver uses a fourth-order accurate central spatial finite difference scheme. For the time integration, a stable third-order Runge-Kutta scheme is employed. As mentioned before in Section 3, mixture-averaged transport properties are used in the DNS.

### D.2.2 Reacting forced homogeneous isotropic turbulence

The DNS study [67] involves a statistically steady, isotropic, and homogeneous turbulent flow in an unconfined space. The schematic of the simulation setup is presented in Figure 9. The flame is initialized by a planar surface separating half of the domain containing methane/air mixture at

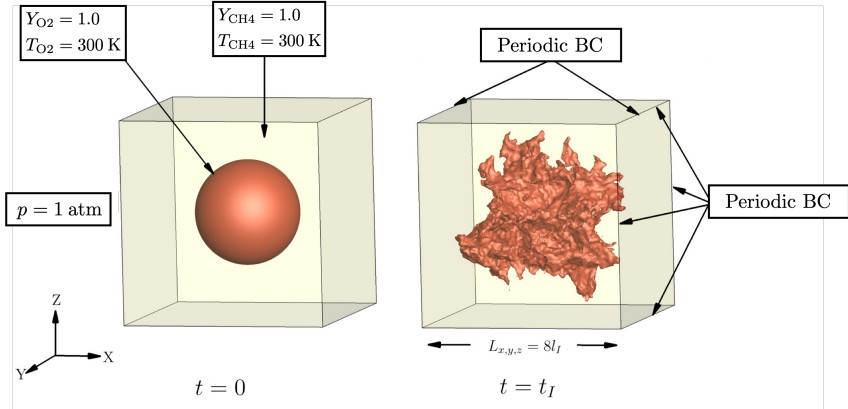

Figure 8: Schematic of the DNS configuration of inert HIT. $t_I$ denotes large-eddy timescale, while $l_I$ denotes integral lengthscale. Adapted from [20], Copyright 2022, with permission from Elsevier.

700 K and $3.04 \times 10^7 \, \mathrm{erg \, cm^{-3}}$ pressure, and another half with hot products, and is immersed in a high-intensity turbulent flow field with Kolmogorov type spectrum. The idea is to investigate the process of flame interaction with steady homogeneous isotropic turbulence. However, the flow needs to be constantly stirred at the largest scale to ensure a steady energy cascade to smaller scales so that the turbulence-flame interaction at the quasi-steady state can be studied. A spectral turbulence-driving method is used in the study, the details of which are available in Poludnenko and Oran [67]. This driving method produces statistically steady forced-HIT flows with arbitrarily complex energy spectra. In particular, it is possible to achieve Kolmogorov type turbulence with inertial range of energy cascade extending up to energy injection scale. The other advantage of this method is that it does not introduce any artificial large-scale anisotropy, compression, or rarefaction. Prior to ignition, all domain boundaries are periodic. At ignition, boundary conditions along the left and right $z$-boundaries (as shown in Figure 9) are switched to zero-order extrapolation to prevent any non-physical pressure build-up in the domain and the formation of artificial large-scale rarefaction waves at the boundaries.

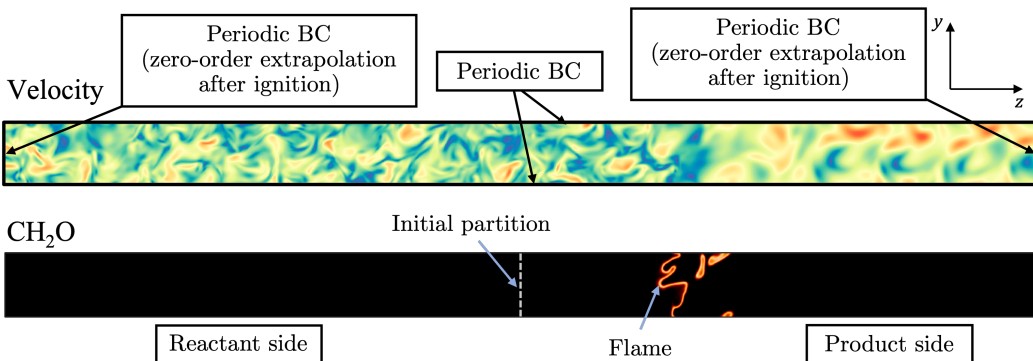

Figure 9: Schematic of the DNS configuration of reacting forced HIT [67]. The top and bottom contour plots correspond to the magnitude of velocity and $CH_2O$ species mass fractions, respectively.

The computational domain aspect ratio is $1 \times 1 \times 16$, with a grid size of $257 \times 257 \times 4097$, including 16 grid points per unit laminar flame thermal thickness. The cell size is $2.62 \times 10^{-4} \, \mathrm{cm}$. The turbulent velocity at energy injection scale ($L = 0.067 \, \mathrm{cm}$) length scale is $213.92 \, \mathrm{cms^{-1}}$ with turbulent root-mean-squared (RMS) velocity of $245.83 \, \mathrm{cms^{-1}}$, resulting in an eddy turnover time of $3.14 \times 10^{-4} \, \mathrm{s}$. The same velocity quantities corresponding to the integral length scale ($l = 0.0196 \, \mathrm{cm}$) are $141.93 \, \mathrm{cms^{-1}}$ and $132.2 \, \mathrm{cms^{-1}}$. The ignition delay time of the mixture is three times the eddy turn-over time, and the total simulation runtime is 16 times the eddy turn-over time. The Damköhler and Karlovitz numbers are 0.66 and 9.97, respectively.

The DNS calculation is performed using the code Athena-RFX [67], which implements higher-order fully conservative Godunov-type methods for integration of fluid equations. The numerics in this work are third-order accurate in space and second-order accurate in time. More details are available in the original paper [67]. The foundational fuel chemistry model (FFCM-1) [91] with 22 species and 107 reactions is used as the chemical mechanism.

### D.2.3  Reacting jet flows

The DNS configurations by Brouzet et al. [29] involve two parametric variations of 3D reacting turbulent premixed methane/air round-jet flames with high-fidelity acoustics to investigate the effect of different chemical mechanisms on flame dynamics. The setup is initialized with methane/air combustion products at adiabatic flame temperature and at atmospheric pressure. The jet Reynolds and Mach numbers are 5300 and 0.36, respectively. A schematic representation of the DNS configuration is shown in Figure 10. The two variations of the reacting jet correspond to two different chemical mechanisms: (i) a semi-global $CH_4$-BFER mechanism with 2 reactions [92] , and (ii) a skeletal COFFEE mechanism [93] with 14 species and 38 reactions. In both configurations, the domain size is $20D \times 16D \times 16D$. The grid sizes are $1811 \times 721 \times 721$ and $1546 \times 676 \times 676$ for the BFER and COFFEE cases, respectively. These meshes correspond to 10 and 12 grid points per unit thermal flame thickness in the streamwise direction, and 12 and 16 points in the transverse and spanwise directions.

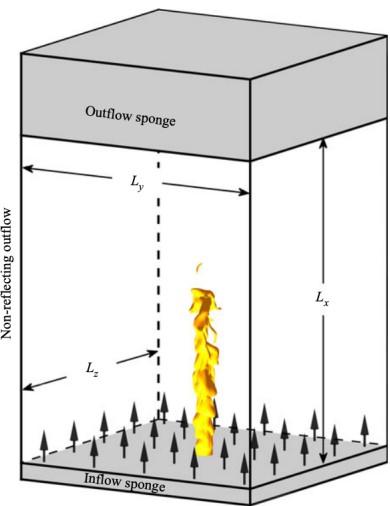

Figure 10: Schematic of the DNS configuration of reacting jet flows. Reprinted from [29], Copyright 2021, with permission from Cambridge University Press.

The DNS is performed using the code NTMIX-CHEMKIN [94], which solves fully compressible Navier-Stokes equations along with energy and species conservation equations in Cartesian coordinates. The solver uses an eight-order explicit central spatial difference scheme and a third-order Runge-Kutta time integration scheme. Ideal gas law and mixture-averaged species-specific properties are used for the simulations. Further details of the DNS configuration and solver are provided in Brouzet et al. [29].

### D.2.4  Non-reacting transcritical channel flow

The study by Guo et al. [69] involves six different configurations of wall-bounded DNS in the transcritical regime. The schematic of the DNS setup is shown in Figure 11. They used nitrogen $N_2$ as the working fluid with a critical pressure and temperature of $p_c = 3.39$ MPa and $T_c = 126.19$ K. These studies consider the flow of $N_2$ inside a channel with a hot top and a cold bottom wall with temperatures $T_{hot}$ and $T_{cold}$, respectively. The six variations correspond to different temperature ratio (TR) between the two walls. The channel is periodic in streamwise and spanwise direction, while the wall boundary conditions are enforces at two walls. The domain dimensions are $L_x \times 2L_y \times L_z$, where $L_x/L_y = 2\pi$, $L_z/L_y = 4\pi/3$ and the channel height is $2L_y = 9.0132 \times 10^{-5}$ m. A Cartesian grid (with mesh size $384 \times 256 \times 384$) is used for all six configurations.

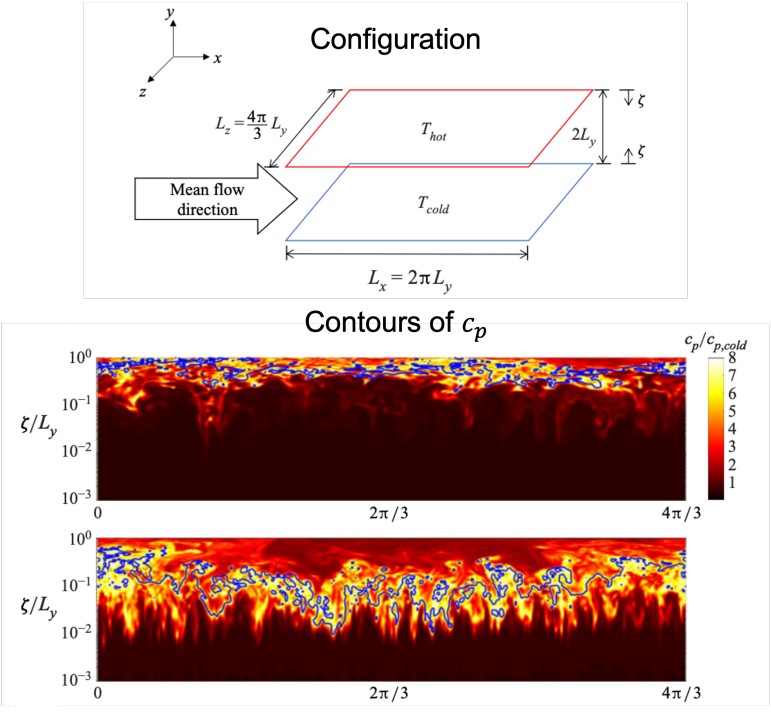

Figure 11: Schematic of the DNS configuration of non-reacting transcritical channel flows. The contour plots correspond to the $c_p$ for configurations TR3 and TR1.9. Adapted from [69], Copyright 2022, with permission from Cambridge University Press.

A summary of the individual configurations is provided here in Table 5, with more details being available in the original paper [69]. The operating conditions are chosen such that they cover the range of density ratio, $\Omega = \rho_{\text{hot}}/\rho_{\text{cold}}$, between 1 and 20, where $\rho$ is the density of the fluid at the wall. The name of the configurations are based on different TRs. Configurations TR3, TR1.9, TR1.4 and TR1.3 are transcritical, whereas the other two configurations are sub-critical. The pressure for all configurations is set to be $3.87\,\mathrm{MPa}$, which is higher than $p_c$. The bulk Reynolds number is $3.5 \times 10^4$. Table 5 also reports the friction Reynolds number ($Re_\tau$) for two walls using the channel half-height $L_y$ as the length scale.

A compressible finite-volume solver [64] is used for these DNS. The governing equations are solved using a strong stability-preserving Runge-Kutta scheme with third-order accuracy in time stepping, and a fourth-order accurate central spatial finite difference, which reduces to third-order for non-uniform meshes. As the conditions of these simulations are in the transcritical regime, the Peng-Robinson EoS is used, which provides better accuracy in predicting thermodynamic variables than ideal gas in the investigated regime. To avoid the pressure oscillations and to obtain physically realizable solutions, an entropy-stable double-flux model [64] is used along with second-order accurate essentially non-oscillatory (ENO) scheme and Harten-Lax-Van Leer contact (HLLC) Riemann flux computations.

### D.2.5 Reacting channel flow

This DNS configuration by Jiang et al. [28] investigates the flame-wall interaction for methane/air flames diluted by hot combustion products in a 3D turbulent V-flame configuration inside a channel with isothermal hot and cold walls. The simulation setup is shown in Figure 12. At the inlet of the channel, the reactant mixture consists of a mixture of cold reactants (30%) and hot combustion products from 1D premixed freely-propagating flame simulation (70%), resulting in an inlet temperature of $T_{\text{in}} = 1705\,\mathrm{K}$ at $2\,\mathrm{atm}$ pressure. The hot and cold wall temperatures are fixed at 1200 and 400 K, respectively. The inlet turbulence is generated with a non-reacting simulation of the same channel. Then, the results collected at a sampling plane of $x/H = 4$ are fed into the reacting simulation. This turbulence generation allows coupling of the velocity and temperature fluctuations at the inlet.

Table 5: Summary of the operating conditions of six different DNS configurations of non-reacting transcritical channel flows [69]. $T_{r,\text{cold}} = T_{\text{cold}}/T_c$, $T_{r,\text{hot}} = T_{\text{hot}}/T_c$ and $\rho_{r,0} = \rho_0/\rho_c$, where subscript 0 and $c$ indicate volume averaged and critical quantity, respectively.

| Configs. | $T_{r,\text{cold}}$ | $T_{r,\text{hot}}$ | $\rho_{r,0}$ | $\Omega$ | $\text{Re}_{\tau,\text{cold}}$ | $\text{Re}_{\tau,\text{hot}}$ |
|----------|------|------|------|-------|-----|------|
| TR3 | 0.79 | 2.38 | 1.16 | 17.84 | 430 | 300 |
| TR1.9 | 0.79 | 1.51 | 1.60 | 10.05 | 440 | 610 |
| TR1.4 | 0.79 | 1.11 | 1.92 | 5.24 | 500 | 1370 |
| TR1.3 | 0.79 | 1.03 | 2.09 | 2.89 | 570 | 1530 |
| TR1.25 | 0.79 | 0.99 | 2.19 | 1.60 | 590 | 1290 |
| TR1 | 0.79 | 0.79 | 2.36 | 1.00 | 700 | 700 |

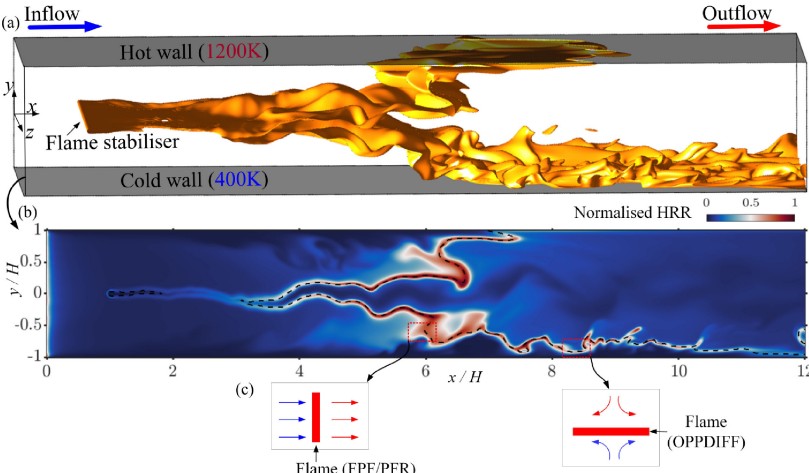

Figure 12: Schematic of the 3D DNS domain of reacting channel flow. The bottom figure shows a snapshot of 2D normalized heat release rate contour along with the black dashed line, which is an iso-line of methane progress variable at a particular value. Reprinted from [28], Copyright 2021, with permission from Elsevier.

Velocity fluctuations are first produced using the Passot-Pouquet spectrum for the turbulent kinetic energy. The inlet turbulence for the non-reacting simulation was then generated by rescaling these fluctuations with the RMS profiles of a fully developed channel flow at a Reynolds number of 3200. Next, this is fed into the domain with a convection velocity 25% lower than the mean inlet velocity at the centerline. This accounts for a correction to the Taylor's hypothesis due to the high near-wall shear stress. Non-reflecting Navier-Stokes Characteristic Boundary Condition (NSCBC) is used for the outlet boundary, and a periodic boundary condition is used in the $z$-direction. For the reacting case to ignite, a cylindrical hot patch is imposed at $y/H = 0$ and $x/H = 1$ with a diameter of $0.03H$, which creates two branches of the V-flame that interact with two walls.

The domain size is $12H \times 2H \times 3H$, with a grid size of $1000 \times 250 \times 250$, which stretches from $5\,\mu\text{m}$ at the wall to $30\,\mu\text{m}$ at the centerline in the $y$-direction, and $30\,\mu\text{m}$ uniform grid in both $x$- and $z$-direction, and ensures at least one grid point within one wall unit and a mean grid size less than 1.4 times the Kolmogorov length scale. There are around 20 grid points inside the flame thickness as well.

The numerical solver used for the DNS study is NTMIX-CHEMKIN [94]. Similar to the study of Brouzet et al. [29], this solver features an eighth-order central finite difference scheme for spatial derivatives and a third-order Runge-Kutta time integrator. A tenth-order explicit filter is also used to eliminate spurious oscillations at high wave numbers. Ideal gas law is used as the EoS. A reduced mechanism for methane/air combustion with 23 species, 12 quasi-steady species and 205 reactions is developed for this study.

### D.2.6 Partially-premixed slot burner

This DNS configuration [27] involves a turbulent lifted hydrogen jet flame in heated co-flow air. Figure 13 shows the schematic of the simulation setup. A diluted fuel mixture (65% $H_2$ and 35% $N_2$ by volume) is issued from the central slot at an inlet temperature of $400\,\mathrm{K}$. This central jet is surrounded on either side by co-flowing heated air streams with an inlet temperature of $850\,\mathrm{K}$, at atmospheric pressure. The mean inlet axial velocity $U_{\mathrm{in}}$ is given by:

$$U_{\mathrm{in}} = U_c + \frac{U_{\mathrm{jet}} - U_c}{2}\left[\tanh\left(\frac{y + H/2}{0.1H}\right) - \tanh\left(\frac{y - H/2}{0.1H}\right)\right], \tag{7}$$

where the mean inlet jet ($U_{\mathrm{jet}}$) and co-flow ($U_c$) velocities are 240 and $2\,\mathrm{ms}^{-1}$, respectively. The jet width at the inlet is $2\,\mathrm{mm}$. The other quantities, such as temperature and species mass fractions also follow the same profile (Equation (7)). The jet Reynolds number is 8000. Velocity fluctuations, $u'$, which is 10% of $U_{\mathrm{jet}}$, is obtained by generating an auxiliary homogeneous isotropic turbulence field. These fluctuations are then fed from the inlet using Taylor's hypothesis. This $2000 \times 1600 \times 400$ computational domain is $15H \times 20H \times 3H$ in the streamwise $x$-, transverse $y$-, and spanwise $z$-directions, respectively, resulting in a total of 1.28 billion cells. A uniform grid size of $15\,\mu\mathrm{m}$ is placed in the $x$- and $z$-directions, while the $y$-directional grid is algebraically stretched outside the flame and shear zones. Improved non-reflecting boundary conditions [95, 96] are adopted in the $x$- and $y$-directions, while periodic boundary conditions are applied in the $z$-direction. The data is collected after four jet flow-through times after the flame becomes statistically stationary.

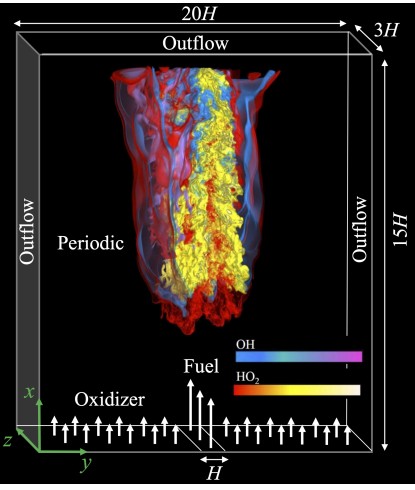

Figure 13: Schematic of the DNS of slot-burner setup. The contours correspond to OH and $HO_2$ species. Reprinted from [27], Copyright 2021, with permission from Elsevier.

The Sandia DNS code, S3D [65], is employed for solving the compressible Navier–Stokes, species conservation, and total energy equations. Spatial derivatives are approximated with an eighth-order central difference scheme, and a tenth-order filter is used to remove any spurious high-frequency fluctuations in the solution. For time integration, a fourth-order explicit Runge-Kutta method is used. The employed detailed hydrogen-air chemical mechanism composed of 9 species and 21 elementary reaction steps was developed by Li et al. [97]

### D.2.7 Freely-propagating flame

This DNS configuration [70] presents a statistically-planar, freely-propagating flame. BLASTNet contains 22 parametric variations of this configuration that differ by three essential parameters involving turbulence: (i) turbulence intensity, characterized by the RMS velocity $u'$, (ii) inflow velocity, $U_{\mathrm{in}}$, and (iii) integral length scale, $l_I$. A schematic diagram of the setup is shown in Figure 14. These configurations represent a series of hydrogen-premixed turbulent flames in autoignitive reheat combustion conditions that provide rich information on regimes of turbulent spontaneous ignition and turbulent deflagration.

Table 6: Summary of the simulation parameters for all DNS runs [70].

| Config. Index | $U_{\text{in}}/S_L$ | $u'/S_L$ | $\text{Ka}_u$ | $\text{Re}_{t,u}$ | $\text{Da}_{\text{ign}}$ | $t_{\text{end}}/\tau_{\text{ign},0}$ | $L_x/L_y$ | $N_x \times N_y \times N_z$ |
|---|---|---|---|---|---|---|---|---|
| $l_I/l_f = 1.5$ | | | | | | | | |
| 1 | 2.45 | 0.5 | 2.4 | 53 | 0.24 | 13 | 9 | $1152 \times 128 \times 128$ |
| 2 | 2.45 | 1.0 | 6.8 | 105 | 0.12 | 4.6 | 9 | $1152 \times 128 \times 128$ |
| 3 | 2.45 | 1.5 | 13.0 | 158 | 0.08 | 3.6 | 9 | $1152 \times 128 \times 128$ |
| 4 | 3.67 | 0.5 | 2.4 | 53 | 0.24 | 7.4 | 11 | $1408 \times 128 \times 128$ |
| 5 | 3.67 | 1.0 | 6.8 | 105 | 0.12 | 5.4 | 11 | $1408 \times 128 \times 128$ |
| 6 | 3.67 | 1.5 | 13.0 | 158 | 0.08 | 3.8 | 11 | $1408 \times 128 \times 128$ |
| 7 | 3.67 | 2.0 | 19.0 | 211 | 0.06 | 2.9 | 11 | $1716 \times 156 \times 156$ |
| 8 | 3.67 | 3.0 | 36.0 | 316 | 0.04 | 1.8 | 11 | $2816 \times 256 \times 256$ |
| 9 | 4.63 | 0.5 | 2.4 | 53 | 0.24 | 7.2 | 14 | $1792 \times 128 \times 128$ |
| 10 | 4.63 | 1.0 | 6.8 | 105 | 0.12 | 8.2 | 14 | $1792 \times 128 \times 128$ |
| 11 | 4.63 | 1.5 | 13.0 | 158 | 0.08 | 6.8 | 14 | $1792 \times 128 \times 128$ |
| 12 | 5.51 | 0.5 | 2.4 | 53 | 0.24 | 6.5 | 14 | $1792 \times 128 \times 128$ |
| 13 | 5.51 | 1.0 | 6.8 | 105 | 0.12 | 7.4 | 14 | $1792 \times 128 \times 128$ |
| 14 | 5.51 | 2.0 | 19.0 | 211 | 0.06 | 5.0 | 14 | $2184 \times 156 \times 156$ |
| $l_I/l_f = 3.0$ | | | | | | | | |
| 15 | 3.67 | 0.5 | 1.7 | 105 | 0.49 | 11.0 | 5.5 | $1408 \times 256 \times 256$ |
| 16 | 3.67 | 1.0 | 4.8 | 211 | 0.24 | 5.3 | 5.5 | $1408 \times 256 \times 256$ |
| 17 | 3.67 | 1.5 | 8.9 | 316 | 0.16 | 1.6 | 5.5 | $1408 \times 256 \times 256$ |
| 18 | 4.63 | 0.5 | 1.7 | 105 | 0.49 | 7.8 | 7.0 | $1792 \times 256 \times 256$ |
| 19 | 4.63 | 1.0 | 4.8 | 211 | 0.24 | 7.4 | 7.0 | $1792 \times 256 \times 256$ |
| 20 | 4.63 | 1.5 | 8.9 | 316 | 0.16 | 5.1 | 7.0 | $1792 \times 256 \times 256$ |
| 21 | 5.51 | 0.5 | 1.7 | 105 | 0.49 | 9.3 | 7.0 | $1792 \times 256 \times 256$ |
| 22 | 5.51 | 1.5 | 8.9 | 316 | 0.16 | 5.8 | 7.0 | $1792 \times 256 \times 256$ |

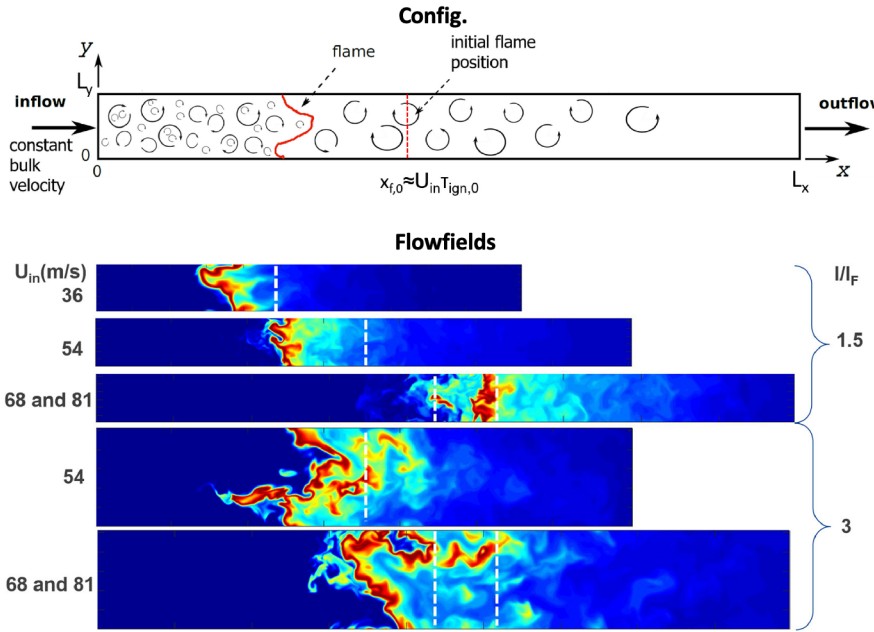

Figure 14: The top figure shows the DNS configuration of freely propagating flame. The bottom figure shows the cross-sectional views of the $H_2O_2$ contours along with the initial position of the spontaneous ignition front (dashed white line). Reprinted from [70], Copyright 2019, with permission from Elsevier.

The turbulent flames are initialized with an ignition front. For the initial flat spontaneous ignition front, the thermo-chemical conditions are chosen to be representative of those at the end of the first stage of

a heavy-duty gas turbine sequential combustor, but at a lower pressure of $1\,\mathrm{atm}$ for all configurations. The mixture of fuel and products of first stage hydrogen-air combustion at an equivalence ratio of 0.43 and initial temperature of 773 K is used at the inlet of the domain. This mixture is equivalent to an equivalence ratio of 0.35 and $T_u = 990\,\mathrm{K}$, and its ignition delay time ($\tau_{\mathrm{ign},0}$) and laminar flame speed ($S_L$) are identified to be $0.55\,\mathrm{ms}$ and $14.7\,\mathrm{ms}^{-1}$, respectively. The reference laminar flame thickness, $l_f$, is evaluated to be $0.66\,\mathrm{mm}$. After initialization, the ignition front is superimposed on a turbulent flow-field using a one-to-one correspondence in x-space (Figure 14). Depending on varying $U_{\mathrm{in}}$ and $u'$, the flame may stabilize at a position far away from the inlet (a turbulent spontaneous ignition front) or the introduction of turbulence may trigger the transition to a deflagration, where the flame front propagates towards the inlet.

The width of the domain in the $y$- and $z$-directions is $L_y = L_z = 5.26 l_I$, and the length in the streamwise direction is $L_x$, which is different for individual configuration. The other associated parameters for all 22 configurations are summarized in Table 6. The turbulent Reynolds number for the unburnt gas is defined as $\mathrm{Re}_{t,u} = u' l_I / \nu_u$, where $\nu_u$ is the kinematic viscosity of the unburnt gas. The Karlovitz number, $\mathrm{Ka}_u$, is defined as the ratio of the flame characteristic time $t_f = l_f / S_L$ to the Kolmogorov characteristic time $t_{\nu u} = (\nu_u l / u'^3)^{1/2}$. The ignition Damköhler number is $\mathrm{Da}_{\mathrm{ign}} = \tau_I / \tau_{\mathrm{ign},0}$, with the large-eddy turnover time $\tau_I = l_I / u'$. For all configurations, the simulations are run until a statistically steady state is achieved.

The low Mach number form of the governing equations is solved using the energy conservative, finite difference code NGA [66] and high turbulence simulations are enabled by the linear velocity forcing method. NGA is second-order accurate in both space and time, and it uses a semi-implicit Crank-Nicolson time integration scheme. A third-order bounded QUICK scheme, BQUICK, is used for scalar transport. Ideal gas law is used as the EoS for a mixture of perfect gases. A detailed chemical mechanism [97] for hydrogen combustion with 9 species and 21 reactions is used for all configurations.

# E   Additional Momentum128 3D SR Dataset Details

The Momentum128 3D SR dataset is a processed subset of BLASTNet 2.0, and available for download at `https://www.kaggle.com/datasets/waitongchung/blastnet-momentum-3d-sr-dataset`.

## E.1   Data Format and Directory Structure

The Momentum128 3D SR Dataset contains velocity and density sub-volumes (see Appendix E.1.1) extracted and processed from BLASTNet 2.0, along with descriptive metadata (Appendix E.1.2), web metadata (Appendix E.1.3), and instructions for reading the data in Appendix E.1.4.

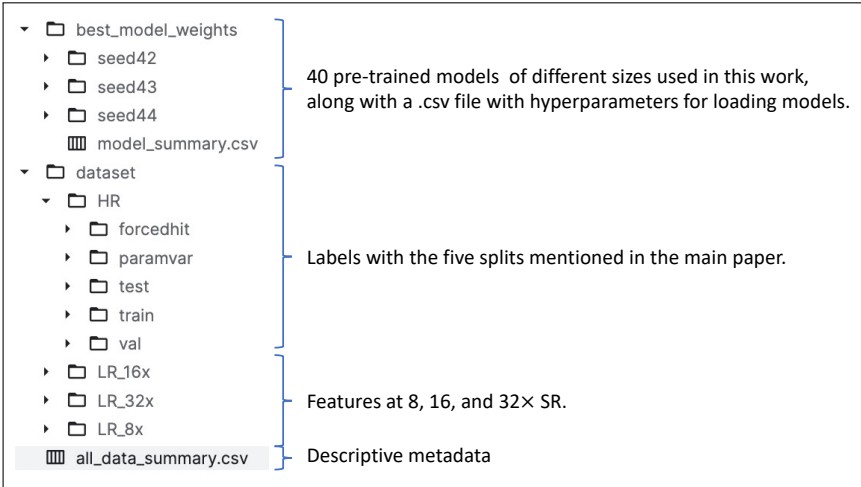

Figure 15: Directory structure of the Momentum128 3D SR dataset.

### E.1.1 Files on Flow Physics

All 2000 sub-volumes (labels with $128 \times 128 \times 128$ number of voxels) of density and three velocity components [$\rho$,**u**] are also presented in little-endian single-precision binary format, in a similar fashion to BLASTNet 2.0 (see Appendix D.1.1), which can be read with `np.fromfile` or `np.memmap`. This is shown in Figure 15, which also shows the five data splits described in Section 3.2. In addition, Favre-filtered features for 8, 16, and $32\times$ SR are also provided, along with pre-trained weights from all models reported in this study. We include a notebook with the method used for obtaining tricubic interpolation in the code repository described in Appendix A. The sub-volume files are named with `<Variable Name and SI Unit>_id<hash value>.dat`, where the hash value provides a unique ID based on the spatial coordinates of the sub-volume location and the index of configuration.

### E.1.2 Descriptive Metadata

In addition to the sub-volumes, we provide `.csv` files that provide information on hash ID, Kaggle ID, short configuration description, k-means cluster index, and spatial grid size for the different dataset splits used in this work.

### E.1.3 Structured Web Metadata

A `http://schema.org` metadata has been added to `https://blastnet.github.io/datasets`, and tested with `https://search.google.com/test/rich-results`.

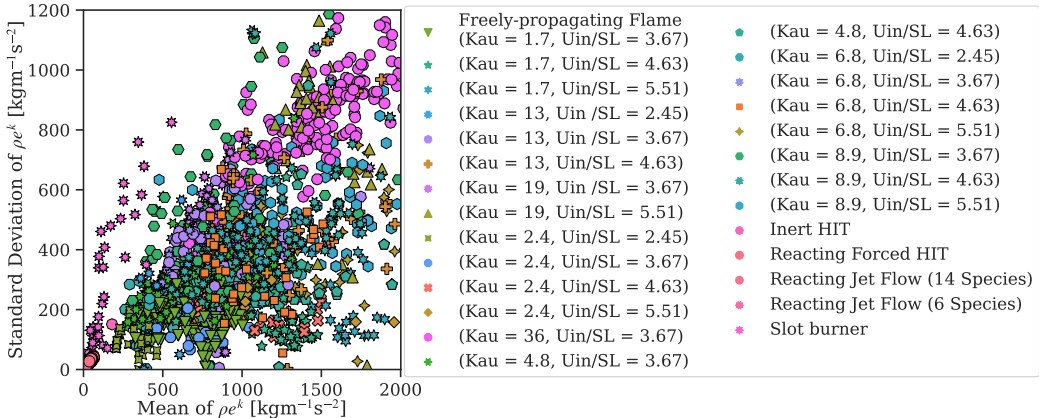

Figure 16: Statistics of the specific kinetic energy $\rho e^k$ of each $128^3$ sub-volume in the Momentum128 3D SR dataset. Kaggle links to source (raw) data is available in Table 4.

### E.1.4 Reading Data

Similar to BLASTNet 2.0, this data can be read by using `np.fromfile/np.memmap` to load and reshape the data. Links to tutorials and dataloaders to perform this are also shared in Appendix A. We also attach a Kaggle notebook for training and evaluating with this data.

### E.1.5 DNS Configurations

We select the 2,000 sub-volumes from BLASTNet 2.0 to form a 67 GB dataset that can fit into a single Kaggle repository in order to develop labels for a compressible turbulence benchmark dataset that can be easily downloaded. Mean and standard deviation of the specific kinetic energy from the resulting sub-volumes are shown in Figure 16, which is a more detailed version of Figure 2.

### E.1.6 Favre-filtered DNS and Coarse-grid Simulations

For the Momentum128 3D SR low-resolution feature samples, we employed Favre-filtering (Equation (3) to generate a canonical surrogate, known as finite-volume optimal LES [74], that has direct

theoretical connections [73, 45] to coarse-grained simulations (also known as implicit LES). While having an implicit LES solution and corresponding DNS data as a feature-label pair within BLAST-Net would be ideal, it is not feasible to obtain a matching implicit LES-DNS pair due to the stochastic nature and time-dependency of fluid simulations. Specifically, small changes in the system (such as grid size) can result in widely different flow behavior due to the chaotic nature of turbulence [72]. As such, numerous works involving turbulence modeling, involving analytic [43, 44] and SR [46, 37] have conventionally employed filtered DNS in place of implicit LES flowfields.

Figure 17 demonstrates the quantitative and qualitative relationship between $8\times$ low-resolution and super-resolved implicit LES and Favre-filtered DNS through normalized turbulent kinetic energy (TKE) and velocity magnitude flowfields, respectively. Prior to evaluation, we normalized length by the domain length; velocity is normalized by its RMS. The TKE spectra is a common tool for analyzing the turbulent properties across different lengthscales (up to the DNS wavenumber $\kappa_{norm}^{DNS}$) through Fourier transform operations [26]. Note that the wavenumber $\kappa$ is inversely proportional to length, *i.e.*, larger wavenumbers correspond to smaller lengthscales. To generate the implicit LES solution, we perform a coarse-grid simulation of the HIT DNS configuration detailed in Appendix D.2.1, with a $16^3$ spatial grid. It can be seen that TKE spectra is truncated in a similar fashion in both low-resolution implicit LES and Favre-filtered DNS at $\kappa_{norm}^{LES}$, demonstrating that turbulence is under-resolved beyond this wavenumber due to the coarse grid. We perform $8\times$ SR on both low-resolution flowfields with the 50.2M gradient-loss RRDB model. The super-resolved Favre-filtered DNS demonstrates excellent agreement between with ground truth DNS, the super-resolved implicit LES demonstrates maintains reasonable accuracy with the ground truth DNS spectra. The deviation in TKE spectra is within a reasonable margin of error, especially compared with similar analysis in other ML works [35, 98] involving turbulence. This result demonstrates that an ML model trained on Favre-Filtered DNS can be employed towards super-resolving implicit LES flowfields while maintaining reasonable spectral behavior.

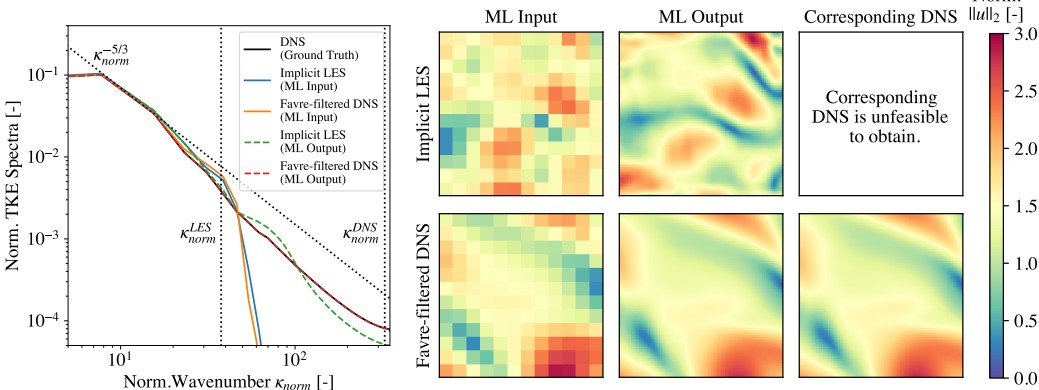

Figure 17: Comparison between normalized turbulent kinetic energy (TKE) and velocity magnitudes between ML (RRDB with gradient loss; 50.2M parameters) inputs and outputs from $8\times$ implicit LES (coarse-grid simulations) and Favre-Filtered DNS of HIT configurations. An ML model trained on Favre-Filtered DNS can be employed towards super-resolving implicit LES flowfields while maintaining reasonable spectral behavior.

## F    Additional Experiment Details

Here, we provide further information that supplements Sections 4 and 5.

### F.1    Additional Metrics Details

For this work, we employ a conventional definition of normalized root-mean-squared error (NRMSE) for evaluating the ML models:

$$\text{NRMSE}(\hat{\phi}, \phi) = \frac{\sum_{i=1}^{N_{vox}} \sum_{j=1}^{N_{samp}} (\phi_{ijk} - \hat{\phi}_{ijk})^2}{\sum_{i=1}^{N_{vox}} \sum_{j=1}^{N_{samp}} \sum_{k=1}^{N_c} \phi_{ij}^2} , \tag{8}$$

with $N_{vox}$ number of voxels and $N_{samp}$ number of samples of model prediction $\hat{\phi}$ and ground truth $\phi$. As discussed in Section 4.2, NRMSE is used to evaluate super-resolved density and velocity, along with subgrid-scale stress divergence, volume-averaged kinetic energy, and volume-averaged turbulent dissipation rate. When evaluating the NRMSE of turbulent dissipation rate, the samples are weighted such that the kinematic viscosity is unity, in order to emphasize contributions from the predicted velocity components.

We also evaluate our models with the a 3D version of the structural similarity index measure (SSIM) [82]. SSIM is used to evaluate super-resolved density and velocity, along with subgrid-scale stress divergence. SSIM is calculated by passing a sliding window of size $9\times9\times9$ (similar to the original SSIM paper [82]) across model prediction $\hat{\phi}$ and ground truth $\phi$, and evaluating their statistical quantities. Specifically, SSIM is defined by:

$$\text{SSIM}(\hat{\phi}, \phi) = \frac{1}{N_{samp}N_w N_c} \sum_{i=1}^{N_{samp}} \sum_{j=1}^{N_w} \left( \frac{2\mu_{\hat{\phi}}\mu_\phi + c_1^2}{\mu_{\hat{\phi}}^2 + \mu_\phi^2 + c_1^2} \cdot \frac{2\sigma_{\hat{\phi}\phi} + c_2^2}{\sigma_{\hat{\phi}}^2 + \sigma_\phi^2 + c_2^2} \right)_{ij}, \tag{9}$$

with mean $\mu_{\{\hat{\phi},\phi\}}$, variance $\sigma^2_{\{\hat{\phi},\phi\}}$, and covariance $\sigma_{\hat{\phi}\phi}$ for $N_w$ number of sliding windows. In computer vision applications with RGB images, $c_1 = 0.01$ and $c_2 = 0.03$ are typically used to ensure numerical stability [82]. However, we found these values insufficient for numerical stability in this work. Hence, we employed $c_1 = 0.1$ and $c_2 = 0.3$. Note that for $\text{SSIM}_{sgs}$ (Equation (4b)), the edge voxels of $\nabla \cdot \tau^{sgs}$ are neglected prior to evaluation, to remove voxels with first-order spatial differencing.

Table 7: Hyperparameters of RRDB, EDSR, and RCAN models investigated in this work.

| RRDB Parameters | 0.6M | 0.9 M | 1.4M | 2.7M | 4.9M | 11.4M | 17.8M | 50.2M |
|---|---|---|---|---|---|---|---|---|
| Residual (RRDB) Blocks | 1 | 1 | 1 | 1 | 2 | 5 | 8 | 23 |
| First Channel Size | 4 | 16 | 32 | 64 | 64 | 64 | 64 | 64 |
| Kernel Size | 3 | 3 | 3 | 3 | 3 | 3 | 3 | 3 |
| RRDB Growth Factor | 32 | 32 | 32 | 32 | 32 | 32 | 32 | 32 |
| Residual Scaling | 0.2 | 0.2 | 0.2 | 0.2 | 0.2 | 0.2 | 0.2 | 0.2 |
| EDSR Parameters | 0.5M | 1.0M | 1.4M | 2.8M | 5.1M | 11.1M | 17.8M | 34.6M |
| Residual Blocks | 32 | 32 | 32 | 32 | 32 | 32 | 32 | 32 |
| Channel Size | 14 | 20 | 24 | 34 | 46 | 68 | 86 | 120 |
| Kernel Size | 3 | 3 | 3 | 3 | 3 | 3 | 3 | 3 |
| Residual Scaling | 0.1 | 0.1 | 0.1 | 0.1 | 0.1 | 0.1 | 0.1 | 0.1 |
| RCAN Parameters | 0.5M | 0.9M | 1.5M | 2.7M | 5.1M | 11.8M | 16.4M | 48.3 M |
| Residual Blocks | 1 | 1 | 1 | 1 | 1 | 10 | 20 | 20 |
| Channel Size | 26 | 34 | 44 | 60 | 64 | 64 | 64 | 64 |
| Residual Groups | 1 | 1 | 1 | 1 | 1 | 2 | 3 | 10 |
| Kernel Size | 3 | 3 | 3 | 3 | 3 | 3 | 3 | 3 |
| Residual Scaling | 1 | 1 | 1 | 1 | 1 | 1 | 1 | 1 |
| Conv-FNO Parameters | − | 0.6M | 1.8M | 2.6M | 5.2M | 9.4M | 20.6M | 32.9M |
| FNO modes | − | 2 | 2 | 2 | 2 | 2 | 2 | 2 |
| FNO Channel Size | − | 14 | 20 | 24 | 34 | 46 | 68 | 86 |
| Conv-FNO Blocks | − | 32 | 32 | 32 | 32 | 32 | 32 | 32 |
| Convolutional Kernel Size | − | 3 | 3 | 3 | 3 | 3 | 3 | 3 |
| Residual Scaling | − | 0.1 | 0.1 | 0.1 | 0.1 | 0.1 | 0.1 | 0.1 |

### F.2 Additional Model Details

In this work, three well-studied 2D ResNet-based [76] SR models are modified from their original repositories for 3D SR: (i) Residual-in-Residual Dense Block (RRDB) [22], (ii) Enhanced Deep Residual Super-resolution (EDSR) [23], and (iii) Residual Channel Attention Networks (RCAN) [24].

We choose to study these models due to their difference in architecture paradigms. Specifically, RRDB employs a residual layers within residual layers; EDSR features an expanded network width; RCAN utilizes long skip connections and channel attention mechanisms. In addition, we consider a model that employs Conv-FNO blocks. Specifically, outputs of an FNO layer and a convolutional layer were added to the outputs of each residual block in the EDSR, in a similar fashion to both Conv-FNO and U-FNO models [25]. This modification enables us to examine combining FNO layers with convolution blocks that have been demonstrated to perform well in SR applications.

To investigate the scaling behavior of the model architectures, we vary the number of parameters by changing the network depth and width. All other hyperparameters are maintained from their original studies, with all models initialized via He et al. [78]. Specifically, the architecture settings for RRDB, EDSR, RCAN, and Conv-FNO are shown in Table 7. In this table, we list the the number of residual blocks, growth factor in RRDB blocks, the channel width, kernel size, number of FNO modes. RCAN residual groups, and residual scaling factors – which are arguments for the model objects in the code described in Appendix A. The hyperparameters within the Conv-FNO model was first determined by comparing two approaches with the same number of parameters (3.0M): (i) one with large number of Fourier modes, and (ii) one with deep and wide Conv-FNO blocks. Since the approach number of modes with deep and wide Conv-FNO blocks demonstrated better validation MSE, another hyperparameter search was performed to determine the optimal number of Fourier modes ranging until the GPU memory was fully consumed at five Fourier modes. We scale the Conv-FNO in Section 5 by increasing the FNO channel size since this approach led to good scaling behavior, especially when compared to increasing the number of Fourier modes.

Table 8: Validation MSE for different hyperparameters of Conv-FNO.

| Conv-FNO Parameters | 3.0M | 3.0M | 5.2 M | 10.9M | 21.7 M | 39.8M |
|---|---|---|---|---|---|---|
| FNO Modes | 12 | 1 | 2 | 3 | 4 | 5 |
| FNO Channel Size | 6 | 34 | 34 | 34 | 34 | 34 |
| Conv-FNO Blocks | 6 | 32 | 32 | 32 | 32 | 32 |
| ↓Val. MSE [$\times 10^{-3}$] | 278 | 175 | **5.3** | 87.8 | 182 | 156 |

## F.3  Additional Loss Details

Similar to other turbulent SR studies [37, 46], all models are trained with mean-squared-error (MSE) loss $L_{\text{MSE}}$, unless otherwise stated. Specifically, for a predicted channel quantity $\hat{\phi}$ and ground truth $\phi$:

$$L_{\text{MSE}} = \frac{1}{N_{vox}N_{samp}N_c} \sum_{i=1}^{N_{vox}} \sum_{j=1}^{N_{samp}} \sum_{k=1}^{N_c} (\phi_{ijk} - \hat{\phi}_{ijk})^2 \,, \tag{10}$$

for $N_{vox}$ number of voxels, $N_{samp}$ number of samples, and $N_c$ number of channel variables. When comparing the use of MSE and mean absolute error (MAE) loss, we found that both models trained with both losses resulted in the similar validation MSE at the end of 1500 epochs. However, MSE loss demonstrated better stability at early timesteps, as shown with the 34.6M EDSR 8× model in Figure 18. This increased robustness motivated our choice of MSE as a loss function.

In addition, we train variants of the RRDB with a physics-informed gradient-based loss resulting in:

$$L_{phys} = (1 - \lambda)L_{\text{MSE}} + \lambda L_{grad} \,, \quad \text{where} \tag{11a}$$

$$L_{grad} = \frac{\Delta^2}{3N_{vox}N_{samp}N_c} \sum_{i=1}^{N_{vox}} \sum_{j=1}^{N_{samp}} \sum_{k=1}^{N_c} \sum_{l=1}^{3} [(\nabla \phi_{ijk})_l - (\nabla \hat{\phi}_{ijk})_l]^2 \,, \tag{11b}$$

The gradient terms are evaluated using `torch.gradient`, which corresponds to a second-order central differencing scheme that is optimized for GPU calculations. This is done on both super-resolved and ground truth fields before inputting the gradient terms into the MSE function. This gradient term enables the ML models to implicitly learn transport phenomena that arise in flow physics PDEs. For example, advection in the mass conservation equation can expressed as:

$$\nabla \cdot (\rho \mathbf{u}) = \mathbf{u} \cdot \nabla \rho + \rho \nabla \cdot \mathbf{u} \,, \tag{12}$$

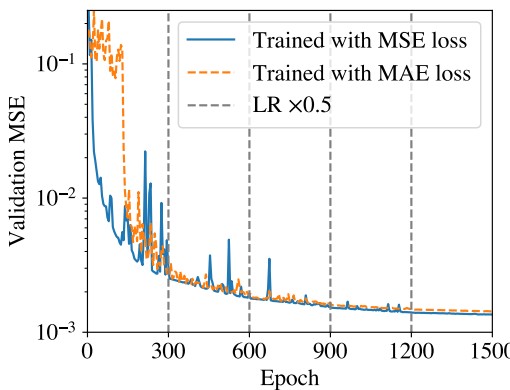

Figure 18: Validation MSE when comparing EDSR 50.2M models trained with MSE and MAE loss.

which requires the gradients of all channel variables to be predicted correctly. These arguments can also be applicable to the advection of momentum, which is the transport term responsible for turbulent phenomena [26].

In Section 5, we employ the weighting factor $\lambda = 0.99$, which was determined from a hyperparameter search on RRDB 2.7M models, with the validation MSE for various lambda shown in Table 9.

Table 9: Validation MSE for different weighting factor $\lambda$ for the gradient-based loss.

| Weighting factor $\lambda$ | 0.000 | 0.100 | 0.300 | 0.500 | 0.700 | 0.900 | **0.990** | 0.999 |
|---|---|---|---|---|---|---|---|---|
| $\downarrow$Val. MSE [$\times 10^{-3}$] | 2.18 | 2.20 | 2.14 | 2.12 | 1.97 | 1.74 | **1.42** | 1.64 |

### F.4 Additional Data Augmentation Details

Data augmentation is performed via variants of random rotation and flip transformations – which we modified to ensure augmented data remains consistent with mass conservation. Specifically, this is necessary for maintaining the reflective and rotational invariance of the divergence of momentum $\nabla \cdot (\rho \mathbf{u})$, after transformation. The steps to ensure this are summarized in Figure 19, which demonstrates how flip and rotation operations can still result in continuity-consistent transformations on a 2D image. These operations have been extended for 3D random flip and rotation, which we employed during training. Links to code, with implementations of these transformations and corresponding unit tests, are provided in the code described in Appendix A.

### F.5 Additional Training Details

For evaluation, we select models with the best MSE after training for 1500 epochs with a batch size of 64 across 16 Nvidia V100 GPUs. Learning rate is initialized at `1e-4`, and halved every 300 epochs. Both the number of training iterations and learning scheduling are chosen to match other SR studies [22–24] and are found to be sufficient for the SR prediction as shown in Figure 20, where flat validation loss curves are seen at 1500 epochs for vanilla RRDB 50.2M at all SR ratios shown. We note that at $32\times$ SR, the model begins to show overfitting after 600 epochs. This may be because the SR models cannot learn a generalizable pattern from the insufficient information contained within the coarse-grained features at high SR ratios.

### F.6 FLOPs estimation

In this work, theoretical FLOPs for the ML models is estimated via `THOPs` (https://github.com/Lyken17/pytorch-OpCounter) which has been used in other studies [99, 100]. Einstein summation operations in FNO layers were evaluated through modifying `THOPS` with `numpy.einsum_path`, while Fourier and inverse Fourier transforms are estimated as $5N_{points} \log N_{points}$ [53].

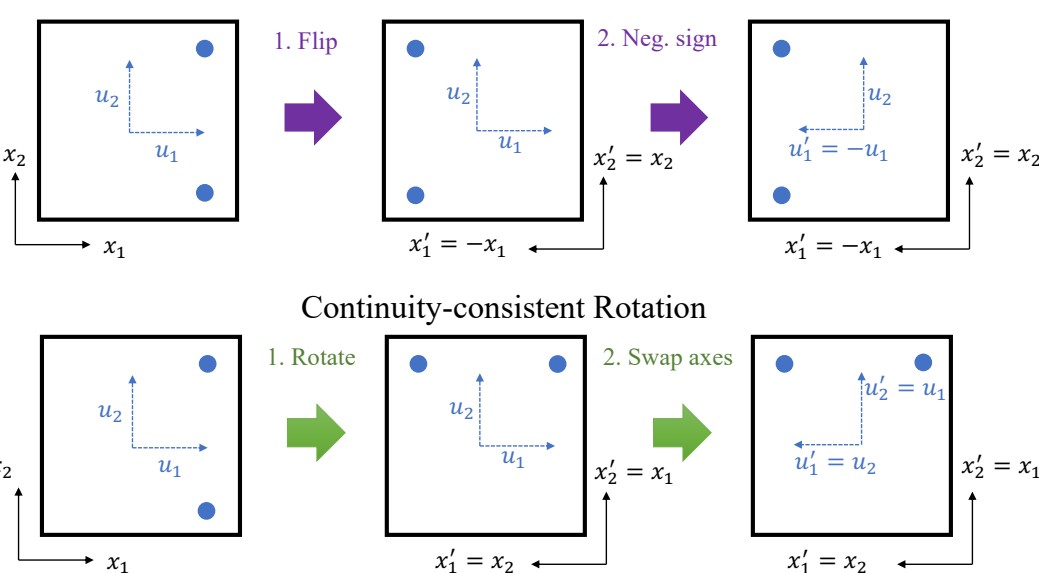

Figure 19: Continuity-consistent augmentation on a 2D image that preserves reflective and rotational invariance of the $\nabla \cdot (\rho \mathbf{u})$.

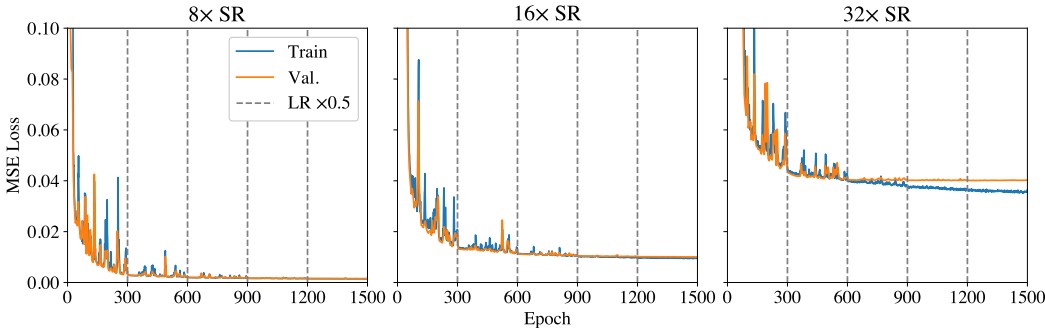

Figure 20: Vanilla RRDB 50.2M MSE loss at 8, 16, and 32× SR, with initial learning rate LR = `1e-4`.

For tricubic interpolation, an interpolant $p$ at coordinates $x$, $y$, and $z$ is defined by [101]:

$$p(x,y,z) = \sum_{i=0}^{3}\sum_{j=0}^{3}\sum_{k=0}^{3} a_{ijk} x^i y^j z^k \,. \tag{13}$$

For a single point, this totals to 63 additions and 84 multiplications for the matrix multiplication applications, with additional 9 multiplications for the exponents. The elements of tensor $a_{ijk}$ can be found in a 64-element vector $\alpha$, which can in turn be found via. Hence:

$$\alpha = A_1^{-1} b \,, \tag{14}$$

where $A_1$ is a 64×64 integer matrix of constants that is predefined in the algorithm, and $b$ is a 64-element vector which contains derivatives from the input data $\phi$. For a uniform spatial grid, $b$ can be evaluated by evaluating derivatives on a sub-volume with 4×4×4 vertices, which can be flattened to a 64-element vector $\phi_{sub}$:

$$\begin{aligned} 8\alpha &= A_1^{-1} A_2 \phi_{sub} \\ &= B\phi_{sub} \,, \end{aligned} \tag{15}$$

where $A_2$ contains integer coefficients for finite-differencing and $B = A_1^{-1} A_2$ contains 2765 zero elements. This totals to 4032 additions and 4096 multiplications, when considering dense matrices.

If sparse multiplication is used, this results in 1301 multiplications and approximately 1237 additions, with 64 multiplications for dividing by 8. Putting this together:

$$\text{FLOPs}_{tri}^{dense} = 8328 N_{vox} N_c \,, \tag{16}$$

$$\text{FLOPs}_{tri}^{sparse} = 2738 N_{vox} N_c \,, \tag{17}$$

In this work, our samples contain $128^3$ voxels and 4 channels. Thus, tricubic interpolation costs **23 GFLOPs** (reported in Section 5) for sparse matrix multiplication, while employing dense matrix multiplication costs 69 GFLOPs, per inference with batch size of 1.

## G   Additional Results

### G.1   Additional Evaluation

In Figure 21, we present normalized absolute error in specific kinetic energy $|\epsilon_{\rho e^k}|/\rho e_{max}^k$ from model predictions shown in Figure 3.

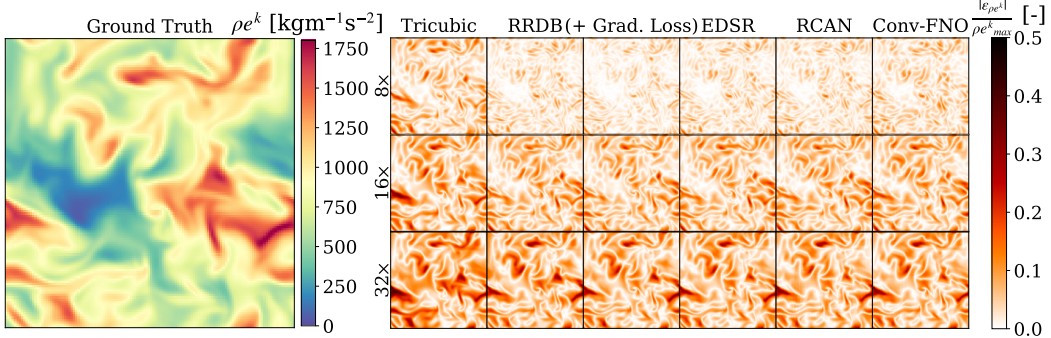

Figure 21: Normalized absolute error in specific kinetic energy $|\epsilon_{\rho e^k}|/\rho e_{max}^k$ from model predictions shown in Figure 3.

Tables 10 to 12 report additional results from evaluating the models from Tables 2 and 3 with NMRSE metrics. In Table 13, we report the SSIM used for evaluating the scaling behavior of all $8\times$ SR models discussed in Section 5.

Table 10: Comparison of NRMSE of five models at 16 and $32\times$ SR ratios, evaluated with the baseline test set. Mean and standard deviation from three seeds are reported here. **Bold** term represents best mean.

| Models | $\downarrow\text{NRMSE}_{\rho,\mathbf{u}}$ $(\times 10^{-2})$ | $\downarrow\text{NRMSE}_{sgs}$ $(\times 10^{-1})$ | $\downarrow\text{NRMSE}_{E^k}$ $(\times 10^{-4})$ | $\downarrow\text{NRMSE}_{\epsilon}$ $(\times 10^{-1})$ |
|---|---|---|---|---|
| RRDB 16× | 5.6±0.0 | 4.2±0.1 | **5.5±0.1** | 4.5±0.0 |
| (+ Grad. Loss) | **5.3±0.0** | **3.9±0.1** | 7.7±1.0 | **3.7±0.0** |
| EDSR 16× | 6.0±0.2 | 4.5±0.2 | 6.1±0.1 | 4.7±0.1 |
| RCAN 16× | 6.7±0.6 | 5.3±0.6 | 6.7±0.6 | 4.9±0.1 |
| Conv-FNO 16× | 7.4±0.3 | 5.9±0.3 | 9.0±0.8 | 5.1±0.0 |
| Tricubic 32× | 28.3 | 9.5 | 156.2 | 9.6 |
| RRDB 32× | 20.2±0.1 | 7.7±0.1 | **39.3±1.9** | 8.3±0.0 |
| (+ Grad. Loss) | **19.8±0.0** | **7.5±0.1** | 48.1±2.2 | 8.3±0.0 |
| EDSR 32× | 20.1±0.0 | 7.7±0.1 | 45.4±1.3 | 8.4±0.0 |
| RCAN 32× | 20.3±0.1 | 7.9±0.1 | 43.7±2.8 | **8.1±0.1** |
| Conv-FNO 32× | 20.5±0.1 | 8.0±0.1 | 48.2±3.5 | 8.4±0.1 |

Table 11: Comparison of NRMSE of five models at 16 and $32\times$ SR ratios, evaluated with the forced HIT set. Mean and standard deviation from three seeds are reported here. **Bold** term represents best mean.

| Models | $\downarrow$NRMSE$_{\rho,\mathbf{u}}$ $(\times 10^{-3})$ | $\downarrow$NRMSE$_{sgs}$ $(\times 10^{-2})$ | $\downarrow$NRMSE$_{E^k}$ $(\times 10^{-6})$ | $\downarrow$NRMSE$_{\epsilon}$ $(\times 10^{-4})$ |
|---|---|---|---|---|
| Tricubic 16$\times$ | 36.2 | 61.2 | 7637.2 | 2517.8 |
| RRDB 16$\times$ | 3.5±0.1 | 9.7±0.1 | 40.8±15.1 | 48.9±3.6 |
| (+ Grad. Loss) | **3.3±0.0** | **9.4±0.2** | 242.4±89.6 | **45.6±12.5** |
| EDSR 16$\times$ | 4.0±0.3 | 13.0±1.3 | **23.3±7.1** | 83.6±21.9 |
| RCAN 16$\times$ | 5.1±1.1 | 18.8±4.9 | 46.6±22.3 | 97.9±30.2 |
| Conv-FNO 16$\times$ | 6.6±0.8 | 24.8±2.8 | 124.2±86.1 | 90.0±10.1 |
| Tricubic 32$\times$ | 118.4 | 81.0 | 68869.9 | 6689.3 |
| RRDB 32$\times$ | 47.6±0.7 | 45.5±1.7 | **1985.7±554.5** | 1753.9±57.2 |
| (+ Grad. Loss) | **45.6±0.3** | **41.3±0.5** | 2753.6±1260.0 | 1758.2±84.3 |
| EDSR 32$\times$ | 48.4±1.0 | 47.7±1.8 | 5709.1±1100.1 | 2100.7±86.9 |
| RCAN 32$\times$ | 49.6±1.9 | 47.5±2.0 | 6288.2±809.4 | **1481.6±274.2** |
| Conv-FNO 32$\times$ | 48.8±0.5 | 49.1±1.7 | 5991.4±105.7 | 1906.3±177.4 |

Table 12: Comparison of NRMSE of five models at 8, 16, and $32\times$ SR ratios, evaluated with the parametric variation set. Mean and standard deviation from three seeds are reported here. **Bold** term represents best mean.

| Models | $\downarrow$NRMSE$_{\rho,\mathbf{u}}$ $(\times 10^{-2})$ | $\downarrow$NRMSE$_{sgs}$ $(\times 10^{-1})$ | $\downarrow$NRMSE$_{E^k}$ $(\times 10^{-4})$ | $\downarrow$NRMSE$_{\epsilon}$ $(\times 10^{-1})$ |
|---|---|---|---|---|
| Tricubic 8$\times$ | 37.5 | 70.6 | 301.1 | 3329.1 |
| RRDB 8$\times$ | 4.7±0.1 | 18.1±0.4 | 7.0±3.5 | 551.5±7.0 |
| (+ Grad. Loss) | **2.7±0.0** | **8.7±0.1** | 15.9±5.2 | **232.7±11.1** |
| EDSR 8$\times$ | 4.2±0.3 | 16.2±1.5 | 1.4±0.4 | 487.7±39.7 |
| RCAN 8$\times$ | 4.3±0.0 | 16.3±0.2 | **1.3±0.3** | 493.3±5.1 |
| Conv-FNO 8$\times$ | 8.9±0.8 | 37.2±2.2 | 6.3±3.1 | 830.3±38.5 |
| Tricubic 16$\times$ | 114.7 | 83.7 | 1416.9 | 7139.8 |
| RRDB 16$\times$ | 39.6±0.3 | 36.0±0.4 | 83.9±22.2 | 3409.4±37.1 |
| (+ Grad. Loss) | **36.0±0.3** | **30.9±0.3** | 140.9±27.9 | **2923.1±57.6** |
| EDSR 16$\times$ | 42.2±1.7 | 40.0±2.6 | 52.6±5.7 | 3556.5±108.8 |
| RCAN 16$\times$ | 48.7±5.2 | 48.3±6.7 | **47.0±6.4** | 3538.0±108.9 |
| Conv-FNO 16$\times$ | 54.8±2.9 | 55.8±2.8 | 104.5±29.5 | 3617.7±80.0 |
| Tricubic 32$\times$ | 242.6 | 93.0 | 4743.3 | 9374.6 |
| RRDB 32$\times$ | 162.0±0.2 | 72.2±0.3 | 615.0±110.0 | 7546.0±19.1 |
| (+ Grad. Loss) | **161.5±0.3** | **70.5±0.6** | 820.2±151.2 | 7694.2±3.9 |
| EDSR 32$\times$ | 162.4±0.1 | 74.0±1.1 | 491.3±2.5 | 7623.6±83.0 |
| RCAN 32$\times$ | 164.6±0.8 | 73.3±1.3 | **427.5±73.5** | **6961.8±261.0** |
| Conv-FNO 32$\times$ | 166.7±1.7 | 74.2±0.3 | 590.9±91.9 | 7450.4±278.4 |

## G.2 Effects of Different Normalization during Evaluation

For the results in Section 5, all evaluation sets are normalized with their own mean and standard deviation, as shown in Table 14, prior to testing. Note that we evaluate the same mean and standard deviation for all three velocity components. This is to account for the random rotation applied to the features and labels, as detailed in Appendix F.4, which results in the velocity channels swapping axes with each other during training.

When comparing the two normalization approaches in Table 15, we observe that poor performance is seen during evaluation on the Forced HIT set when normalizing via the train set, with SSIM$_{sgs}$ = 0,

Table 13: Summary of models, of different number of parameters $N_p$, trained at $8\times$ SR. **Bold** term represents best mean for a given model approach.

| Models | $N_p$ | Baseline Test Set | | Param. Variation Set | | Forced HIT Set | | ↓GFLOPs |
|---|---|---|---|---|---|---|---|---|
| | | ↑SSIM$_{\rho,\mathbf{u}}$ | ↑SSIM$_{sgs}$ | ↑SSIM$_{\rho,\mathbf{u}}$ | ↑SSIM$_{sgs}$ | ↑SSIM$_{\rho,\mathbf{u}}$ | ↑SSIM$_{sgs}$ | |
| Tricubic | – | 0.820 | 0.431 | 0.800 | 0.418 | 0.951 | 0.711 | 22 |
| RRDB | 0.6M | 0.476±0.044 | 0.022±0.009 | 0.495±0.050 | 0.027±0.012 | 0.610±0.078 | 0.055±0.028 | **10** |
| | 0.9M | 0.754±0.003 | 0.248±0.026 | 0.775±0.004 | 0.330±0.044 | 0.977±0.000 | 0.523±0.065 | 76 |
| | 1.4M | 0.841±0.003 | 0.517±0.012 | 0.839±0.003 | 0.574±0.005 | 0.991±0.000 | 0.791±0.006 | 273 |
| | 2.7M | 0.878±0.001 | 0.608±0.002 | 0.867±0.002 | 0.649±0.002 | 0.996±0.000 | 0.845±0.003 | 1041 |
| | 4.9M | 0.884±0.005 | 0.635±0.016 | 0.873±0.006 | 0.677±0.015 | 0.996±0.000 | 0.856±0.004 | 1059 |
| | 11.4M | 0.888±0.002 | 0.657±0.005 | 0.878±0.001 | 0.701±0.009 | 0.996±0.000 | 0.867±0.004 | 1112 |
| | 17.8M | 0.893±0.001 | 0.672±0.004 | 0.884±0.000 | 0.719±0.005 | 0.996±0.000 | 0.873±0.002 | 1165 |
| | 50.2M | **0.907±0.003** | **0.715±0.004** | **0.898±0.003** | **0.755±0.002** | **0.997±0.000** | **0.891±0.003** | 1430 |
| (+ Grad. Loss) | 0.6M | 0.475±0.002 | 0.069±0.002 | 0.495±0.002 | 0.083±0.003 | 0.473±0.034 | 0.054±0.004 | **10** |
| | 0.9M | 0.809±0.011 | 0.558±0.012 | 0.827±0.009 | 0.618±0.010 | 0.988±0.003 | 0.819±0.016 | 76 |
| | 1.4M | 0.872±0.001 | 0.665±0.003 | 0.872±0.003 | 0.713±0.002 | 0.996±0.000 | 0.882±0.003 | 273 |
| | 2.7M | 0.907±0.004 | 0.733±0.003 | 0.899±0.004 | 0.768±0.002 | 0.997±0.000 | 0.916±0.002 | 1041 |
| | 4.9M | 0.912±0.002 | 0.745±0.003 | 0.904±0.002 | 0.781±0.001 | 0.997±0.000 | 0.921±0.003 | 1059 |
| | 11.4M | 0.920±0.001 | 0.767±0.004 | 0.911±0.002 | 0.801±0.001 | 0.997±0.000 | 0.931±0.004 | 1112 |
| | 17.8M | 0.917±0.004 | 0.771±0.002 | 0.909±0.002 | 0.802±0.002 | 0.997±0.000 | 0.923±0.003 | 1165 |
| | 50.2M | **0.936±0.003** | **0.802±0.003** | **0.929±0.001** | **0.825±0.001** | **0.998±0.000** | **0.944±0.005** | 1430 |
| EDSR | 0.5M | 0.799±0.013 | 0.480±0.017 | 0.812±0.009 | 0.543±0.012 | 0.989±0.002 | 0.784±0.021 | 34 |
| | 1.0M | 0.834±0.012 | 0.531±0.022 | 0.838±0.009 | 0.584±0.020 | 0.993±0.001 | 0.829±0.013 | 66 |
| | 1.4M | 0.855±0.008 | 0.567±0.015 | 0.855±0.007 | 0.617±0.012 | 0.995±0.000 | 0.857±0.006 | 94 |
| | 2.8M | 0.883±0.007 | 0.618±0.012 | 0.876±0.004 | 0.658±0.010 | 0.997±0.000 | 0.889±0.007 | 181 |
| | 5.1M | 0.901±0.004 | 0.665±0.011 | 0.888±0.004 | 0.698±0.010 | 0.998±0.000 | 0.909±0.001 | 325 |
| | 11.1M | 0.915±0.000 | 0.707±0.002 | 0.902±0.000 | 0.735±0.001 | 0.998±0.000 | 0.919±0.002 | 695 |
| | 17.8M | 0.918±0.004 | 0.718±0.011 | 0.904±0.004 | 0.747±0.011 | 0.998±0.000 | 0.926±0.007 | 1101 |
| | 34.6M | **0.928±0.004** | **0.748±0.012** | **0.916±0.005** | **0.775±0.010** | **0.999±0.000** | **0.937±0.005** | 2122 |
| RCAN | 0.5M | 0.869±0.006 | 0.581±0.011 | 0.865±0.005 | 0.624±0.012 | 0.996±0.001 | 0.869±0.010 | 100 |
| | 0.9M | 0.878±0.010 | 0.605±0.014 | 0.871±0.009 | 0.645±0.013 | 0.997±0.000 | 0.886±0.011 | 166 |
| | 1.5M | 0.900±0.002 | 0.643±0.003 | 0.890±0.002 | 0.672±0.003 | 0.998±0.000 | 0.911±0.002 | 272 |
| | 2.7M | 0.905±0.001 | 0.660±0.004 | 0.893±0.002 | 0.685±0.006 | 0.998±0.000 | 0.914±0.000 | 495 |
| | 5.1M | 0.916±0.002 | 0.712±0.007 | 0.902±0.003 | 0.741±0.007 | 0.998±0.000 | 0.924±0.004 | 578 |
| | 11.9M | 0.913±0.012 | 0.713±0.030 | 0.901±0.013 | 0.743±0.028 | 0.998±0.000 | 0.927±0.010 | 633 |
| | 16.4M | **0.928±0.000** | **0.753±0.002** | **0.916±0.001** | **0.778±0.001** | **0.999±0.000** | **0.941±0.003** | 671 |
| | 48.3M | 0.917±0.002 | 0.724±0.007 | 0.905±0.001 | 0.753±0.006 | 0.998±0.000 | 0.932±0.005 | 931 |
| Conv-FNO | 0.6M | 0.446±0.013 | 0.038±0.017 | 0.465±0.014 | 0.045±0.019 | 0.491±0.011 | 0.050±0.011 | 30 |
| | 1.8M | 0.453±0.002 | 0.048±0.000 | 0.473±0.002 | 0.060±0.000 | 0.493±0.001 | 0.058±0.001 | 77 |
| | 2.6M | 0.457±0.000 | 0.053±0.000 | 0.476±0.001 | 0.062±0.003 | 0.493±0.005 | 0.062±0.001 | 108 |
| | 5.2M | 0.491±0.000 | 0.155±0.015 | 0.502±0.001 | 0.178±0.002 | 0.696±0.021 | 0.205±0.025 | 210 |
| | 9.4M | 0.682±0.059 | 0.279±0.118 | 0.719±0.051 | 0.342±0.127 | 0.961±0.014 | 0.541±0.136 | 376 |
| | 20.6M | 0.802±0.002 | 0.504±0.002 | 0.816±0.003 | 0.571±0.009 | 0.988±0.001 | 0.802±0.001 | 805 |
| | 33.0M | **0.840±0.018** | **0.556±0.022** | **0.840±0.013** | **0.606±0.017** | **0.992±0.002** | **0.839±0.008** | 1276 |

Table 14: Mean and standard deviation of channel quantities. Values rounded to 2 significant figures.

| Split Dataset | $\rho$ [kgm$^{-3}$] | | $u_i$ [ms$^{-1}$] | |
|---|---|---|---|---|
| | Mean | Std. Dev. | Mean | Std. Dev. |
| Train Set | 0.23 | 0.068 | 28 | 48.0 |
| Baseline Test Set | 0.24 | 0.068 | 29 | 48.0 |
| Parametric Variation Set | 0.23 | 0.059 | 34 | 55.0 |
| Forced HIT | 11.00 | 4.600 | 0 | 1.4 |

and low SSIM$_{\rho,\mathbf{u}}$ when compared to tricubic interpolation. This is due to the highly different conditions (much higher density and lower velocity) in the Forced HIT set. However, when comparing the two normalization approaches in Table 16, we observe that slightly better performance is seen across the two metrics during evaluation on the Parametric Variation set when normalizing via

Table 15: Evaluating models on the Forced HIT set, with normalization involving means and standard deviations from train and Forced HIT sets.

| Model | $N_p$ | $\uparrow$SSIM$_{\rho,\mathbf{u}}$ | | $\uparrow$SSIM$_{sgs}$ | |
|---|---|---|---|---|---|
| | | Normalization with Mean and Std. Dev. from: | | | |
| | | Forced HIT Set | Train Set | Forced HIT Set | Train Set |
| Tricubic 8× | – | 0.951 | **0.951** | 0.711 | **0.711** |
| RRDB 8× | 50.2M | 0.997±0.000 | 0.258±0.003 | 0.891±0.003 | 0.000±0.000 |
| (+ Grad. Loss) | | 0.998±0.000 | 0.255±0.005 | **0.944±0.005** | 0.000±0.000 |
| EDSR 8× | 34.6M | 0.999±0.000 | 0.259±0.001 | 0.937±0.005 | 0.000±0.000 |
| RCAN 8× | 16.4M | **0.999±0.000** | 0.261±0.001 | 0.941±0.003 | 0.000±0.000 |
| Conv-FNO 8× | 33.0M | 0.849±0.009 | 0.619±0.014 | 0.249±0.001 | -0.000±0.000 |
| Tricubic 16× | – | 0.876 | **0.876** | 0.432 | **0.432** |
| RRDB 16× | 50.3M | 0.971±0.000 | 0.248±0.000 | 0.805±0.003 | 0.000±0.000 |
| (+ Grad. Loss) | | **0.973±0.000** | 0.248±0.001 | **0.816±0.001** | 0.000±0.000 |
| EDSR 16× | 37.8M | 0.969±0.001 | 0.249±0.000 | 0.783±0.008 | 0.000±0.000 |
| RCAN 16× | 17.3M | 0.961±0.009 | 0.244±0.005 | 0.737±0.050 | 0.000±0.000 |
| Conv-FNO 16× | 34.6M | 0.644±0.012 | 0.357±0.022 | 0.245±0.001 | 0.000±0.000 |
| Tricubic 32× | – | 0.758 | **0.758** | 0.156 | **0.156** |
| RRDB 32× | 50.4M | 0.845±0.001 | 0.242±0.001 | 0.494±0.011 | 0.000±0.000 |
| (+ Grad. Loss) | | **0.850±0.000** | 0.245±0.001 | **0.516±0.012** | 0.000±0.000 |
| EDSR 32× | 40.9M | 0.845±0.001 | 0.244±0.002 | 0.463±0.005 | 0.000±0.000 |
| RCAN 32× | 18.2M | 0.837±0.003 | 0.239±0.003 | 0.448±0.012 | 0.000±0.000 |
| Conv-FNO 32× | 36.2M | 0.472±0.001 | 0.177±0.002 | 0.238±0.002 | 0.000±0.000 |

Table 16: Evaluating models on the Parametric Variation set, with normalization involving means and standard deviations from train and Parametric Variation sets.

| Model | $N_p$ | $\uparrow$SSIM$_{\rho,\mathbf{u}}$ | | $\uparrow$SSIM$_{sgs}$ | |
|---|---|---|---|---|---|
| | | Normalization with Mean and Std. Dev. from: | | | |
| | | Param. Var. Set | Train Set | Param. Var. Set | Train Set |
| Tricubic 8× | – | 0.800 | 0.800 | 0.418 | 0.418 |
| RRDB 8× | 50.2M | 0.898±0.003 | 0.901±0.003 | 0.755±0.002 | 0.760±0.003 |
| (+ Grad. Loss) | | **0.929±0.001** | **0.932±0.002** | **0.825±0.001** | **0.830±0.001** |
| EDSR 8× | 34.6M | 0.916±0.005 | 0.917±0.005 | 0.775±0.010 | 0.779±0.010 |
| RCAN 8× | 16.4M | 0.916±0.001 | 0.918±0.000 | 0.778±0.001 | 0.784±0.000 |
| Tricubic 16× | – | 0.620 | 0.620 | 0.173 | 0.173 |
| RRDB 16× | 50.3M | 0.700±0.001 | 0.703±0.001 | 0.512±0.002 | 0.518±0.001 |
| (+ Grad. Loss) | | **0.719±0.004** | **0.721±0.003** | **0.556±0.002** | **0.559±0.001** |
| EDSR 16× | 37.8M | 0.693±0.005 | 0.695±0.005 | 0.481±0.019 | 0.484±0.019 |
| RCAN 16× | 17.3M | 0.665±0.024 | 0.668±0.023 | 0.415±0.058 | 0.417±0.059 |
| Tricubic 32× | – | 0.476 | 0.476 | 0.087 | 0.087 |
| RRDB 32× | 50.4M | 0.482±0.000 | 0.484±0.000 | 0.186±0.006 | 0.187±0.004 |
| (+ Grad. Loss) | | **0.483±0.001** | **0.485±0.000** | **0.188±0.002** | **0.192±0.002** |
| EDSR 32× | 40.9M | 0.481±0.002 | 0.482±0.001 | 0.187±0.004 | 0.184±0.004 |
| RCAN 32× | 18.2M | 0.469±0.002 | 0.470±0.002 | 0.185±0.005 | 0.181±0.005 |

the train set compared to normalizing via the Parametric Variation set ($\sim$15% higher mean and standard deviation). Nevertheless, in this work, we choose to normalize via the evaluation sets during inferencing in order to employ a consistent approach that enables sufficiently good performance across all evaluation sets. Note that results on normalization via the test set is not shown, since the mean and standard deviation of the test set is similar to those from the train set.

## H Datasheets for Datasets

The following datasheets for BLASTNet 2.0 and Momentum128 3D SR Datasets are based on Datasheets for Datasets [102].

### H.1 BLASTNet 2.0

> ## MOTIVATION

**For what purpose was the dataset created?** Was there a specific task in mind? Was there a specific gap that needed to be filled? Please provide a description.
BLASTNet 2.0 was developed to provide researchers in reacting and non-reacting flow physics communities with high-fidelity publicly accessible simulation data for ML applications. With 2.2 TB, 744 full-domain samples, and 34 configurations, BLASTNet can effectively address gaps in data availability and aid in fostering open/fair ML development within reacting and non-reacting flow physics communities. This data is useful for fluid flows in a wide range of ML applications tied to propulsion, energy, and the environment. Specifically, scientific tasks related to these domains may include turbulent closure modeling [37], spatio-temporal modeling [49], and inverse modeling [18].

**Who created this dataset (e.g., which team, research group) and on behalf of which entity (e.g., company, institution, organization)?**
BLASTNet was initiated by Wai Tong Chung, Ki Sung Jung, Jacqueline H. Chen, and Matthias Ihme [8]. The datasets in BLASTNet were generated by the following researchers:

1. Wai Tong Chung, Jack Guo, Davy Brouzet and Matthias Ihme at Stanford University.
2. Jacqueline H. Chen and Ki Sung Jung at Sandia National Laboratory.
3. Mohsen Talei and Bin Jiang at University of Melbourne.
4. Bruno Savard at Polytechnique Montréal.
5. Alexei Poludnenko at University of Connecticut.

Bassem Akoush, Pushan Sharma and Alex Tamkin at Stanford University contributed in administering, improving, maintaining, and documenting the dataset curation process.

**What support was needed to make this dataset?** (e.g.who funded the creation of the dataset? If there is an associated grant, provide the name of the grantor and the grant name and number, or if it was supported by a company or government agency, give those details.)
This work is funded by:

1. The U.S. Department of Energy National Nuclear Security Administration, under award No. DE-NA0003968.
2. The NASA Early Stage Innovation Program with award No. 80NSSC22K0257.
3. The Department of Energy Office of Energy Efficiency Renewable Energy (EERE) with award No. DE-EE0008875.
4. Wai Tong Chung received partial financial support from the Stanford Institute for Human-centered Artificial Intelligence Graduate Fellowship.

This research used resources of the National Energy Research Scientific Computing Center (NERSC), a U.S. Department of Energy Office of Science User Facility located at Lawrence Berkeley National Laboratory, operated under Contract No. DE-AC02-05CH11231 using NERSC award ERCAP0021046.

**Any other comments?**
No.

## COMPOSITION

**What do the instances that comprise the dataset represent (e.g., documents, photos, people, countries)?**   Are there multiple types of instances (e.g., movies, users, and ratings; people and interactions between them; nodes and edges)? Please provide a description.
Each instance consists of 3D domain of velocity, temperature, pressure, density and mass fractions of chemical species. These variables are saved as flat arrays in separate `.dat` binary files, which can be loaded and reshaped to construct the 3D volumes. To enable high I/O speed in loading arrays, the data has consistent little-endian single-precision binaries that can be read with `np.fromfile`/`np.memmap`.

**How many instances are there in total (of each type, if appropriate)?**
BLASTNet contains a total of 744 full-domain samples from a diverse collection of 34 DNS configuration.

**Does the dataset contain all possible instances or is it a sample (not necessarily random) of instances from a larger set?**   If the dataset is a sample, then what is the larger set? Is the sample representative of the larger set (e.g., geographic coverage)? If so, please describe how this representativeness was validated/verified. If it is not representative of the larger set, please describe why not (e.g., to cover a more diverse range of instances, because instances were withheld or unavailable).
BLASTNet data covers the full physical three-dimensional spatial domain defined during simulation. However, the data contains different number of timesteps, *e.g.*, inert HIT [20] has uniform samples for 99 timesteps, whereas some of the configurations [27, 29, 67] contain single snapshots.

**What data does each instance consist of?**   "Raw" data (e.g., unprocessed text or images) or features? In either case, please provide a description.
Each instance consists of 3D flowfields of velocity, temperature, pressure, density and mass fractions of chemical species. These are raw simulation data that have been pre-processed to a consistent format for ML applications.

**Is there a label or target associated with each instance?**   If so, please provide a description.
Yes. All the thermodynamic and chemical quantities are labeled and can be used directly for certain regression problems. On the other hand, the flow-field data can also be used to derive additional labels for particular applications.

**Is any information missing from individual instances?**   If so, please provide a description, explaining why this information is missing (e.g., because it was unavailable). This does not include intentionally removed information, but might include, e.g., redacted text.
No.

**Are relationships between individual instances made explicit (e.g., users' movie ratings, social network links)?**   If so, please describe how these relationships are made explicit.
Yes. All instances are related by the same governing equations. Some configurations share multiple time instances. These timesteps are made explicit in a descriptive `info.json` file.

**Are there recommended data splits (e.g., training, development/validation, testing)?**   If so, please provide a description of these splits, explaining the rationale behind them.
No.

**Are there any errors, sources of noise, or redundancies in the dataset?**   If so, please provide a description.
Yes. Well-established high-order numerical solvers [29, 64–67] have been employed, with spatial discretization schemes ranging from 2nd- to 8th-order accuracy and time advancement accuracy ranging from 2nd- to 4th-order. Thus, this data is still subject to small numerical errors. While the

compressible reacting flow equations (Equation (18)), solved to generate this data, are valid for a wide range of conditions, errors can originate from certain assumptions in chemical modeling. Specifically, skeletal finite-rate mechanisms and mixture-averaged transport used in these reacting flow DNS have been validated for their configurations, but are not as fully-representative of real-world reactions. However, rectifying this would require detailed mechanisms (introducing an order of magnitude more PDEs) and multi-component transport that can result in intractable calculations [86].

$$\partial_t \rho + \nabla \cdot (\rho \mathbf{u}) = 0 \, , \tag{18a}$$

$$\partial_t (\rho \mathbf{u}) + \nabla \cdot (\rho \mathbf{u} \otimes \mathbf{u}) = -\nabla p + \nabla \cdot \boldsymbol{\tau} \, , \tag{18b}$$

$$\partial_t (\rho e^t) + \nabla \cdot [\mathbf{u}(\rho e^t + p)] = -\nabla \cdot \mathbf{q} + \nabla \cdot [(\boldsymbol{\tau}) \cdot \mathbf{u}] \, , \tag{18c}$$

$$\partial_t (\rho Y_k) + \nabla \cdot (\rho \mathbf{u} Y_k) = -\nabla \cdot \mathbf{j}_k + \dot{\omega}_k \, , \tag{18d}$$

**Is the dataset self-contained, or does it link to or otherwise rely on external resources (e.g., websites, tweets, other datasets)?** If it links to or relies on external resources, a) are there guarantees that they will exist, and remain constant, over time; b) are there official archival versions of the complete dataset (i.e., including the external resources as they existed at the time the dataset was created); c) are there any restrictions (e.g., licenses, fees) associated with any of the external resources that might apply to a future user? Please provide descriptions of all external resources and any restrictions associated with them, as well as links or other access points, as appropriate.
BLASTNet is self-contained.

**Does the dataset contain data that might be considered confidential (e.g., data that is protected by legal privilege or by doctor-patient confidentiality, data that includes the content of individuals' non-public communications)?** If so, please provide a description.
No.

**Does the dataset contain data that, if viewed directly, might be offensive, insulting, threatening, or might otherwise cause anxiety?** If so, please describe why.
No.

**Does the dataset relate to people?** If not, you may skip the remaining questions in this section.
No.

**Does the dataset identify any subpopulations (e.g., by age, gender)?** If so, please describe how these subpopulations are identified and provide a description of their respective distributions within the dataset.
No.

**Is it possible to identify individuals (i.e., one or more natural persons), either directly or indirectly (i.e., in combination with other data) from the dataset?** If so, please describe how.
No.

**Does the dataset contain data that might be considered sensitive in any way (e.g., data that reveals racial or ethnic origins, sexual orientations, religious beliefs, political opinions or union memberships, or locations; financial or health data; biometric or genetic data; forms of government identification, such as social security numbers; criminal history)?** If so, please provide a description.
No.

**Any other comments?**
No.

## COLLECTION

**How was the data associated with each instance acquired?** Was the data directly observable (e.g., raw text, movie ratings), reported by subjects (e.g., survey responses), or indirectly inferred/derived from other data (e.g., part-of-speech tags, model-based guesses for age or language)? If data was reported by subjects or indirectly inferred/derived from other data, was the data validated/verified? If so, please describe how.

BLASTNet represents a collection of multiple DNS datasets, with analysis previously published in [27–29, 67, 20, 69, 70]. However, the data was not publicly available until the release of this work.

**Over what timeframe was the data collected?** Does this timeframe match the creation timeframe of the data associated with the instances (e.g., recent crawl of old news articles)? If not, please describe the timeframe in which the data associated with the instances was created. Finally, list when the dataset was first published.

This dataset was collected from published work between years 2019 and 2022.

**What mechanisms or procedures were used to collect the data (e.g., hardware apparatus or sensor, manual human curation, software program, software API)?** How were these mechanisms or procedures validated?

The DNS cases are performed using well-established numerical solvers [29, 64–67] in this research community, with analysis of the flowfields typically validated via peer-review.

**What was the resource cost of collecting the data?** (e.g. what were the required computational resources, and the associated financial costs, and energy consumption - estimate the carbon footprint. See Strubell *et al.*[103] for approaches in this area.)

Compute cost of all DNS is summarized in Table 17. However, we note that BLASTNet curates previously-unreleased **already-generated** data. Thus, the additional compute cost in developing this dataset is insignificant.

Table 17: Computing cost for dataset collection.

| Dataset | Cost [CPU-hr] |
|---|---|
| Non-reacting HIT [20] | $0.029 \times 10^6$ |
| Reacting forced HIT [67] | $\sim 10^6$ |
| Reacting jet flows (BFER case) [29] | $0.54 \times 10^6$ |
| Reacting jet flows (COFFEE case) [29] | $0.64 \times 10^6$ |
| Non-reacting transcritical Channel Flow [69] | $0.384 \times 10^6$ |
| Reacting channel flow [28] | $0.5 \times 10^6$ |
| Partially premixed slot burner [27] | $2.5 \times 10^6$ |
| Freely Propagating Flame [70] | $12 \times 10^6$ |

**If the dataset is a sample from a larger set, what was the sampling strategy (e.g., deterministic, probabilistic with specific sampling probabilities)?**

There is no sampling involved in BLASTNet 2.0, as it contains all potential instances within three-dimensional spatial domains.

**Who was involved in the data collection process (e.g., students, crowdworkers, contractors) and how were they compensated (e.g., how much were crowdworkers paid)?**

Neither crowdworkers nor contractors were used in data collection. The BLASTNet datasets were created and processed by the authors of this work for no additional payment outside of typical salary and stipend.

**Were any ethical review processes conducted (e.g., by an institutional review board)?** If so, please provide a description of these review processes, including the outcomes, as well as a link or other access point to any supporting documentation.

No.

**Does the dataset relate to people?** If not, you may skip the remainder of the questions in this section.
No.

**Did you collect the data from the individuals in question directly, or obtain it via third parties or other sources (e.g., websites)?**
No.

**Were the individuals in question notified about the data collection?** If so, please describe (or show with screenshots or other information) how notice was provided, and provide a link or other access point to, or otherwise reproduce, the exact language of the notification itself.
N/A.

**Did the individuals in question consent to the collection and use of their data?** If so, please describe (or show with screenshots or other information) how consent was requested and provided, and provide a link or other access point to, or otherwise reproduce, the exact language to which the individuals consented.
N/A.

**If consent was obtained, were the consenting individuals provided with a mechanism to revoke their consent in the future or for certain uses?** If so, please provide a description, as well as a link or other access point to the mechanism (if appropriate)
N/A.

**Has an analysis of the potential impact of the dataset and its use on data subjects (e.g., a data protection impact analysis)been conducted?** If so, please provide a description of this analysis, including the outcomes, as well as a link or other access point to any supporting documentation.
N/A.

**Any other comments?**
No.

---

## PREPROCESSING / CLEANING / LABELING

**Was any preprocessing/cleaning/labeling of the data done(e.g.,discretization or bucketing, tokenization, part-of-speech tagging, SIFT feature extraction, removal of instances, processing of missing values)?** If so, please provide a description. If not, you may skip the remainder of the questions in this section.
Yes. The data was generated from different numerical solvers, initially exists in a range of formats (.vtk, .vtu, .tec, and .dat) that are not readily formatted for training ML models. Thus, all data are processed into a consistent format – little-endian single-precision binaries that can be read with np.fromfile/np.memmap. The choice of this data format enables high I/O speed in loading arrays. We provide .json files that store additional information on configurations, chemical mechanisms and transport properties.

**Was the "raw" data saved in addition to the preprocessed/cleaned/labeled data (e.g., to support unanticipated future uses)?** If so, please provide a link or other access point to the "raw" data.
No, raw data are not available for public. This is because flow physics datasets are typically stored in a variety of double-precision formats that cannot fit into open repositories, introducing challenges, along with auxiliary derived quantities. However, the shared data are single-precision versions of the original data, with additional files shared in metadata for obtaining the auxiliary quantities.

**Is the software used to preprocess/clean/label the instances available?** If so, please provide a link or other access point.

`Python` is used to preprocess the raw data in a convenient format for ML applications.

**Any other comments?**
No.

## USES

**Has the dataset been used for any tasks already?** If so, please provide a description.
Yes, part of the dataset was introduced previously as BLASTNET 1.0 in [8, 9], and was used for semantic segmentation and reaction closure modeling.

**Is there a repository that links to any or all papers or systems that use the dataset?** If so, please provide a link or other access point.
No.

**What (other) tasks could the dataset be used for?**
BLASTNet can be used for multiple tasks such as subgrid-scale modeling [37], spatial and temporal prediction [49]. Specifically, this dataset is useful for developing closure models [38] in computational fluid dynamics (in both reacting and non-reacting flows) to capture the interactions between resolved and unresolved turbulence scales and generate closure terms or correction factors to add to the existing turbulence models. Lastly, BLASTNet can also be employed for solving inverse problems [18] via ML methods.

**Is there anything about the composition of the dataset or the way it was collected and preprocessed/cleaned/labeled that might impact future uses?** For example, is there anything that a future user might need to know to avoid uses that could result in unfair treatment of individuals or groups (e.g., stereotyping, quality of service issues) or other undesirable harms (e.g., financial harms, legal risks) If so, please provide a description. Is there anything a future user could do to mitigate these undesirable harms?
No.

**Are there tasks for which the dataset should not be used?** If so, please provide a description.
BLASTNet 2.0 data contains data from reacting or non-reacting DNS configurations, which covers a specific range of flow physics conditions. Consequently, this dataset cannot be for unrepresented conditions such as rarefied, multi-phase, micro-fluid, Non-Newtonian, or magneto-hydrodynamic flows.

**Any other comments?**
No.

## DISTRIBUTION

**Will the dataset be distributed to third parties outside of the entity (e.g., company, institution, organization) on behalf of which the dataset was created?** If so, please provide a description.
Yes, the dataset is publicly available. To circumvent Kaggle storage constraints, we partition our data into a network of <100 GB subsets, with each subset containing a separate simulation configuration. This partitioned data can then be uploaded as separate datasets on Kaggle, with links consolidated in `https://blastnet.github.io`.

**How will the dataset will be distributed (e.g., tarball on website, API, GitHub)?** Does the dataset have a digital object identifier (DOI)?

The data is available through multiple Kaggle repositories, with links consolidated at `https://blastnet.github.io/datasets`. Each dataset shares a common DOI at `https://doi.org/10.5281/zenodo.7242864`.

**When will the dataset be distributed?**
The BLASTNet 1.0 dataset is available online and can be accessed and downloaded via Kaggle. We are planning to release a fully-tested version of BLASTNet 2.0 prior to NeurIPS 2023, depending on the outcome of the review process. However, a publicly-available version that can be peer-reviewed is released on June 14 2023.

**Will the dataset be distributed under a copyright or other intellectual property (IP) license, and/or under applicable terms of use (ToU)?** If so, please describe this license and/or ToU, and provide a link or other access point to, or otherwise reproduce, any relevant licensing terms or ToU, as well as any fees associated with these restrictions.
Licensed via CC BY-SA NC 4.0.

**Have any third parties imposed IP-based or other restrictions on the data associated with the instances?** If so, please describe these restrictions, and provide a link or other access point to, or otherwise reproduce, any relevant licensing terms, as well as any fees associated with these restrictions.
No.

**Do any export controls or other regulatory restrictions apply to the dataset or to individual instances?** If so, please describe these restrictions, and provide a link or other access point to, or otherwise reproduce, any supporting documentation.
No.

**Any other comments?**
No.

---

## MAINTENANCE

**Who is supporting/hosting/maintaining the dataset?**
BLASTNet is hosted on Kaggle. Stanford Laboratory of Fluids in Complex Systems (`https://web.stanford.edu/group/ihmegroup/cgi-bin/MatthiasIhme`) will maintain this dataset, along with the support of contributors from Sandia National Laboratory, University of Melbourne, Polytechnique Montr'eal, and University of Connecticut.

**How can the owner/curator/manager of the dataset be contacted (e.g., email address)?**
The BLASTNet team can be reached at `blast.net.data@gmail.com`. Users are encouraged to submit any issues/inquiries to our GitHub page `https://github.com/blastnet/blastnet.github.io/issues`. Lastly, Wai Tong Chung (`wtchung@stanford.edu`) and Matthias Ihme (`mihme@stanford.edu`) can be reached for further inquiries and collaborations.

**Is there an erratum?** If so, please provide a link or other access point.
Yes. A detailed issue tracker is provided in `https://github.com/blastnet/blastnet.github.io/issues`.

**Will the dataset be updated (e.g., to correct labeling errors, add new instances, delete instances)?** If so, please describe how often, by whom, and how updates will be communicated to users (e.g., mailing list, GitHub)?
Yes, the team will release minor updates to the dataset in case of any identified errors. These updates will be communicated via the landing page `https://blastnet.github.io` and GitHub

```
https://github.com/blastnet/blastnet.github.io/issues.
```

**If the dataset relates to people, are there applicable limits on the retention of the data associated with the instances (e.g., were individuals in question told that their data would be retained for a fixed period of time and then deleted)?** If so, please describe these limits and explain how they will be enforced.
N/A.

**Will older versions of the dataset continue to be supported/hosted/maintained?** If so, please describe how. If not, please describe how its obsolescence will be communicated to users.
Yes. BLASTNet 2.0 builds on data from BLASTNet 1.0. Thus, all previous data will be preserved with each version update. In addition, version histories and logs from individual Kaggle repositories will be used to inform users of changes. Changes will also be communicated to users via the landing page `https://blastnet.github.io`.

**If others want to extend/augment/build on/contribute to the dataset, is there a mechanism for them to do so?** If so, please provide a description. Will these contributions be validated/verified? If so, please describe how. If not, why not? Is there a process for communicating/distributing these contributions to other users? If so, please provide a description.
Yes. We plan to accept contributions via `https://blastnet.github.io/contribute`. Any contributions will be checked for accuracy, robustness, reliability, and formats prior to new release. New contributions will be tracked via our version history.

**Any other comments?**
No.

**H.2 Momentum 128 3D SR**

## MOTIVATION

**For what purpose was the dataset created?** Was there a specific task in mind? Was there a specific gap that needed to be filled? Please provide a description.

Momentum128 3D SR consists of BLASTNet 2.0 data – processed for 3D super-resolution (SR) of density and velocity at $8, 16$ and $32\times$ SR, while mitigating constraints in memory and grid properties. The single largest sample in BLASTNet is 92 GB (with 1.3B voxels and 15 channels), which is too large to fit in a typical GPU memory. In Momentum128 3D SR, the BLASTNet 2.0 data is downsampled to $128^3$ sub-volumes of density and velocity fields to maintain a low memory footprint. This dataset is useful for training/evaluating 3D SR models, as well as developing closure models [38] in computational fluid dynamics (in both reacting and non-reacting flows) to capture the interactions between resolved and unresolved turbulence scales.

**Who created this dataset (e.g., which team, research group) and on behalf of which entity (e.g., company, institution, organization)?**

Momentum128 3D SR is created by Wai Tong Chung. The raw datasets used to develop Momentum128 3D SR are contributed by the following researchers:

1. Wai Tong Chung, Davy Brouzet and Matthias Ihme at Stanford University.

2. Jacqueline H. Chen and Ki Sung Jung at Sandia National Laboratory.

3. Mohsen Talei at University of Melbourne.

4. Bruno Savard at Polytechnique Montréal.

5. Alexei Poludnenko at University of Connecticut.

Bassem Akoush, Pushan Sharma and Alex Tamkin at Stanford University contributed in administering, improving, maintaining, and documenting the dataset curation process.

**What support was needed to make this dataset?** (e.g.who funded the creation of the dataset? If there is an associated grant, provide the name of the grantor and the grant name and number, or if it was supported by a company or government agency, give those details.)

This work is funded by:

1. The U.S. Department of Energy National Nuclear Security Administration, under award No. DE-NA0003968.

2. The NASA Early Stage Innovation Program with award No. 80NSSC22K0257.

3. The Department of Energy Office of Energy Efficiency Renewable Energy (EERE) with award No. DE-EE0008875.

4. Wai Tong Chung received partial financial support from the Stanford Institute for Human-centered Artificial Intelligence Graduate Fellowship.

This research used resources of the National Energy Research Scientific Computing Center (NERSC), a U.S. Department of Energy Office of Science User Facility located at Lawrence Berkeley National Laboratory, operated under Contract No. DE-AC02-05CH11231 using NERSC award ERCAP0021046.

**Any other comments?**
No.

## COMPOSITION

**What do the instances that comprise the dataset represent (e.g., documents, photos, people, countries)?** Are there multiple types of instances (e.g., movies, users, and ratings; people and interactions between them; nodes and edges)? Please provide a description.

Each instance consists of 3D domain of velocity, temperature, pressure, density and mass fractions of chemical species. These variables are saved as flat arrays in separate `.dat` binary files, which can be loaded and reshaped to construct the 3D volumes. To enable high I/O speed in loading arrays, the data has consistent little-endian single-precision binaries that can be read with `np.fromfile`/`np.memmap`.

**How many instances are there in total (of each type, if appropriate)?**
Momentum128 3D SR contains a total of 2000 sub-volumes of density and velocity samples from a diverse collection of 27 DNS configuration.

**Does the dataset contain all possible instances or is it a sample (not necessarily random) of instances from a larger set?** If the dataset is a sample, then what is the larger set? Is the sample representative of the larger set (e.g., geographic coverage)? If so, please describe how this representativeness was validated/verified. If it is not representative of the larger set, please describe why not (e.g., to cover a more diverse range of instances, because instances were withheld or unavailable).
Momentum128 3D SR is a subset of BLASTNet 2.0. (`https://blastnet.github.io`), downsampled to preserve statistical characteristics of the larger dataset. This is done via maintaining a good proportion of similar clusters (obtained via k-means with four statistical moments of three velocity components used as inputs).

**What data does each instance consist of?** "Raw" data (e.g., unprocessed text or images) or features? In either case, please provide a description.
Labels consists of 3D flowfields of DNS-fidelity velocity and density (of size $128^3$). Features consist of corresponding 8,16, and $32\times$ Favre-filtered flowfields from the labels.

**Is there a label or target associated with each instance?** If so, please provide a description.
Yes. The data is fully labeled for 8,16, and $32\times$ SR in turbulent flow applications.

**Is any information missing from individual instances?** If so, please provide a description, explaining why this information is missing (e.g., because it was unavailable). This does not include intentionally removed information, but might include, e.g., redacted text.
No.

**Are relationships between individual instances made explicit (e.g., users' movie ratings, social network links)?** If so, please describe how these relationships are made explicit.
Yes. Features and labels from the same DNS configuration and spatial location share a unique hash identifier.

**Are there recommended data splits (e.g., training, development/validation, testing)?** If so, please provide a description of these splits, explaining the rationale behind them.
Yes. We provide train, validation, and test splits in an 80:10:10 ratio. Two additional split sets from unseen DNS configurations are recommended for evaluating out-of-distribution behavior and data normalization.

**Are there any errors, sources of noise, or redundancies in the dataset?** If so, please provide a description.
Yes. Well-established high-order numerical solvers [29, 64–67] have been employed, with spatial discretization schemes ranging from 2nd- to 8th-order accuracy and time advancement accuracy ranging from 2nd- to 4th-order. Thus, this data is still subject to small numerical errors.

**Is the dataset self-contained, or does it link to or otherwise rely on external resources (e.g., websites, tweets, other datasets)?** If it links to or relies on external resources, a) are there guarantees that they will exist, and remain constant, over time; b) are there official archival versions of the complete dataset (i.e., including the external resources as they existed at the time the dataset

was created); c) are there any restrictions (e.g., licenses, fees) associated with any of the external resources that might apply to a future user? Please provide descriptions of all external resources and any restrictions associated with them, as well as links or other access points, as appropriate.
Momentum128 3D SR is self-contained.

**Does the dataset contain data that might be considered confidential (e.g., data that is protected by legal privilege or by doctor-patient confidentiality, data that includes the content of individuals' non-public communications)?** If so, please provide a description.
No.

**Does the dataset contain data that, if viewed directly, might be offensive, insulting, threatening, or might otherwise cause anxiety?** If so, please describe why.
No.

**Does the dataset relate to people?** If not, you may skip the remaining questions in this section.
No.

**Does the dataset identify any subpopulations (e.g., by age, gender)?** If so, please describe how these subpopulations are identified and provide a description of their respective distributions within the dataset.
No.

**Is it possible to identify individuals (i.e., one or more natural persons), either directly or indirectly (i.e., in combination with other data) from the dataset?** If so, please describe how.
No.

**Does the dataset contain data that might be considered sensitive in any way (e.g., data that reveals racial or ethnic origins, sexual orientations, religious beliefs, political opinions or union memberships, or locations; financial or health data; biometric or genetic data; forms of government identification, such as social security numbers; criminal history)?** If so, please provide a description.
No.

**Any other comments?**
No.

---

## COLLECTION

---

**How was the data associated with each instance acquired?** Was the data directly observable (e.g., raw text, movie ratings), reported by subjects (e.g., survey responses), or indirectly inferred/derived from other data (e.g., part-of-speech tags, model-based guesses for age or language)? If data was reported by subjects or indirectly inferred/derived from other data, was the data validated/verified? If so, please describe how.
Momentum128 3D SR is sampled from BLASTNet 2.0, which was in turn validated through analysis in previous publications [27, 29, 67, 20, 70].

**Over what timeframe was the data collected?** Does this timeframe match the creation timeframe of the data associated with the instances (e.g., recent crawl of old news articles)? If not, please describe the timeframe in which the data associated with the instances was created. Finally, list when the dataset was first published.
This dataset was collected from work done between years 2019 and 2022.

**What mechanisms or procedures were used to collect the data (e.g., hardware apparatus or sensor, manual human curation, software program, software API)?** How were these mechanisms or procedures validated?
The DNS cases are performed using well-established numerical solvers [29, 64–67] in this research community.

**What was the resource cost of collecting the data?** (e.g. what were the required computational resources, and the associated financial costs, and energy consumption - estimate the carbon footprint. See Strubell *et al.*[103] for approaches in this area.)
Compute cost of all employed DNS is summarized in Table 17. However, we note that this data curates **already-generated** data. Thus, the additional compute cost in developing this dataset is insignificant.

Table 18: Computing cost for dataset collection.

| Dataset | Cost [CPU-hr] |
|---|---|
| Non-reacting HIT [20] | $0.029 \times 10^6$ |
| Reacting forced HIT [67] | $\sim 10^6$ |
| Reacting jet flows (BFER case) [29] | $0.54 \times 10^6$ |
| Reacting jet flows (COFFEE case) [29] | $0.64 \times 10^6$ |
| Partially premixed slot burner [27] | $2.5 \times 10^6$ |
| Freely Propagating Flame [70] | $12 \times 10^6$ |

**If the dataset is a sample from a larger set, what was the sampling strategy (e.g., deterministic, probabilistic with specific sampling probabilities)?**
Momentum128 3D SR is created by sampling $128^3$ sub-volumes of density and velocity from the uniform regions of all BLASTNet 2.0 datasets. This results in 12750 sub-volume samples, out of which 2000 sub-volumes are sampled to create Momentum128 3D SR. To ensure statistical representativeness of the original dataset, these 2000 samples are collected in these steps:

1. Extracting four statistical moments of the three velocity-components of each of the 12750 sub-volumes.

2. Applying k-means clustering with the elbow method to partition the sub-volumes in 18 clusters.

3. Sampling 2000 sub-volumes with balanced proportions from each cluster to generate the labels.

4. Favre-filtering the labels to generate the features.

**Who was involved in the data collection process (e.g., students, crowdworkers, contractors) and how were they compensated (e.g., how much were crowdworkers paid)?**
Neither crowdworkers nor contractors were used in data collection. The BLASTNet datasets were created and processed by the authors of this work for no additional payment outside of typical salary and stipend.

**Were any ethical review processes conducted (e.g., by an institutional review board)?** If so, please provide a description of these review processes, including the outcomes, as well as a link or other access point to any supporting documentation.
No.

**Does the dataset relate to people?** If not, you may skip the remainder of the questions in this section.
No.

**Did you collect the data from the individuals in question directly, or obtain it via third parties or other sources (e.g., websites)?**

No.

**Were the individuals in question notified about the data collection?** If so, please describe (or show with screenshots or other information) how notice was provided, and provide a link or other access point to, or otherwise reproduce, the exact language of the notification itself.
N/A.

**Did the individuals in question consent to the collection and use of their data?** If so, please describe (or show with screenshots or other information) how consent was requested and provided, and provide a link or other access point to, or otherwise reproduce, the exact language to which the individuals consented.
N/A.

**If consent was obtained, were the consenting individuals provided with a mechanism to revoke their consent in the future or for certain uses?** If so, please provide a description, as well as a link or other access point to the mechanism (if appropriate)
N/A.

**Has an analysis of the potential impact of the dataset and its use on data subjects (e.g., a data protection impact analysis)been conducted?** If so, please provide a description of this analysis, including the outcomes, as well as a link or other access point to any supporting documentation.
N/A.

**Any other comments?**
No.

## PREPROCESSING / CLEANING / LABELING

**Was any preprocessing/cleaning/labeling of the data done(e.g.,discretization or bucketing, tokenization, part-of-speech tagging, SIFT feature extraction, removal of instances, processing of missing values)?** If so, please provide a description. If not, you may skip the remainder of the questions in this section.
Momentum128 3D SR is created by sampling $128^3$ sub-volumes of density and velocity from the uniform regions of all BLASTNet 2.0 datasets. This results in 12750 sub-volume samples, out of which 2000 sub-volumes are sampled to create Momentum128 3D SR. To ensure statistical representativeness of the original dataset, these 2000 samples are collected in these steps:

1. Extracting four statistical moments of the three velocity-components of each of the 12750 sub-volumes.

2. Applying k-means clustering with the elbow method to partition the sub-volumes in 18 clusters.

3. Sampling 2000 sub-volumes with balanced proportions from each cluster to generate the labels.

4. Favre-filtering the labels to generate the features.

**Was the "raw" data saved in addition to the preprocessed/cleaned/labeled data (e.g., to support unanticipated future uses)?** If so, please provide a link or other access point to the "raw" data.
Yes. BLASTNet 2.0 is publicly available via `https://blastnet.github.io`.

**Is the software used to preprocess/clean/label the instances available?** If so, please provide a link or other access point.
`Python` is used to sample from BLASTNet 2.0. Code for Favre-filtering is attached to `https://github.com/blastnet/blastnet2_sr_benchmark`.

**Any other comments?**
No.

**USES**

**Has the dataset been used for any tasks already?** If so, please provide a description.
No.

**Is there a repository that links to any or all papers or systems that use the dataset?** If so, please provide a link or other access point.
No.

**What (other) tasks could the dataset be used for?**
The Momentum128 3D SR dataset was specifically sub-sampled for the purpose of super-resolution and turbulence closure modeling. This dataset can be used for closure modeling via direct regression of the subgrid-scale terms. This high-fidelity data can also be employed for generative and deterministic reconstruction.

**Is there anything about the composition of the dataset or the way it was collected and preprocessed/cleaned/labeled that might impact future uses?** For example, is there anything that a future user might need to know to avoid uses that could result in unfair treatment of individuals or groups (e.g., stereotyping, quality of service issues) or other undesirable harms (e.g., financial harms, legal risks) If so, please provide a description. Is there anything a future user could do to mitigate these undesirable harms?
No.

**Are there tasks for which the dataset should not be used?** If so, please provide a description.
This dataset is specifically generated for SR and turbulent closure modeling purpose from BLASTNet 2.0. Thus, it should not be used for other tasks.

**Any other comments?**
No.

**DISTRIBUTION**

**Will the dataset be distributed to third parties outside of the entity (e.g., company, institution, organization) on behalf of which the dataset was created?** If so, please provide a description.
Yes, the dataset is publicly available at `https://www.kaggle.com/datasets/waitongchung/blastnet-momentum-3d-sr-dataset`.

**How will the dataset will be distributed (e.g., tarball on website, API, GitHub)?** Does the dataset have a digital object identifier (DOI)?
The data is available through Kaggle at `https://www.kaggle.com/datasets/waitongchung/blastnet-momentum-3d-sr-dataset`. This dataset shares a common DOI with BLASTNet 2.0, at `https://doi.org/10.5281/zenodo.7242864`.

**When will the dataset be distributed?**
We are planning to release a fully-tested version of Momentum128 3D SR prior to NeurIPS 2023, depending on the outcome of the review process. However, a publicly-available version that can be peer-reviewed is released on June 14 2023.

**Will the dataset be distributed under a copyright or other intellectual property (IP) license, and/or under applicable terms of use (ToU)?** If so, please describe this license and/or ToU, and

provide a link or other access point to, or otherwise reproduce, any relevant licensing terms or ToU, as well as any fees associated with these restrictions.
Licensed via CC BY-SA NC 4.0.

**Have any third parties imposed IP-based or other restrictions on the data associated with the instances?** If so, please describe these restrictions, and provide a link or other access point to, or otherwise reproduce, any relevant licensing terms, as well as any fees associated with these restrictions.
No.

**Do any export controls or other regulatory restrictions apply to the dataset or to individual instances?** If so, please describe these restrictions, and provide a link or other access point to, or otherwise reproduce, any supporting documentation.
No.

**Any other comments?**
No.

## MAINTENANCE

**Who is supporting/hosting/maintaining the dataset?**
Momentum128 3D SR is hosted on Kaggle. Stanford Laboratory of Fluids in Complex Systems (`https://web.stanford.edu/group/ihmegroup/cgi-bin/MatthiasIhme`) will maintain this dataset, along with the support of contributors from Sandia National Laboratory, University of Melbourne, Polytechnique Montreal, and University of Connecticut.

**How can the owner/curator/manager of the dataset be contacted (e.g., email address)?**
The BLASTNet team can be reached at `blast.net.data@gmail.com`. The users are encouraged to submit any inquiries to GitHub page `https://github.com/blastnet/blastnet.github.io/issues`. Lastly, Wai Tong Chung (`wtchung@stanford.edu`) and Matthias Ihme (`mihme@stanford.edu`) can be reached for further inquiries and collaborations.

**Is there an erratum?** If so, please provide a link or other access point.
Yes. A detailed issue tracker is provided in `https://github.com/blastnet/blastnet.github.io/issues`.

**Will the dataset be updated (e.g., to correct labeling errors, add new instances, delete instances)?** If so, please describe how often, by whom, and how updates will be communicated to users (e.g., mailing list, GitHub)?
Yes, the team will release minor updates to the dataset in case of any identified errors. Theses updates will be announced in the landing page `https://blastnet.github.io` and GitHub `https://github.com/blastnet/blastnet.github.io/issues`.

**If the dataset relates to people, are there applicable limits on the retention of the data associated with the instances (e.g., were individuals in question told that their data would be retained for a fixed period of time and then deleted)?** If so, please describe these limits and explain how they will be enforced.
N/A.

**Will older versions of the dataset continue to be supported/hosted/maintained?** If so, please describe how. If not, please describe how its obsolescence will be communicated to users.
Yes. Version histories and logs from the Kaggle repository will be used to inform users of changes.

**If others want to extend/augment/build on/contribute to the dataset, is there a mechanism for them to do so?** If so, please provide a description. Will these contributions be validated/verified? If so, please describe how. If not, why not? Is there a process for communicating/distributing these contributions to other users? If so, please provide a description.

Yes. Kaggle hosts a comments section that enables user feedback and suggestions. In addition, contact information of the data curators are also provided in `https://blastnet.github.io`.

**Any other comments?**

No.

