# OpenReview forum: "Turbulence in Focus: Benchmarking Scaling Behavior of 3D Volumetric Super-Resolution with BLASTNet 2.0 Data"
_NeurIPS.cc/2023/Track/Datasets_and_Benchmarks — NeurIPS 2023 Datasets and Benchmarks Poster_

### Official Review · Reviewer_o9yf · 2023-07-05
**Turbulence in Focus: Benchmarking Scaling Behavior of 3D Volumetric Super-Resolution with BLASTNet 2.0 Data.**

**Rating:** 7
**Confidence:** 5
**Correctness:** The evaluations are appropriate and p…
**Clarity:** The paper is well written and organized.

**Strengths:**

The previous studies on flow physics simulation datasets have predominantly focused on Reynolds averaged Navier Stokes simulations (RANS) and large eddy simulations (LES) using two-dimensional model. However, it is important to note that two-dimensional flow simulations have limitations in capturing turbulence due to the inability to resolve vortex and eddy structures inherent in three-dimensional models.
In contrast, BLASTNet 2.0 addresses this limitation by employing a three-dimensional model for simulating compressible turbulent flows. This choice enables BLASTNet 2.0 to effectively capture and represent the complex structures associated with turbulence. By utilizing a three-dimensional approach, BLASTNet 2.0 enhances the fidelity of the simulations and provides a more accurate representation of turbulent flow phenomena.


**Additional Feedback:**

N/A

**Documentation:**

The authors included documentation and intended uses with datasets. In the documentation, there is sufficient detail on datasets.

**Ethics:**

I do not suspect any ethical concerns.

**Limitations:**

The authors adequately addressed the limitations and potential negative societal impact of their work.

**Opportunities For Improvement:**

In table 1, the range of PDEs for BLASTNet 2.0 is stated as 5 to 27. However, the model introduced by Huang has 7 PDEs, even though it covers two-dimensional simulations. Is it reasonable to assume that the higher number of PDEs in BLASTNet 2.0 is due to its three-dimensional nature and the inclusion of chemical species equations? The authors are advised to clarify the exact number and nature of the PDEs in BLASTNet 2.0 to provide a more comprehensive understanding.
On page 4, the authors mention that the direct numerical simulations (DNS) utilized in this study employed spatial discretization schemes ranging from 2nd- to 8th-order accuracy. However, it is important to note that 2nd and 8th order accuracy schemes have significant differences in their ability to resolve turbulence. The authors should provide further clarification on how the range of accuracy schemes used in the DNS affects the resolution of turbulence.
Regarding Figure 3, it is suggested to consider including the contour of absolute errors in addition to the existing contours of errors show in Figure 4. This addition would provide a more comprehensive visual representation of the absolute errors in the model’s prediction.
On page 9, the authors mention that the DNS data used in the study originated from proprietary-licensed numerical solvers. It would be helpful if the authors specify the type or name of the numerical solvers employed in this study. Additionally, providing a reference or citation to the numerical solvers used would enhance the reproducibility and credibility of the research.


**Relation To Prior Work:**

The authors addressed what this study differs from previous contributions.

**Summary And Contributions:**

This paper introduces BLASTNet 2.0, which incorporates 744 full-domain samples from 34 direct numerical simulations to create a dataset for 3D compressible turbulent reacting flow and non-reacting flow. The authors concluded that the super-resolution (SR) models in BLASTNet 2.0 exhibit  strong performance in terms of structural similarity image measure (SSIM) based metrics and effectively capture intricate turbulent structures at 8× SR. The availability of BLASTNet 2.0 serves as a valuable resource for evaluating models related to scientific and engineering turbulent flows.

---

> ### Author Response · Authors · 2023-08-20
> **Our Response**
>
> We are grateful for the positive review from the reviewer, especially on the value of BLASTNet 2.0, as well as the clarity and good documentation of this work. Regarding the opportunities for improvement:
>
> 1. **Clarification on the number of PDEs:**
>
> We have provided a footnote to Table 1 to provide more information on this. The high number of  PDEs in BLASTNet 2.0 are indeed due to both the 3D space and the number of chemical species.
>
> 2. **Numerics and Turbulence:**
>
> We now include a small discussion on effects of numerics on the grid requirement in Section 3.1.
>
> 3. **Additional Plot of Absolute Errors:**
>
> We have added this plot of absolute errors to Appendix G.1 in the supplementary material.
>
> 4. **Solver Information:**
>
> In section 6, we now specified a reference to the documentation of solver and configuration within Appendix D.2 in the supplementary material. References to solvers are also included in Section 3.1.

---

### Official Review · Reviewer_5NmK · 2023-07-20
**Turbulence in Focus: Benchmarking Scaling Behavior of 3D Volumetric Super-Resolution with BLASTNet 2.0 Data**

**Rating:** 7
**Confidence:** 4
**Clarity:** Yes

**Strengths:**

- This paper curates a diverse public 3D compressible turbulent flow dataset
- This paper benchmarks the performance and cost of 3D versions of three popular SR models with the proposed dataset.
- This paper shows that SR model performance can scale with the logarithm of model size and cost.
- This paper demonstrates the persisting benefits of a popular gradient-based physics-informed loss with increasing model size.
- The proposed dataset comes with a well-established website and tutorial, which is convenient for researchers to use.

**Additional Feedback:**

See "Opportunities For Improvement"

**Correctness:**

Dataset is constructed in a solid way. But the benchmark need improvement and further validation of the usefulness of the task. See "Opportunities For Improvement" for more detail.

**Documentation:**

Documentation is good.

**Limitations:**

Limitation is well discussed.

**Opportunities For Improvement:**

- This paper benchmarked super-resolution tasks on their published dataset. But I think a more interesting task is time series prediction using neural PDE solvers, because the prediction task is of great importance and more practically useful in AI for PDE. So I recommend authors benchmark the following models [1][2][3] to benchmark the result of the prediction task, these models are all neural operator models used to predict PDE solutions.
- Super-resolution task seems to use downsampled fluid fields as input and predict high-resolution output. I think this is not a practically useful setting. In practice, if we perform low-resolution simulations to save cost, then the simulation result is not the downsampled solution from high-resolution simulations due to the numerical error in low-resolution simulations. If the low-resolution input is obtained from sensor observations of the real scene, then the fluid field should be sparse instead of densely covering the whole scene. Could authors elaborate more on how super-resolution tasks are practically useful?

[1] Li, Zongyi, et al. "Fourier neural operator for parametric partial differential equations." arXiv preprint arXiv:2010.08895 (2020).\
[2] Tran, Alasdair, et al. "Factorized fourier neural operators." arXiv preprint arXiv:2111.13802 (2021).\
[3] Helwig, Jacob, et al. "Group Equivariant Fourier Neural Operators for Partial Differential Equations." arXiv preprint arXiv:2306.05697 (2023).

**Relation To Prior Work:**

Yes

**Summary And Contributions:**

This paper proposes a BLASTNet 2.0, a 2.2 TB network-of-datasets containing 744 full domain samples from 34 high-fidelity direct numerical simulations. The aim is to address the current limited availability of 3D high-fidelity reacting and non-reacting compressible turbulent flow simulation data. In addition, this paper benchmarks three deep learning models for 3D super-resolution. They further perform neural scaling analysis on these models to examine the performance of different architectures, and to assess the benefits of a popular gradient-based physics-informed loss function across different model sizes.

---

> ### Author Response · Authors · 2023-08-20
> **Our Response**
>
> We appreciate the positive  comments on dataset diversity and accessibility, use of a physics-based model, our neural scaling analysis, as well as the clarity and good documentation of this work. Regarding the opportunities for improvement:
>
>
> 1. **Benchmarking of Fourier Neural Operator (FNO):**
>
> We have new results in Section 5 to consider the inclusion of Convolutional FNO (Conv-FNO) models (with 9 model variations, 27 runs in total). In total, we benchmarked 2 types of AI for Science/PDE approaches (Conv-FNO and gradient-based loss), on top of 3 types of vanilla ResNet-based approaches with different architecture paradigms, which we believe is a good balance for readers interested in both PDE and vanilla ML approaches.
>
> 2. **On Time Series Prediction as a Benchmark Task:**
>
> We agree that time series prediction is an interesting task for PDE-linked problems. However, we believe that super-resolution is an equally important task within the application of turbulent flow physics, as elaborated below and  shown by numerous works [1,2,3,4,5] from the flow physics community in section 2 of the manuscript. We also believe that super-resolution is a task that is suitable for the broad range of ML disciplines that NeurIPS attracts. The findings reported in this paper on 3D volumetric super-resolution can provide detailed insights into a broad range of ML applications within 3D computer vision applications and scientific imaging techniques, especially given the gap in rigorous benchmarks in 3D volumetric super-resolution in the current literature.
> Due to the length of a NeurIPS article, it is not feasible to perform additional analysis on PDE time-stepping in addition to detailed discussion of our 2.2 TB dataset and detailed analysis of the 49 SR models (147 runs in total) already present in the manuscript. Such an important task deserves a future publication that could be based on BLASTNet data.
>
>
> 3. **Justification of Turbulent SR as a useful ML task:**
>
> We thank the reviewer for raising the need to provide more clarity on the utility of super-resolution in the context of flow physics, which we have now included within Section 2 in the revised manuscript.
> Specifically, in many practical engineering analyses, coarse-grained simulations with under-resolved grid sizes (known as implicit large-eddy simulations (LES)) are commonly used to bypass the extensive costs of fully-resolved direct numerical simulations [1,6,7,8,9,10]. We note that errors that arises from coarse-graining not only from numerical error; the dominant source of error arises from the missing physics (i.e., sub-grid scale stress in Eq (5) in the manuscript) due to the under-resolved grid [6,7,8]. Before the advent of ML-based solutions, algebraic expressions of turbulence models have traditionally represented the under-resolved physics [9,10]. Today, SR methods provide correction/closure models that can improve real-time computational fluid dynamics predictions [1,2].
> From an experimental standpoint, sparse sensors are not the only way to extract experimental information. Imaging techniques such as Schlieren [3], particle image velocimetry [4], and tomographic [5] techniques are often used to discover physical properties within turbulent flows.
>
> We must clarify that within this work, we are not simply downsampling the higher-resolution flowfields to low-resolution flowfields. Instead, we are employing filtering (specifically Favre-filtering) on the high-fidelity data, to generate finite-volume optimal LES [6]. which has a direct theoretical connection to coarse-grained simulations, as noted by literature [7,8] dating back to the 1990s from the flow physics community. This is now further detailed in Sections 2, 3.2, and 6.

---

> > ### Author Response · Authors · 2023-08-20
> > **References**
> >
> > 1. M. Bode, M. Gauding, Z. Lian, D. Denker, M. Davidovic, K. Kleinheinz, J. Jitsev, and H. Pitsch. Using physics-informed enhanced super-resolution generative adversarial networks for subfilter modeling in turbulent reactive flows. Proceedings of the Combustion Institute, 38(2):2617–2625, 2021.
> > 2. K. Fukami, K. Fukagata, and K. Taira. Super-resolution reconstruction of turbulent flows with machine learning. Journal of Fluid Mechanics, 870:106–120, 2019.
> > 3. Z. Wang, X. Li, L. Liu, X. Wu, P. Hao, X. Zhang, and F. He. Deep-learning-based superresolution reconstruction of high-speed imaging in fluids. Physics of Fluids, 34(3):037107, 2022.
> > 4. Z. Deng, C. He, Y. Liu, and K. C. Kim. Super-resolution reconstruction of turbulent velocity fields using a generative adversarial network-based artificial intelligence framework. Physics of Fluids, 31(12), 2019.
> > 5. N. Janssens, M. Huysmans, and R. Swennen. Computed tomography 3D super-resolution with generative adversarial neural networks: Implications on unsaturated and two-phase fluid flow. Materials, 13(6):1397, 2020.
> > 6. R. D. Moser, N. P. Malaya, H. Chang, P. S. Zandonade, P. Vedula, A. Bhattacharya, and A. Haselbacher. Theoretically based optimal large-eddy simulation. Physics of Fluids, 21 (10): 105104, 2009.
> > 7. M. Germano. Turbulence: The filtering approach. Journal of Fluid Mechanics, 238:325–336, 1992.
> > 8. R. D. Moser, S. W. Haering, and G. R. Yalla. Statistical properties of subgrid-scale turbulence models. Annual Review of Fluid Mechanics, 53:255–286, 2021.
> > 9. M. Germano, U. Piomelli, P. Moin, and W. H. Cabot. A dynamic subgrid-scale eddy viscosity model. Phys. Fluids A, 3(7):1760–1765, 1991.
> > 10. A. W. Vreman. An eddy-viscosity subgrid-scale model for turbulent shear flow: Algebraic theory and applications. Phys. Fluids, 16(10):3670–3681, 2004.

---

> > ### Comment · Reviewer_5NmK · 2023-08-24
> >
> > Thanks for the explanation, I agree that PDE time-stepping might be out of the scope of this paper. I want to discuss more on the generation of low-resolution data. Authors mentioned they use filtering to obtain LR data from HR data, which has a connection to coarse-grained simulation. Instead of using a filter method to mimic low-resolution simulations, why don't authors directly perform coarse-grained simulations to obtain LR data? Obviously, the latter can accurately reflect the use case in practice.
> >
> > Another question: based on the explanation from authors, when performing low-resolution simulation, we cannot simply use coarser grids and keep the same solver used in high-resolution simulation, is that right? My understanding is that some modified solvers need to be used in order to mitigate the error in low-resolution simulation?

---

> > > ### Author Response · Authors · 2023-08-24
> > > **2nd Response to Further Questions on Filtering and Coarse simulations**
> > >
> > > Thank you for the quick response and for the additional questions! We also appreciate your recognition of the scope of this work. On the new questions raised:
> > >
> > > 1. **On filtering:** We agree with the reviewer that having a corresponding low-resolution simulation with a high-resolution simulation as an input-output pair would be ideal. While this can be done for non-turbulent flows, turbulent flows are stochastic and chaotic systems [1] where it is not possible to obtain matching low-resolution (LR) and high-resolution (HR) pairs for every timestep. This is because the small changes (e.g. from the grid size) will lead to propagating differences as simulations advance in time, leading to widely different flowfields between HR and LR simulations – a typical symptom of chaos. Thus, we have to employ filtering to obtain canonical surrogates [2] for training inputs for SR problems. As also mentioned in the overall response, this approach has strong theoretical backing dating back to the 1990s [3,4], and is widely used [5,6] as training inputs for SR problems, and we have an extensive discussion on filtering and coarse simulations in Appendix E.1.6 in the Supplementary Material, as well as Section 2, 3.2, and 6 in the manuscript.
> > >
> > > 2. **On coarse-grid simulations:** A turbulent flow simulation that spatially resolves the lengthscales associated with the conversion of kinetic energy to heat is known as a direct numerical simulation (DNS) [7]. Coarser grids can be used to simulate the same configuration with the propagating errors (linked to chaotic systems) described above – is used as a lower-fidelity alternative if computational resources are not available for DNS [2,3,4]. These errors have been algebraically expressed via Eq. (5) in the manuscript. The reviewer is correct in pointing out that modifications can be made to the solver to ameliorate this error. This modification is known as turbulence modeling, and has traditionally been treated with correction terms expressed as different kinds of algebraic models [8,9]. However, we note that these modifications will not yield an exact corresponding match to DNS, but are still useful for engineering applications [4]. This also highlights one of the utility of SR of turbulent flows – which can construct these correction terms (visualized in the bottom row of Figure 4) with greater versatility than traditional approaches [5]. This is also why, in this work, we have paid special attention to evaluating SSIM_sgs and NRMSE_sgs, which are physics-based metrics related to these solver modifications.
> > >
> > > Please let us know if this manuscript requires further clarity, and we are always happy to discuss further with the reviewer.
> > >
> > > **References**
> > >
> > > 1. J. M. Ottino. Mixing, chaotic advection, and turbulence. Annual Review of Fluid Mechanics, 22(1):207–254, 1990.
> > > 2. R. D. Moser, N. P. Malaya, H. Chang, P. S. Zandonade, P. Vedula, A. Bhattacharya, and A. Haselbacher. Theoretically based optimal large-eddy simulation. Physics of Fluids, 21 (10): 105104, 2009.
> > > 3. M. Germano. Turbulence: The filtering approach. Journal of Fluid Mechanics, 238:325–336, 1992.
> > > 4. R. D. Moser, S. W. Haering, and G. R. Yalla. Statistical properties of subgrid-scale turbulence models. Annual Review of Fluid Mechanics, 53:255–286, 2021.
> > > 5. M. Bode, M. Gauding, Z. Lian, D. Denker, M. Davidovic, K. Kleinheinz, J. Jitsev, and H. Pitsch. Using physics-informed enhanced super-resolution generative adversarial networks for subfilter modeling in turbulent reactive flows. Proceedings of the Combustion Institute, 38(2):2617–2625, 2021
> > > 6. K. Fukami, K. Fukagata, and K. Taira. Super-resolution reconstruction of turbulent flows with machine learning. Journal of Fluid Mechanics, 870:106–120, 2019.
> > > 7. S. B. Pope. Turbulent Flows. Cambridge University Press, 2000.
> > > 8. M. Germano, U. Piomelli, P. Moin, and W. H. Cabot. A dynamic subgrid-scale eddy viscosity model. Phys. Fluids A, 3(7):1760–1765, 1991.
> > > 9. A. W. Vreman. An eddy-viscosity subgrid-scale model for turbulent shear flow: Algebraic theory and applications. Phys. Fluids, 16(10):3670–3681, 2004.

---

> > > > ### Comment · Reviewer_5NmK · 2023-08-24
> > > >
> > > > Thanks authors for the detailed reply. I am still trying to get how to use the trained model in practice. If we cannot get paired LR solutions through simulations during training due to the chaotic nature of the system, it means that in practice, when we want to accelerate simulation, we also cannot perform low-resolution simulation due to the same reason in training, otherwise the input distribution will be different from what model saw during training. Then in this case, once the model is trained using this dataset, the only use case in practice is to use sensors to observe the fluid flow to get LR fields and then use the trained model to super-resolved the fluid field. Am I correctly understanding the use case?

---

> > > > > ### Author Response · Authors · 2023-08-24
> > > > > **3rd Response on Low Resolution Inputs during Test Time**
> > > > >
> > > > > We thank the reviewer for further review of the low-resolution (LR) inputs. Errors in chaotic numerical systems arise from single-step errors between timesteps that become large errors as these single-step errors accumulate and propagate [1]. This means if the single-step errors are reduced sufficiently by the correction terms generated by the SR models, the flowfield in the next timestep will not have significant propagated errors that changes the distribution of the new SR input  – which is also the basis of many autoregressive ML approaches [2,3,4] during test time. We acknowledge that this relies on the SR model’s ability to reduce errors sufficiently well between single steps. We illustrate that this single-step error (that result from coarse grids) can be mitigated with our present SR approach in Figure 17 (Appendix E.1.6). In this figure, the SR model (trained on filtered data) is shown to recover ground truth spectral distribution of turbulent kinetic energy (in a homogeneous isotropic turbulence configuration) sufficiently well when fed with both coarse simulation and filtered inputs with similar spectral distributions (but different instantaneous flowfields).
> > > > >
> > > > > Let us know if there are further concerns regarding this.
> > > > >
> > > > > Edit 1: Added a title to distinguish from other responses
> > > > > Edit 2: Added year to Ref [3]
> > > > >
> > > > > **References**
> > > > > 1. J. M. Ottino. Mixing, chaotic advection, and turbulence. Annual Review of Fluid Mechanics, 22(1):207–254, 1990.
> > > > > 2. K. Stachenfeld, D. B. Fielding, D. Kochkov, M. Cranmer, T. Pfaff, J. Godwin, C. Cui, S. Ho, P. Battaglia, and A. Sanchez-Gonzalez. Learned simulators for turbulence. In The Tenth International Conference on Learning Representations, 2022.
> > > > > 3. A. Van den Oord, N. Kalchbrenner, L. Espeholt, L., O. Vinyals, and A. Graves. Conditional image generation with pixelcnn decoders. Advances in Neural Information Processing Systems, 29, 2016.
> > > > > 4. A. Radford, K. Narasimhan, T., Salimans, and I. Sutskever. Improving language understanding by generative pre-training, 2018.

---

> > > > > > ### Comment · Reviewer_5NmK · 2023-08-25
> > > > > >
> > > > > > Thanks for pointing out Figure 17. For the filtered input, the super-resolved output looks very similar to the ground truth (this is reasonable, actually you can see the filtered input itself is quite similar to the ground truth except that it is more blurry). For the low-resolution simulation input (implicit LES), can you show visually how similar the SR output is with the ground truth (DNS result)? Even though the spectral distribution is similar, I think MSE and visual comparison is a more direct and intuitive way to evaluate the super-resolution result.  My understanding is that if simulated low-resolution flow is very different from filtered ground truth, then the SR model won't recover the ground truth well. So that's why I was asking, when using the trained SR model in reality, whether we should use a sensor to obtain a low-resolution flow to mimic the filtered ground truth, and then use SR model to recover it. Otherwise, if the model takes in simulated low-resolution flow, and the output is still not visually similar to the ground truth, how this could be useful?

---

> > > > > > > ### Author Response · Authors · 2023-08-25
> > > > > > > **4th Response on Usefulness of Coarse Simulations and Integrating Sensor Data with Coarse Simulations**
> > > > > > >
> > > > > > > We thank the reviewer for further inquiries of the direct comparisons of low-resolution (LR) and filtered data.
> > > > > > >
> > > > > > > 1. **On the usefulness of coarse-grid simulations with accurate statistical/spectral behavior but different instantaneous data**  We appreciate that the reviewer has recognized that our SR model is capable of recovering turbulent spectral distributions.  We would like to point out that for many practical engineering and ecological flow systems of interest, recovering statistical and spectral distributions correctly is sufficient for coarse-grid simulations to be useful even if instantaneous flow fields differ. For example, coarse-grid simulations that manage to capture these turbulent distributions have been demonstrated to evaluate important global quantities such as lift-coefficients and averaged velocity fields that are essential for designing aircraft and skyscrapers [1,2]. This is why large-eddy simulations and turbulence modeling remains an active simulation technique within flow physics, even though coarse simulations cannot fully recover instantaneous flowfields of a corresponding direct numerical simulation due to the chaotic nature of turbulent flows [3]. This is why we believe that the turbulent kinetic energy spectra evaluated in Figure 17 is a suitable demonstration of the usefulness of these SR models on coarse simulations.
> > > > > > >
> > > > > > > However we do agree that these coarse simulations cannot be applied to predict time-dependent quantities (such as rare physical events) that rely on instantaneous flowfields without careful treatment, which we will address below.
> > > > > > >
> > > > > > > 2. **On Assimilating Sensor Data** We would also like to thank the reviewer for raising the idea of potentially assimilating sensor information into the coarse-grid simulation to mimic filtered DNS as a pathway for fully-utilizing the SR model. In fact, one of our co-authors (M. Ihme) is a pioneer in assimilating sensor data into coarse/large-eddy simulations of 3D compressible turbulent flow conditions to overcome our aforementioned challenges related to chaotic flows [4,5,6]. This approach would indeed help create better matches between high- and low-resolution simulations but we note that large amounts of computational resources and large ensembles of sensor data are typically required to achieve this. Thus, the full potential of this assimilating method has still yet to be fully investigated for 3D compressible turbulent flows, i.e., the sensor method suggested by the reviewer is currently not sufficiently mature for a full-fledged benchmark study in today’s computational and data landscape. However, we do believe that the framework for data curation presented in this work can lead to the datasets needed for an SR demonstration of this nature in a future work.
> > > > > > >
> > > > > > > We hope that this response address the reviewer’s concerns and we are open to further discussion, if needed.
> > > > > > >
> > > > > > > **References**
> > > > > > >
> > > > > > > 1. K. Goc, O. Lehmkuhl, G. Park, S. Bose, and P. Moin. Large eddy simulation of aircraft at affordable cost: A milestone in computational fluid dynamics. Flow 1:E14, 2021.
> > > > > > >
> > > > > > > 2. H. Xiao, and A.S. Morgans. Attenuation of the unsteady loading on a high-rise building using top-surface open-loop control. Journal of Fluid Mechanics 968:A17, 2023.
> > > > > > >
> > > > > > > 3. R. D. Moser, S. W. Haering, and G. R. Yalla. Statistical properties of subgrid-scale turbulence models. Annual Review of Fluid Mechanics, 53:255–286, 2021.
> > > > > > >
> > > > > > > 4. J.W. Labahn, H. Wu, S.R. Harris, B. Coriton, J.H. Frank, and M. Ihme. Ensemble Kalman filter for assimilating experimental data into large-eddy simulations of turbulent flows. Flow, Turbulence and Combustion, 104:861-893, 2019.
> > > > > > >
> > > > > > > 5. H. Yu, T. Jaravel, M. Ihme, M. P. Juniper, and L. Magri. Data assimilation and optimal calibration in nonlinear models of flame dynamics. Journal of Engineering for Gas Turbines and Power, 141(12):121010, 2019.
> > > > > > >
> > > > > > > 6. J.W. Labahn, H. Wu, B. Coriton, J. H. Frank, and M. Ihme. Data assimilation using high-speed measurements and LES to examine local extinction events in turbulent flames. Proceedings of the Combustion Institute 37(2):2259-2266,2019.

---

> > > > > > > > ### Comment · Reviewer_5NmK · 2023-08-26
> > > > > > > >
> > > > > > > > Thanks for the explanation. Based on discussion with the authors, I am convinced that in flow physics, even instantaneous LR and HR flow fields are not similar (which means the SR model cannot recover HR predictions that are visually similar to the DNS ground truth), it could also somehow be useful to super-resolve the LES simulation to make some distribution match the HR flow for some further analysis. I will increase my score to 7 to support the acceptance of this paper.

---

> > > > > > > > > ### Author Response · Authors · 2023-08-26
> > > > > > > > > **5th Response**
> > > > > > > > >
> > > > > > > > > We thank the reviewer for the support, and are grateful that the reviewer has raised the score for this manuscript!

---

### Official Review · Reviewer_yabe · 2023-07-23
**This paper presents an impressive 3D volumetric super-resolution dataset, but the experiments and evaluation in the benchmark need to be enhanced.**

**Rating:** 6
**Confidence:** 5
**Clarity:** Yes, this paper is well-written.

**Strengths:**

- This 3D turbulence dataset is large-scale and high-fidelity and has good promise for future research in scientific machine learning.

- This paper includes a discussion of neural scaling analysis. It is crucial for analyzing model scalability, especially for the super-resolution of high-dimensional scientific data.

- This paper considers the physics-informed loss and compares the performance between the data-driven method and the physics-informed strategy.

**Additional Feedback:**

No.

**Correctness:**

Generally, the claims in the manuscript are correct. However, I have one concern regarding the authors' claim in the Conclusion.

On Page 9, lines 295-299, the authors show the benefits of physics-informed loss in a large-scale problem. However, this specific task may not be sufficient evidence to refute the general claim [1] regarding the limitations of physics-informed machine learning in small model scenarios. There are two main reasons for this. Firstly, this paper only considers the gradient loss, while typical PINN methods incorporate multiple physics loss terms, such as governing equations and boundary conditions. Including these additional terms increases computational memory and cost compared to using just one gradient loss. For instance, dealing with second-order derivative terms and multiple physics loss terms in 2D wave propagation problems poses challenges to computational resources due to large computational graphs. Therefore, if the authors were to include multiple loss terms in this paper, such as the governing equations in Eq. (1a-1d), it could also present a challenge. Secondly, the network architectures are different. PINNs typically consider fully-connected neural networks [1] while the networks in this paper are conv-based.

Reference:

[1] Karniadakis, G. E., Kevrekidis, I. G., Lu, L., Perdikaris, P., Wang, S., & Yang, L. (2021). Physics-informed machine learning. Nature Reviews Physics, 3(6), 422-440.

**Documentation:**

Yes, the data details are sufficient.

**Ethics:**

No.

**Limitations:**

The authors have provided a paragraph discussing the limitations of this paper. Please also see my comments in **Opportunities For Improvement** and **Correctness**.

**Opportunities For Improvement:**

The experiments and evaluations included in the benchmark are not comprehensive and need to be enhanced.

- **Downsampling**: On Page 5, the authors use Favre filtering to conduct downsampling. Since this dataset is from simulation, is it possible to directly simulate the corresponding low-resolution simulation data and do super-resolution? It would better align with real-world applications.

- **Loss function**: On Page 6, this paper considers MSE as the loss function based on the previous work [1,2]. However, in the computer vision community, researchers use L1 loss more frequently [3,4] due to empirically better performance. Also, in the community of scientific machine learning, some papers [5,6] also apply L1 loss as the standard loss function.

- **Physics-informed loss**: this paper considers the gradient terms as a loss regularizer besides data loss. I agree that incorporating the gradient information can definitely help the training but I am not sure the gradient loss can be considered a typical physics-informed loss. Based on Section F.3 in the supplementary, I think it is more like an implicit inductive bias instead of explicit "physics loss". An example of physics loss in fluid flow study can be found in [7]. Also, how do the authors calculate the gradient terms in the context of super-resolution?

- **Evaluation metrics**: this paper only considers SSIM as an evaluation metric, which is good to measure the multi-scale features of turbulence dynamics. However, it would strengthen the paper and convince the readers to also consider the relative l2 norm [1,5,6] and some domain-specific metrics [5,7] that measure the physical properties. Since the objective is scientific data, it would be good to understand more about pixel-wise differences and the preservation of physical properties.

References:

[1] Fukami, K., Fukagata, K., & Taira, K. (2019). Super-resolution reconstruction of turbulent flows with machine learning. Journal of Fluid Mechanics, 870, 106-120.

[2] Bode, M., Gauding, M., Lian, Z., Denker, D., Davidovic, M., Kleinheinz, K., ... & Pitsch, H. (2021). Using physics-informed enhanced super-resolution generative adversarial networks for subfilter modeling in turbulent reactive flows. Proceedings of the Combustion Institute, 38(2), 2617-2625.

[3] Lim, B., Son, S., Kim, H., Nah, S., & Mu Lee, K. (2017). Enhanced deep residual networks for single image super-resolution. In Proceedings of the IEEE conference on computer vision and pattern recognition workshops (pp. 136-144).

[4] Liang, J., Cao, J., Sun, G., Zhang, K., Van Gool, L., & Timofte, R. (2021). Swinir: Image restoration using swin transformer. In Proceedings of the IEEE/CVF international conference on computer vision (pp. 1833-1844).

[5] Esmaeilzadeh, S., Azizzadenesheli, K., Kashinath, K., Mustafa, M., Tchelepi, H. A., Marcus, P., ... & Anandkumar, A. (2020, November). Meshfreeflownet: A physics-constrained deep continuous space-time super-resolution framework. In SC20: International Conference for High-Performance Computing, Networking, Storage and Analysis (pp. 1-15). IEEE.

[6] Ren, P., Rao, C., Liu, Y., Ma, Z., Wang, Q., Wang, J. X., & Sun, H. (2022). Physics-informed deep super-resolution for spatiotemporal data. arXiv preprint arXiv:2208.01462.

[7] Wang, R., Kashinath, K., Mustafa, M., Albert, A., & Yu, R. (2020, August). Towards physics-informed deep learning for turbulent flow prediction. In Proceedings of the 26th ACM SIGKDD International Conference on Knowledge Discovery & Data Mining (pp. 1457-1466).

**Relation To Prior Work:**

Yes, this paper clearly discussed the difference between this paper and the existing work.

**Summary And Contributions:**

This paper presents a high-fidelity 3D volumetric super-resolution dataset. The authors have provided a benchmark to test baseline models, perform neural scaling analysis and assess the incorporation of physics-informed loss. This paper finds that (1) the predictive performance can scale with model size and cost, (2) the choice of architecture is important and (3) it is beneficial to incorporate physics-informed loss.

---

> ### Author Response · Authors · 2023-08-20
> **Our Response**
>
> We appreciate the comments provided by the reviewer, especially for recognizing the fidelity and scale of our dataset,  the choice of physics-based approach in this benchmark, our neural scaling analysis, and clarity and good documentation of this work. Regarding the opportunities for improvement:
>
> 1. **Downsampling:**
>
> We would like to clarify that within this work, we are not simply downsampling the higher-resolution flowfields to low-resolution flowfields. While we agree that having a low-high resolution pair would be ideal, it is not feasible to obtain low-high resolution pairs due to time-dependency and stochastic nature of both low- and high-resolution simulations. This is especially true for chaotic systems such as turbulent flows [1], where minor changes in configuration (such as grid size) can lead to vastly different flow behavior. Filtered (specifically Favre-filtered for compressible flows) DNS is a canonical surrogate [2] for coarse-grid simulations, with strong theoretical backing dating back to the 1990s [3,4], and is widely used [5,6] as training inputs for SR problems. We have included this discussion in Section 2, Section 3.2, and acknowledged this as a limitation in Section 6. To further strengthen our work, we provided additional analysis in Appendix E.1.6 in the supplementary material to demonstrate that an ML model trained on filtered DNS can be employed towards SR of coarse-grid simulations.
>
> 2. **Loss Function:**
>
> We recognize that L1 losses are preferred for computer vision. However, during preliminary hyper-parameter searches, we found that differences in  end results for both losses were negligible. To demonstrate this, we include a validation MSE with one of the models  trained with both L1 and L2 losses in the Appendix F.3 in the supplementary material. Ultimately, we chose to employ L2 loss since it penalizes outlier errors that could be detrimental to simulation-related problems.
>
> 3. **Physics-informed Loss:**
>
> We agree with the reviewer that our definitions of “physics-informed” methods can be a bit more nuanced. An issue that we’ve found when writing this manuscript is that the meaning of the term “physics-informed” varies widely within the literature. For example , “physics-informed neural networks” [7] has a restricted definition, where the loss term is modified by the residual of the entire PDE. This definition is close to the reviewer’s definition. In another flow-physics based work, physics-informed losses are used to refer to the residuals of specific transport operators (such as continuity) [6]. In a review of physics-informed ML [8], models with implicit biases related to physics are often considered a category of physics-informed ML approaches. Since, our gradient-based loss contains information regarding the advection operator and leverages implicit bias (both of which are noticed by the reviewer), we chose to consider this modification to be a physics-informed loss in the initial manuscript. However, we agree with the reviewer that more context could be provided to differentiate between implicit and explicit physics-based ML approaches. Thus, we have modified section 2 to include a more detailed discussion of this. We have also changed the term “physics-informed” to “physics-based” to further distinguish from PINN related work, and softened the conclusions to only consider this one instance of physics-based approach.
> We have also now included more information regarding gradient computation in Appendix F.3 in the supplementary material. The gradient terms are evaluated using torch.gradient, which corresponds to a second-order central differencing scheme that is optimized for GPU calculations. This is done on both super-resolved and ground truth fields before inputting the gradient terms into the MSE function.
>
> Edit 1: Fixed references 7 -> 6, 9 -> 8.

---

> > ### Author Response · Authors · 2023-08-20
> > **References**
> >
> > 1. J. M. Ottino. Mixing, chaotic advection, and turbulence. Annual Review of Fluid Mechanics, 22(1):207–254, 1990.
> > 2. R. D. Moser, N. P. Malaya, H. Chang, P. S. Zandonade, P. Vedula, A. Bhattacharya, and A. Haselbacher. Theoretically based optimal large-eddy simulation. Physics of Fluids, 21 (10): 105104, 2009.
> > 3. M. Germano. Turbulence: The filtering approach. Journal of Fluid Mechanics, 238:325–336, 1992.
> > 4. R. D. Moser, S. W. Haering, and G. R. Yalla. Statistical properties of subgrid-scale turbulence models. Annual Review of Fluid Mechanics, 53:255–286, 2021.
> > 5. M. Bode, M. Gauding, Z. Lian, D. Denker, M. Davidovic, K. Kleinheinz, J. Jitsev, and H. Pitsch. Using physics-informed enhanced super-resolution generative adversarial networks for subfilter modeling in turbulent reactive flows. Proceedings of the Combustion Institute, 38(2):2617–2625, 2021
> > 6. K. Fukami, K. Fukagata, and K. Taira. Super-resolution reconstruction of turbulent flows with machine learning. Journal of Fluid Mechanics, 870:106–120, 2019.
> > 7. M. Raissi, P. Perdikaris, and G. Karniadakis. Physics-informed neural networks: A deep learning framework for solving forward and inverse problems involving nonlinear partial differential equations. J. Comp. Phys., 378:686–707, 2019.
> > 8. G. E. Karniadakis, I. G. Kevrekidis, L. Lu, P. Perdikaris, S. Wang, and L. Yang. Physics-informed machine learning. Nature Reviews Physics, 3:422–440, 2021.

---

> > > ### Comment · Reviewer_yabe · 2023-08-28
> > >
> > > Thanks for the authors' responses. I am fine with the downsampling part. However, I don't quite understand the authors' claim:
> > >
> > > > Ultimately, we chose to employ L2 loss since it penalizes outlier errors that could be detrimental to simulation-related problems.
> > >
> > > This paper is doing super-resolution for simulation data. I cannot see where the "outlier" comes from. I suspect the "outlier" in the response means the extreme value in turbulence. If so, those extreme values in turbulence should be captured in super-resolution models since they are essential indexes.

---

> > > > ### Author Response · Authors · 2023-08-28
> > > > **2nd Response on L2 loss**
> > > >
> > > > We thank the reviewer for recognizing our justification for filtering and downsampling, and for further discussions on the L2 loss. Allow us to provide more clarity to our response.
> > > >
> > > > One key difference between L1 (min Σ|Target - Prediction|) and L2 loss (min Σ[Target - Prediction]^2) lies in the squaring of the errors in L2 loss. Since L2 loss minimizes the square of the errors, this would, in principle, discourage the presence of individual voxels with large errors (a.k.a outliers in prediction) in the flowfield, thereby preventing the presence of spurious discontinuities within the flowfield. We felt this was important in a compressible flow problem since natural discontinuities (a.k.a shockwaves) do occur within the flowfield [1].
> > > >
> > > > Nevertheless, we agree with the reviewer that an L1 loss should be used, if it was demonstrated to be superior to the L2 loss. However, negligible differences were found between L1 and L2 losses during our hyperparameter search, as previously mentioned, so we favored a loss function that discouraged the presence of outlier voxels to better avoid the presence of spurious physical phenomena in the ML predictions.
> > > >
> > > > We hope this addresses the reviewer’s question. Let us know if the reviewer has further concerns regarding this.
> > > >
> > > > Edit 1: Added "|" to "Σ|Target - Prediction|". Edit 2: you -> the reviewer
> > > >
> > > > **Reference**
> > > > 1. E. Johnsen, J. Larsson, A.V. Bhagatwala, W.H. Cabot, P. Moin, B.J. Olson, P.S. Rawat, S.K. Shankar SK, B. Sjögreen, H.C. Yee HC, X. Zhong. Assessment of high-resolution methods for numerical simulations of compressible turbulence with shock waves. Journal of Computational Physics 229(4):1213-1237, 2010.

---

### Official Review · Reviewer_AcWj · 2023-07-28
**The dataset could be highly valuable but the benchmarking has significant weakness.**

**Rating:** 7
**Confidence:** 5
**Correctness:** 1. SSIM is not sound as the sole metr…
**Clarity:** The paper is well written.

**Strengths:**

1. The dataset is for 3D simulations which is more complex than 2D simulations and can capture more realistic flow behavior.
2. DNS of compressible reacting flows involves solving the full set of governing equations for fluid dynamics and chemical reactions without making any simplifying assumptions or turbulence models. These simulations are computationally intensive and complex, requiring significant computational resources and time to carry out accurately. The availability of such a high-fidelity dataset can significantly foster ML research in this challenging scientific domain.

**Additional Feedback:**

See above. Addressing these weaknesses and refining the benchmarking approach would enhance the research's impact and contribution to the scientific community.

**Documentation:**

The dataset is well documented.

**Limitations:**

See above for limitations. Additionally, the lack of a mechanism to generate more simulations and expand the dataset is a significant limitation as the solvers are proprietary.

**Opportunities For Improvement:**

While the dataset contribution is strong, the benchmarking approach in the paper has notable weaknesses that could benefit from significant improvement.
1. The paper's choice of three SR models based on ResNet, popular in computer vision, might not be the most suitable for scientific applications like fluid dynamics. It is important to consider recent advancements in scientific machine learning, such as Neural Operators (e.g., FNO), MeshGraphNets, NUNet, among others. FNO for example is designed specifically to handle discretization invariance and may be more relevant for SR in this domain. Discussing and including these approaches in the benchmark would provide a more comprehensive evaluation of the performance of SR methods on the scientific dataset.
2. The metric SSIM commonly used in computer vision may not be appropriate for scientific applications. It would be beneficial to explore and adopt domain-specific evaluation metrics that align with the characteristics of compressible reacting flows. Metrics that take into account physical quantities, conservation laws, or turbulence characteristics would provide more meaningful insights into the accuracy and usefulness of SR models for these simulations.
3. The practice of downsampling from higher resolution to lower resolution, common in computer vision, might not accurately represent scientific simulations' real-world scenarios. Instead, it would be more appropriate to use low-resolution simulations as input for SR models. Coarse simulations lack fine-scale turbulence present in higher resolutions, making the low-resolution input a more realistic baseline. Additionally, considering RANS simulations on a coarse grid could provide a representation of noisy input due to turbulence modeling, mimicking real-world conditions more accurately.

**Relation To Prior Work:**

While the dataset prior work is discussed, the paper is missing major bodies of work in SR for SciML.

**Summary And Contributions:**

This paper presents BLASTNet 2.0, describing the improvements and enhancements made to the original BLASTNet increasing the number of simulations from 10 simulations of 34 simulations for 3D compressible reacting/non-reacting flows. A benchmark for superresolution (SR) on 3 ResNet-based models is also presented on a smaller subset of the 2.0 dataset.

---

> ### Author Response · Authors · 2023-08-20
> **Our Response**
>
> We thank the reviewer for recognizing the scope and utility of this dataset, and for recognizing that this work is well-written. ​​We appreciate the reviewer’s comments for improving this work, which was addressed as follows:
>
> 1. **Inclusion of more scientific ML approaches:**
>
> We have now benchmarked Conv-FNO models (9 variations, 27 runs), as discussed in Section 5. In total, we benchmarked 2 types of scientific ML approaches (Conv-FNO and gradient-based loss), as well as 3 types of vanilla ResNet-based approaches with different architecture paradigms, which we believe is a good balance for readers interested in both scientific and vanilla ML approaches.
>
> In addition, we have included an extensive discussion on scientific ML including MeshGraphNet, NUNet, FNO, Conv-FNO models, as well as physics-based loss functions models, in Section 2. However, we note that several authorities [1,2] within ML for flow physics agree that convolutional networks are suitable for flow physics, as the convolutional layers possess an inductive bias that is suitable for problems involving spatial grids, even outside of computer vision. This, coupled with the performance optimizations via numerous open ML challenges since the release of ResNet in 2015, have resulted in highly-optimized model architectures that we believe are suitable for this benchmark. This is now included in Section 4 of the revised manuscript.
>
>
> 2. **Addition of Scientific Metrics and Validity of SSIM:**
>
> We now included metrics that evaluate the global physical properties based on the (i) volume-averaged kinetic energy (momentum component of energy conservation within a fixed control volume), as well as the (ii)  turbulent dissipation rate (conversion of turbulent energy to heat and related to the smallest length-scales), in addition with the metrics used to evaluate the (iii) unresolved turbulent stresses in the original manuscript. We have also now included metrics that evaluate the normalized root-mean-squared error to provide an alternative to the SSIM method.
>
> However, we note that SSIM is not a metric that is strictly limited to computer vision, and is commonly used to evaluate flow predictions [3,4]. In fact, the use of multiple statistical quantities such as mean, variance, and covariance of moving windows is suited for evaluating the statistical structure of turbulence (involving mean and fluctuating velocity components) [5] . We have now included this in section 4.2.
>
>
> 3. **Justification for using Downsampled Favre-filtered DNS data with SR of flow physics:**
>
> We believe that our choice of inputs and outputs for SR is represent coarse grid simulations (also known as implicit large-eddy simulations (LES)) In many practical engineering analyses, coarse-grained simulations with under-resolved grid sizes are commonly used to bypass the extensive costs of fully-resolved direct numerical simulations [6,7,8,9,10].
> While we agree that having coarse simulation data (RANS or LES) as input would be ideal, it is not feasible to obtain low-high resolution pairs due to time-dependency and stochastic nature of both high- and low-resolution simulations. This is especially true for chaotic systems such as turbulent flows [6], where minor changes in configuration (such as grid size) can lead to vastly different flow behavior. Filtered DNS is a canonical surrogate [7] for coarse-grid simulations, with strong theoretical backing dating back to the 1990s [8,9], and is widely used [10,11] as training inputs for SR problems. We have included this discussion in Section 2, Section 3.2, and acknowledged this as a limitation in Section 6. To further strengthen our work, we provided additional analysis in Appendix E.1.6 in the supplementary material to demonstrate that an ML model trained on filtered DNS can be employed towards SR of coarse-grid simulations.

---

> > ### Author Response · Authors · 2023-08-20
> > **References**
> >
> > 1. Karniadakis, G. E., Kevrekidis, I. G., Lu, L., Perdikaris, P., Wang, S., & Yang, L.. Physics-informed machine learning. Nature Reviews Physics, 3(6), 422-440.
> > 2. D. Kochkov, J. A. Smith, A. Alieva, Q. Wang, M. P. Brenner, and S. Hoyer. Machine learning-accelerated computational fluid dynamics. Proceedings of the National Academy of Sciences, 118(21):e2101784118, 2021.
> > 3. T. Bao, S. Chen, T. T. Johnson, P. Givi, S. Sammak, and X. Jia. Physics guided neural networks for spatio-temporal super-resolution of turbulent flows. In J. Cussens and K. Zhang, editors, Proceedings of the Thirty-Eighth Conference on Uncertainty in Artificial Intelligence, volume 180, pages 118–128, 2022.
> > 4. A. Glaws, R. King, and M. Sprague. Deep learning for in situ data compression of large turbulent flow simulations. Physical Review Fluids, 5:114602, 2020.
> > 5. S. B. Pope. Turbulent Flows. Cambridge University Press, 2000.
> > 6. J. M. Ottino. Mixing, chaotic advection, and turbulence. Annual Review of Fluid Mechanics, 22(1):207–254, 1990.
> > 7. R. D. Moser, N. P. Malaya, H. Chang, P. S. Zandonade, P. Vedula, A. Bhattacharya, and A. Haselbacher. Theoretically based optimal large-eddy simulation. Physics of Fluids, 21 (10): 105104, 2009.
> > 8. M. Germano. Turbulence: The filtering approach. Journal of Fluid Mechanics, 238:325–336, 1992.
> > 9. R. D. Moser, S. W. Haering, and G. R. Yalla. Statistical properties of subgrid-scale turbulence models. Annual Review of Fluid Mechanics, 53:255–286, 2021.
> > 10. M. Bode, M. Gauding, Z. Lian, D. Denker, M. Davidovic, K. Kleinheinz, J. Jitsev, and H. Pitsch. Using physics-informed enhanced super-resolution generative adversarial networks for subfilter modeling in turbulent reactive flows. Proceedings of the Combustion Institute, 38(2):2617–2625, 2021.
> > 11. K. Fukami, K. Fukagata, and K. Taira. Super-resolution reconstruction of turbulent flows with machine learning. Journal of Fluid Mechanics, 870:106–120, 2019.

---

> > ### Comment · Reviewer_AcWj · 2023-08-28
> >
> > The revision incorporates many of the suggestions by the reviewers. The paper has improved in quality, specifically the addition of the FNO-variant and experimental justification for the downsampling method. So, I'm raising my score from 5 to 7.

---

> > > ### Author Response · Authors · 2023-08-28
> > > **2nd Response**
> > >
> > > We thank the reviewer for recognizing our efforts in adding the FNO variant and further information on the downsampling approach. We are very grateful that the reviewer has raised the score for this work!

---

### Author Response · Authors · 2023-08-20
**Overall Response to Reviewers**

We thank the reviewers for appreciating the value, fidelity, diversity, scale, and  convenience of our dataset. We are grateful for the reviewers’ positive notes on our use of neural scaling analysis to benchmark the performance of the 3D super-resolution (SR) models.

We appreciate the reviewers’ comments on opportunities for improving this manuscript, especially on strengthening the benchmark aspect of this work. Based on this feedback, we have addressed and incorporated all discussed opportunities in the attached revised manuscript. A pdf highlighting the difference between original and revised manuscript is provided in the supplementary material as “difference_in_revision.pdf”. The original supplementary material, i.e., appendices are now in “supplementary_material.pdf”. The following revisions are summarized:

1. **Additions to Scientific ML Approaches:** Several reviewers commented on the need for discussing and benchmarking more scientific ML approaches, especially FNO-based models, on top of our gradient-based ML model. To address this, we now included results from benchmarking the 9 variants of Conv-FNO models (in addition to the 40 benchmarked models) in Section 5, as well as a more comprehensive overview of scientific ML approaches in Section 2.

2. **Additional Evaluations:** Several reviewers have commented on the need for more metrics (L2 based and physics-based) and figures. We have now included results from these evaluations via Table 3, as well as the supplementary material, in the revised manuscript.

3. **Justification for using Favre-filtered DNS data with SR of flow physics:** We appreciate the reviewers’ suggestion for matching low resolution simulations with the direct numerical simulation (DNS) data from BLASTNet to construct an SR dataset. While we agree that this would have been ideal, it is not feasible to obtain low-high resolution pairs due to time-dependency and stochastic nature of both high- and low-resolution simulations. This is especially true for chaotic systems such as turbulent flows [1], where minor changes in configuration (such as grid size) can lead to vastly different flow behavior. Filtered DNS is a canonical surrogate [2] for coarse-grid simulations (also known as implicit large-eddy simulations, a widely used fluid simulation approach [2,4,5]), with strong theoretical backing dating back to the 1990s [3,4], and is widely used [5,6] as training inputs for SR problems. We have included this discussion in Section 2, Section 3.2, and acknowledged this as a limitation in Section 6. To further strengthen our work, we provided additional analysis in Appendix E.1.6 in the supplementary material to demonstrate that an ML model trained on filtered DNS can be employed towards SR of coarse-grid simulations.

4. **Justification of Turbulent SR as a useful ML task:** Following one of the reviewer’s request to justify the usefulness of SR tasks in experimental and simulation flow physics, we now included a discussion on the use of SR in studies involving experimental techniques (Schlieren [7], tomography [8], particle-image velocimetry [9]). We also extended our discussion on the use of SR-based models for improving coarse-grid computational fluid dynamics simulations, which has been an emerging area of interest in ML for as demonstrated by numerous related work discussed in Section 2.

5. **On the use of the term “Physics-informed”:** There was also a comment on avoiding the use of the broad term  “physics-informed” ML and loss, to avoid association with a specific framework known as “physics-informed neural networks'' and related approaches. We have modified the term “physics-informed” to “physics-based”, and extended on section 2 to distinguish the language a bit further.

6. **Minor revisions:** We have made minor changes to the manuscript to fit in the additional results and discussion within the new 10 page limit, correct minor typos, and improve readability.

With these revisions, we believe these changes address the main concerns raised by the reviewers and look forward to addressing any additional questions during the discussion period.

---

> ### Author Response · Authors · 2023-08-20
> **References**
>
> 1. J. M. Ottino. Mixing, chaotic advection, and turbulence. Annual Review of Fluid Mechanics, 22(1):207–254, 1990.
> 2. R. D. Moser, N. P. Malaya, H. Chang, P. S. Zandonade, P. Vedula, A. Bhattacharya, and A. Haselbacher. Theoretically based optimal large-eddy simulation. Physics of Fluids, 21 (10): 105104, 2009.
> 3. M. Germano. Turbulence: The filtering approach. Journal of Fluid Mechanics, 238:325–336, 1992.
> 4. R. D. Moser, S. W. Haering, and G. R. Yalla. Statistical properties of subgrid-scale turbulence models. Annual Review of Fluid Mechanics, 53:255–286, 2021.
> 5. M. Bode, M. Gauding, Z. Lian, D. Denker, M. Davidovic, K. Kleinheinz, J. Jitsev, and H. Pitsch. Using physics-informed enhanced super-resolution generative adversarial networks for subfilter modeling in turbulent reactive flows. Proceedings of the Combustion Institute, 38(2):2617–2625, 2021
> 6. K. Fukami, K. Fukagata, and K. Taira. Super-resolution reconstruction of turbulent flows with machine learning. Journal of Fluid Mechanics, 870:106–120, 2019.
> 7. Z. Wang, X. Li, L. Liu, X. Wu, P. Hao, X. Zhang, and F. He. Deep-learning-based superresolution reconstruction of high-speed imaging in fluids. Physics of Fluids, 34(3):037107, 2022.
> 8. Z. Deng, C. He, Y. Liu, and K. C. Kim. Super-resolution reconstruction of turbulent velocity fields using a generative adversarial network-based artificial intelligence framework. Physics of Fluids, 31(12), 2019.
> 9. N. Janssens, M. Huysmans, and R. Swennen. Computed tomography 3D super-resolution with generative adversarial neural networks: Implications on unsaturated and two-phase fluid flow. Materials, 13(6):1397, 2020.

---

### Decision · Program_Chairs · 2023-09-22

**Decision:**

Accept (Poster)

**Comment:**

Paper delivers comprehensive data set of generic compressible turbulent flows.
Proof of concept for super-resolution reconstructions are provided.
Paper is clear and and reproducible, includes website and tutorial.
An extension to time series and more advanced super-resolution strategies is welcome future work,.